# Why Does Sharpness-Aware Minimization Generalize Better Than SGD?

**Zixiang Chen**[*] **Junkai Zhang**[*] **Yiwen Kou  Xiangning Chen  Cho-Jui Hsieh  Quanquan Gu**
Department of Computer Science
University of California, Los Angeles
Los Angeles, CA 90095
{chenzx19,zhang,evankou,xiangning,chohsieh,qgu}@cs.ucla.edu

## Abstract

The challenge of overfitting, in which the model memorizes the training data and fails to generalize to test data, has become increasingly significant in the training of large neural networks. To tackle this challenge, Sharpness-Aware Minimization (SAM) has emerged as a promising training method, which can improve the generalization of neural networks even in the presence of label noise. However, a deep understanding of how SAM works, especially in the setting of nonlinear neural networks and classification tasks, remains largely missing. This paper fills this gap by demonstrating why SAM generalizes better than Stochastic Gradient Descent (SGD) for a certain data model and two-layer convolutional ReLU networks. The loss landscape of our studied problem is nonsmooth, thus current explanations for the success of SAM based on the Hessian information are insufficient. Our result explains the benefits of SAM, particularly its ability to prevent noise learning in the early stages, thereby facilitating more effective learning of features. Experiments on both synthetic and real data corroborate our theory.

## 1 Introduction

The remarkable performance of deep neural networks has sparked considerable interest in creating ever-larger deep learning models, while the training process continues to be a critical bottleneck affecting overall model performance. The training of large models is unstable and difficult due to the sharpness, non-convexity, and non-smoothness of its loss landscape. In addition, as the number of model parameters is much larger than the training sample size, the model has the ability to memorize even randomly labeled data (Zhang et al., 2021), which leads to overfitting. Therefore, although traditional gradient-based methods like gradient descent (GD) and stochastic gradient descent (SGD) can achieve generalizable models under certain conditions, these methods may suffer from unstable training and harmful overfitting in general.

To overcome the above challenge, *Sharpness-Aware Minimization* (SAM) (Foret et al., 2020), an innovative training paradigm, has exhibited significant improvement in model generalization and has become widely adopted in many applications. In contrast to traditional gradient-based methods that primarily focus on finding a point in the parameter space with a minimal gradient norm, SAM also pursues a solution with reduced sharpness, characterized by how rapidly the loss function changes locally. Despite the empirical success of SAM across numerous tasks (Bahri et al., 2021; Behdin et al., 2022; Chen et al., 2021; Liu et al., 2022a), the theoretical understanding of this method remains limited.

Foret et al. (2020) provided a PAC-Bayes bound on the generalization error of SAM to show that it will generalize well, while the bound only holds for the infeasible average-direction perturbation instead of

---

[*]Equal contribution.

37th Conference on Neural Information Processing Systems (NeurIPS 2023).

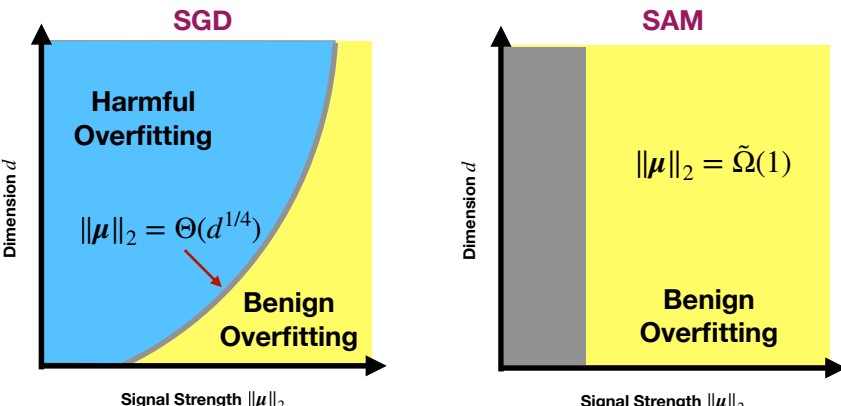

Figure 1: Illustration of the phase transition between benign overfitting and harmful overfitting. The yellow region represents the regime under which the overfitted CNN trained by SGD is guaranteed to have a small excess risk, and the blue region represents the regime under which the excess risk is guaranteed to be a constant order (e.g., greater than $0.1$). The gray region is the regime where the excess risk is not characterized.

practically used ascend-direction perturbation. Andriushchenko and Flammarion (2022) investigated the implicit bias of SAM for diagonal linear networks under global convergence assumption. The oscillations in the trajectory of SAM were explored by Bartlett et al. (2022), leading to a convergence result for the convex quadratic loss. A concurrent work (Wen et al., 2022) demonstrated that SAM can locally regularize the eigenvalues of the Hessian of the loss. In the context of least-squares linear regression, Behdin and Mazumder (2023) found that SAM exhibits lower bias and higher variance compared to gradient descent. However, all the above analyses of SAM utilize the Hessian information of the loss and require the smoothness property of the loss implicitly. The study for non-smooth neural networks, particularly for the classification task, remains open.

In this paper, our goal is to provide a theoretical basis demonstrating when SAM outperforms SGD. In particular, we consider a data distribution mainly characterized by the signal $\boldsymbol{\mu}$ and input data dimension $d$, and prove the following separation in terms of test error between SGD and SAM.

**Theorem 1.1** (Informal statement of Theorems 3.2 and 4.1)**.** *Let $p$ be the strength of the label flipping noise. For any $\epsilon > 0$, under certain regularity conditions, with high probability, there exists $0 \leq t \leq T$ such that the training loss converges, i.e., $L_S(\mathbf{W}^{(t)}) \leq \epsilon$. Besides,*

1. ***For SGD**, when the signal strength $\|\boldsymbol{\mu}\|_2 \geq \Omega(d^{1/4})$, we have $L_{\mathcal{D}}^{0-1}(\mathbf{W}^{(t)}) \leq p + \epsilon$. When the signal strength $\|\boldsymbol{\mu}\|_2 \leq O(d^{1/4})$, we have $L_{\mathcal{D}}^{0-1}(\mathbf{W}^{(t)}) \geq p + 0.1$.*

2. ***For SAM**, provided the signal strength $\|\boldsymbol{\mu}\|_2 \geq \widetilde{\Omega}(1)$, we have $L_{\mathcal{D}}^{0-1}(\mathbf{W}^{(t)}) \leq p + \epsilon$.*

Our contributions are summarized as follows:

- We discuss how the loss landscape of two-layer convolutional ReLU networks is different from the smooth loss landscape and thus the current explanation for the success of SAM based on the Hessian information is insufficient for neural networks.

- To understand the limit of SGD, we precisely characterize the conditions under which benign overfitting can occur in training two-layer convolutional ReLU networks with SGD. To the best of our knowledge, this is the first benign overfitting result for neural network trained with mini-batch SGD. We also prove a phase transition phenomenon for SGD, which is illustrated in Figure 1.

- Under the conditions when SGD leads to harmful overfitting, we formally prove that SAM can achieve benign overfitting. Consequently, we establish a rigorous theoretical distinction between SAM and SGD, demonstrating that SAM strictly outperforms SGD in terms of generalization error. Specifically, we show that SAM effectively mitigates noise learning in the early stages of training, enabling neural networks to learn features more efficiently.

**Notation.** We use lower case letters, lower case bold face letters, and upper case bold face letters to denote scalars, vectors, and matrices respectively. For a vector $\mathbf{v} = (v_1, \cdots, v_d)^\top$, we denote

by $\|\mathbf{v}\|_2 := \left(\sum_{j=1}^d v_j^2\right)^{1/2}$ its 2-norm. For two sequence $\{a_k\}$ and $\{b_k\}$, we denote $a_k = O(b_k)$ if $|a_k| \leq C|b_k|$ for some absolute constant $C$, denote $a_k = \Omega(b_k)$ if $b_k = O(a_k)$, and denote $a_k = \Theta(b_k)$ if $a_k = O(b_k)$ and $a_k = \Omega(b_k)$. We also denote $a_k = o(b_k)$ if $\lim |a_k/b_k| = 0$. Finally, we use $\widetilde{O}(\cdot)$ and $\widetilde{\Omega}(\cdot)$ to omit logarithmic terms in the notation. We denote the set $\{1, \cdots, n\}$ by $[n]$, and the set $\{0, \cdots, n-1\}$ by $\overline{[n]}$, respectively. The carnality of a set $S$ is denoted by $|S|$.

## 2 Preliminaries

### 2.1 Data distribution

Our focus is on binary classification with label $y \in \{\pm 1\}$. We consider the following data model, which can be seen as a special case of sparse coding model (Olshausen and Field, 1997; Allen-Zhu and Li, 2022; Ahn et al., 2022).

**Definition 2.1.** Let $\boldsymbol{\mu} \in \mathbb{R}^d$ be a fixed vector representing the signal contained in each data point. Each data point $(\mathbf{x}, y)$ with input $\mathbf{x} = [\mathbf{x}^{(1)\top}, \mathbf{x}^{(2)\top}, \dots, \mathbf{x}^{(P)\top}]^\top \in \mathbb{R}^{P \times d}, \mathbf{x}^{(1)}, \mathbf{x}^{(2)}, \dots, \mathbf{x}^{(P)} \in \mathbb{R}^d$ and label $y \in \{-1, 1\}$ is generated from a distribution $\mathcal{D}$ specified as follows:

1. The true label $\widehat{y}$ is generated as a Rademacher random variable, i.e., $\mathbb{P}[\widehat{y} = 1] = \mathbb{P}[\widehat{y} = -1] = 1/2$. The observed label $y$ is then generated by flipping $\widehat{y}$ with probability $p$ where $p < 1/2$, i.e., $\mathbb{P}[y = \widehat{y}] = 1 - p$ and $\mathbb{P}[y = -\widehat{y}] = p$.
2. A noise vector $\boldsymbol{\xi}$ is generated from the Gaussian distribution $\mathcal{N}(\mathbf{0}, \sigma_p^2 \mathbf{I})$, where $\sigma_p^2$ is the variance.
3. One of $\mathbf{x}^{(1)}, \mathbf{x}^{(2)}, \dots, \mathbf{x}^{(P)}$ is randomly selected and then assigned as $y \cdot \boldsymbol{\mu}$, which represents the signal, while the others are given by $\boldsymbol{\xi}$, which represents noises.

The data distribution in Definition 2.1 has also been extensively employed in several previous works (Allen-Zhu and Li, 2020; Jelassi and Li, 2022; Shen et al., 2022; Cao et al., 2022; Kou et al., 2023). When $P = 2$, this data distribution aligns with the one analyzed in Kou et al. (2023). This distribution is inspired by image data, where the input is composed of different patches, with only a few patches being relevant to the label. The model has two key vectors: the feature vector and the noise vector. For any input vector $\mathbf{x} = [\mathbf{x}^{(1)}, \dots, \mathbf{x}^{(P)}]$, there is exactly one $\mathbf{x}^{(j)} = y\boldsymbol{\mu}$, and the others are random Gaussian vectors. For example, the input vector $\mathbf{x}$ can be $[y\boldsymbol{\mu}, \boldsymbol{\xi}, \dots, \boldsymbol{\xi}], \dots, [\boldsymbol{\xi}, \dots, y\boldsymbol{\mu}, \boldsymbol{\xi}]$ or $[\boldsymbol{\xi}, \dots, \boldsymbol{\xi}, y\boldsymbol{\mu}]$: the signal patch $y\boldsymbol{\mu}$ can appear at any position. To avoid harmful overfitting, the model must learn the feature vector rather than the noise vector.

### 2.2 Neural Network and Training Loss

To effectively learn the distribution as per Definition 2.1, it is advantageous to utilize a shared weights structure, given that the specific signal patch is not known beforehand. When $P > n$, shared weights become indispensable as the location of the signal patch in the test can differ from the location of the signal patch in the training data.

We consider a two-layer convolutional neural network whose filters are applied to the $P$ patches $\mathbf{x}_1, \cdots, \mathbf{x}_P$ separately, and the second layer parameters of the network are fixed as $+1/m$ and $-1/m$ respectively, where $m$ is the number of convolutional filters. Then the network can be written as $f(\mathbf{W}, \mathbf{x}) = F_{+1}(\mathbf{W}_{+1}, \mathbf{x}) - F_{-1}(\mathbf{W}_{-1}, \mathbf{x})$, where $F_{+1}(\mathbf{W}_{+1}, \mathbf{x})$ and $F_{-1}(\mathbf{W}_{-1}, \mathbf{x})$ are defined as

$$F_j(\mathbf{W}_j, \mathbf{x}) = m^{-1}\sum_{r=1}^m \sum_{p=1}^P \sigma(\langle \mathbf{w}_{j,r}, \mathbf{x}^{(p)}\rangle). \tag{1}$$

Here we consider ReLU activation function $\sigma(z) = \mathbb{1}(z \geq 0)z$, $\mathbf{w}_{j,r} \in \mathbb{R}^d$ denotes the weight for the $r$-th filter, and $\mathbf{W}_j$ is the collection of model weights associated with $F_j$ for $j = \pm 1$. We use $\mathbf{W}$ to denote the collection of all model weights. Denote the training data set by $\mathcal{S} = \{(\mathbf{x}_i, y_i)\}_{i \in [n]}$. We train the above CNN model by minimizing the empirical cross-entropy loss function

$$L_{\mathcal{S}}(\mathbf{W}) = n^{-1}\sum_{i \in [n]} \ell(y_i f(\mathbf{W}, \mathbf{x}_i)),$$

where $\ell(z) = \log(1 + \exp(-z))$.

### 2.3 Training Algorithm

**Minibatch Stochastic Gradient Descent.** For epoch $t$, the training data set $S$ is randomly divided into $H := n/B$ mini batches $\mathcal{I}_{t,b}$ with batch size $B \geq 2$. The empirical loss for batch $\mathcal{I}_{t,b}$ is defined

as $L_{\mathcal{I}_{t,b}}(\mathbf{W}) = (1/B) \sum_{i \in \mathcal{I}_{t,b}} \ell(y_i f(\mathbf{W}, \mathbf{x}_i))$. Then the gradient descent update of the filters in the CNN can be written as

$$\mathbf{w}^{(t,b+1)} = \mathbf{w}^{(t,b)} - \eta \cdot \nabla_{\mathbf{W}} L_{\mathcal{I}_{t,b}}(\mathbf{W}^{(t,b)}), \tag{2}$$

where the gradient of the empirical loss $\nabla_{\mathbf{W}} L_{\mathcal{I}_{t,b}}$ is the collection of $\nabla_{\mathbf{w}_{j,r}} L_{\mathcal{I}_{t,b}}$ as follows

$$\nabla_{\mathbf{w}_{j,r}} L_{\mathcal{I}_{t,b}}(\mathbf{W}^{(t,b)}) = \frac{(P-1)}{Bm} \sum_{i \in \mathcal{I}_{t,b}} \ell_i'^{(t,b)} \cdot \sigma'(\langle \mathbf{w}_{j,r}^{(t,b)}, \boldsymbol{\xi}_i \rangle) \cdot j y_i \boldsymbol{\xi}_i$$

$$+ \frac{1}{Bm} \sum_{i \in \mathcal{I}_{t,b}} \ell_i'^{(t,b)} \cdot \sigma'(\langle \mathbf{w}_{j,r}^{(t,b)}, \widehat{y}_i \boldsymbol{\mu} \rangle) \cdot \widehat{y}_i y_i j \boldsymbol{\mu}, \tag{3}$$

for all $j \in \{\pm 1\}$ and $r \in [m]$. Here we introduce a shorthand notation $\ell_i'^{(t,b)} = \ell'[y_i \cdot f(\mathbf{W}^{(t,b)}, \mathbf{x}_i)]$ and assume the gradient of the ReLU activation function at 0 to be $\sigma'(0) = 1$ without loss of generality. We use $(t, b)$ to denote epoch index $t$ with mini-batch index $b$ and use $(t)$ as the shorthand of $(t, 0)$. We initialize SGD by random Gaussian, where all entries of $\mathbf{W}^{(0)}$ are sampled from i.i.d. Gaussian distributions $\mathcal{N}(0, \sigma_0^2)$, with $\sigma_0^2$ being the variance. From (3), we can infer that the loss landscape of the empirical loss is highly non-smooth because the ReLU function is not differentiable at zero. In particular, when $\langle \mathbf{w}_{j,r}^{(t,b)}, \boldsymbol{\xi} \rangle$ is close to zero, even a very small perturbation can greatly change the activation pattern $\sigma'(\langle \mathbf{w}_{j,r}^{(t,b)}, \boldsymbol{\xi} \rangle)$ and thus change the direction of $\nabla_{\mathbf{w}_{j,r}} L_{\mathcal{I}_{t,b}}(\mathbf{W}^{(t,b)})$. This observation prevents the analysis technique based on the Taylor expansion with the Hessian matrix, and calls for a more sophisticated activation pattern analysis.

**Sharpness Aware Minimization.** Given an empirical loss function $L_S(\mathbf{W})$ with trainable parameter $\mathbf{W}$, the idea of SAM is to minimize a perturbed empirical loss at the worst point in the neighborhood ball of $\mathbf{W}$ to ensure a uniformly low training loss value. In particular, it aims to solve the following optimization problem

$$\min_{\mathbf{W}} L_S^{\text{SAM}}(\mathbf{W}), \quad \text{where} \quad L_S^{\text{SAM}}(\mathbf{W}) := \max_{\|\boldsymbol{\epsilon}\|_2 \leq \tau} L_S(\mathbf{W} + \boldsymbol{\epsilon}), \tag{4}$$

where the hyperparameter $\tau$ is called the perturbation radius. However, directly optimizing $L_S^{\text{SAM}}(\mathbf{W})$ is computationally expensive. In practice, people use the following sharpness-aware minimization (SAM) algorithm (Foret et al., 2020; Zheng et al., 2021) to minimize $L_S^{\text{SAM}}(\mathbf{W})$ efficiently,

$$\mathbf{W}^{(t+1)} = \mathbf{W}^{(t)} - \eta \nabla_{\mathbf{W}} L_S(\mathbf{W} + \widehat{\boldsymbol{\epsilon}}), \quad \text{where} \quad \widehat{\boldsymbol{\epsilon}} = \tau \cdot \frac{\nabla_{\mathbf{W}} L_S(\mathbf{W})}{\|\nabla_{\mathbf{W}} L_S(\mathbf{W})\|_F}. \tag{5}$$

When applied to SGD in (2), the gradient $\nabla_{\mathbf{W}} L_S$ in (5) is further replaced by stochastic gradient $\nabla_{\mathbf{W}} L_{\mathcal{I}_{t,b}}$ (Foret et al., 2020). The detailed algorithm description of SAM in shown in Algorithm 1.

---

**Algorithm 1** Minibatch Sharpness Aware Minimization

---

**Input:** Training set $\mathcal{S} = \cup_{i=1}^n \{(\mathbf{x}_i, \mathbf{y}_i)\}$, Batch size $B$, step size $\eta > 0$, neighborhood size $\tau > 0$.

Initialize weights $\mathbf{W}^{(0)}$.
**for** $t = 0, 1, \ldots, T - 1$ **do**
    Randomly divide the training data set into $H$ mini batches $\{\mathcal{I}_{t,b}\}_{b=0}^{H-1}$.
    **for** $b = 0, 1, \ldots, H - 1$ **do**
        We calculate the perturbation $\widehat{\boldsymbol{\epsilon}}^{(t,b)} = \tau \frac{\nabla_{\mathbf{W}} L_{\mathcal{I}_{t,b}}(\mathbf{W}^{(t,b)})}{\|\nabla_{\mathbf{W}} L_{\mathcal{I}_{t,b}}(\mathbf{W}^{(t,b)})\|_F}$.
        Update model parameters: $\mathbf{W}^{(t,b+1)} = \mathbf{W}^{(t,b)} - \eta \nabla_{\mathbf{W}} L_{\mathcal{I}_{t,b}}(\mathbf{W})|_{\mathbf{W}=\mathbf{W}^{(t,b)}+\widehat{\boldsymbol{\epsilon}}^{(t,b)}}$.
    **end for**
    Update model parameters: $\mathbf{W}^{(t+1,0)} = \mathbf{W}^{(t,H)}$
**end for**

---

## 3 Result for SGD

In this section, we present our main theoretical results for the CNN trained with SGD. Our results are based on the following conditions on the dimension $d$, sample size $n$, neural network width $m$, initialization scale $\sigma_0$ and learning rate $\eta$.

**Condition 3.1.** Suppose there exists a sufficiently large constant $C$, such that the following hold:

1. Dimension $d$ is sufficiently large: $d \geq \widetilde{\Omega}\Big( \max\{nP^{-2}\sigma_p^{-2}\|\boldsymbol{\mu}\|_2^2, n^2, P^{-2}\sigma_p^{-2}Bm\}\Big)$.

2. Training sample size $n$ and neural network width satisfy $m, n \geq \widetilde{\Omega}(1)$.

3. The 2-norm of the signal satisfies $\|\boldsymbol{\mu}\|_2 \geq \widetilde{\Omega}(P\sigma_p)$.

4. The noise rate $p$ satisfies $p \leq 1/C$.

5. The standard deviation of Gaussian initialization $\sigma_0$ is appropriately chosen such that $\sigma_0 \leq \widetilde{O}\Big(\big(\max\big\{P\sigma_p d/\sqrt{n}, \|\boldsymbol{\mu}\|_2\big\}\big)^{-1}\Big)$.

6. The learning rate $\eta$ satisfies $\eta \leq \widetilde{O}\Big(\big(\max\big\{P^2\sigma_p^2 d^{3/2}/(Bm), P^2\sigma_p^2 d/B, n\|\boldsymbol{\mu}\|_2/(\sigma_0 B\sqrt{d}m),$
$n P\sigma_p\|\boldsymbol{\mu}\|_2/(B^2 m\epsilon)\big\}\big)^{-1}\Big)$.

The conditions imposed on the data dimensions $d$, network width $m$, and the number of samples $n$ ensure adequate overparameterization of the network. Additionally, the condition on the learning rate $\eta$ facilitates efficient learning by our model. By concentration inequality, with high probability, the $\ell_2$ norm of the noise patch is of order $\Theta(d\sigma_p^2)$. Therefore, the quantity $d\sigma_p^2$ can be viewed as the strength of the noise. Comparable conditions have been established in Chatterji and Long (2021); Cao et al. (2022); Frei et al. (2022); Kou et al. (2023). Based on the above condition, we first present a set of results on benign/harmful overfitting for SGD in the following theorem.

**Theorem 3.2** (Benign/harmful overfitting of SGD in training CNNs). *For any $\epsilon > 0$, under Condition 3.1, with probability at least $1 - \delta$ there exists $t = \widetilde{O}(\eta^{-1}\epsilon^{-1}mnd^{-1}P^{-2}\sigma_p^{-2})$ such that:*

1. *The training loss converges, i.e., $L_S(\mathbf{W}^{(t)}) \leq \epsilon$.*

2. *When $n\|\boldsymbol{\mu}\|_2^4 \geq C_1 dP^4\sigma_p^4$, the test error $L_{\mathcal{D}}^{0-1}(\mathbf{W}^{(t)}) \leq p + \epsilon$.*

3. *When $n\|\boldsymbol{\mu}\|_2^4 \leq C_3 dP^4\sigma_p^4$, the test error $L_{\mathcal{D}}^{0-1}(\mathbf{W}^{(t)}) \geq p + 0.1$.*

Theorem 3.2 reveals a sharp phase transition between benign and harmful overfitting for CNN trained with SGD. This transition is determined by the relative scale of the signal strength and the data dimension. Specifically, if the signal is relatively large such that $n\|\boldsymbol{\mu}\|_2^4 \geq C_1 d(P-1)^4\sigma_p^4$, the model can efficiently learn the signal. As a result, the test error decreases, approaching the Bayes risk $p$, although the presence of label flipping noise prevents the test error from reaching zero. Conversely, when the condition $n\|\boldsymbol{\mu}\|_2^4 \leq C_3 d(P-1)^4\sigma_p^4$ holds, the test error fails to approach the Bayes risk. This phase transition is empirically illustrated in Figure 2. In both scenarios, the model is capable of fitting the training data thoroughly, even for examples with flipped labels. This finding aligns with longstanding empirical observations.

The negative result of SGD, which encompasses the third point of Theorem 3.2 and the high test error observed in Figure 2, suggests that the signal strength needs to scale with the data dimension to enable benign overfitting. This constraint substantially undermines the efficiency of SGD, particularly when dealing with high-dimensional data. A significant part of this limitation stems from the fact that SGD does not inhibit the model from learning noise, leading to a comparable rate of signal and noise learning during iterative model parameter updates. This inherent limitation of SGD is effectively addressed by SAM, as we will discuss later in Section 4.

### 3.1 Analysis of Mini-Batch SGD

In contrast to GD, SGD does not utilize all the training data at each iteration. Consequently, different samples may contribute to parameters differently, leading to possible unbalancing in parameters. To analyze SGD, we extend the signal-noise decomposition technique developed by Kou et al. (2023); Cao et al. (2022) for GD, which in our case is formally defined as:

**Lemma 3.3.** *Let* $\mathbf{w}_{j,r}^{(t,b)}$ *for* $j \in \{\pm 1\}$, $r \in [m]$ *be the convolution filters of the CNN at the b-th batch of t-th epoch of gradient descent. Then there exist unique coefficients* $\gamma_{j,r}^{(t,b)}$ *and* $\rho_{j,r,i}^{(t,b)}$ *such that*

$$\mathbf{w}_{j,r}^{(t,b)} = \mathbf{w}_{j,r}^{(0,0)} + j \cdot \gamma_{j,r}^{(t,b)} \cdot \|\boldsymbol{\mu}\|_2^{-2} \cdot \boldsymbol{\mu} + \frac{1}{P-1} \sum_{i=1}^{n} \rho_{j,r,i}^{(t,b)} \cdot \|\boldsymbol{\xi}_i\|_2^{-2} \cdot \boldsymbol{\xi}_i. \tag{6}$$

*Further denote* $\overline{\rho}_{j,r,i}^{(t,b)} := \rho_{j,r,i}^{(t,b)} \mathbb{1}(\rho_{j,r,i}^{(t,b)} \geq 0)$, $\underline{\rho}_{j,r,i}^{(t,b)} := \rho_{j,r,i}^{(t,b)} \mathbb{1}(\rho_{j,r,i}^{(t,b)} \leq 0)$. *Then*

$$\mathbf{w}_{j,r}^{(t,b)} = \mathbf{w}_{j,r}^{(0,0)} + j\gamma_{j,r}^{(t,b)}\|\boldsymbol{\mu}\|_2^{-2}\boldsymbol{\mu} + \frac{1}{P-1} \sum_{i=1}^{n} \overline{\rho}_{j,r,i}^{(t,b)}\|\boldsymbol{\xi}_i\|_2^{-2}\boldsymbol{\xi}_i + \frac{1}{P-1} \sum_{i=1}^{n} \underline{\rho}_{j,r,i}^{(t,b)}\|\boldsymbol{\xi}_i\|_2^{-2}\boldsymbol{\xi}_i. \tag{7}$$

Note that (7) is a variant of (6): by decomposing the coefficient $\rho_{j,r,i}^{(t,b)}$ into $\overline{\rho}_{j,r,i}^{(t,b)}$ and $\underline{\rho}_{j,r,i}^{(t,b)}$, we can streamline our proof process. The normalization terms $\frac{1}{P-1}$, $\|\boldsymbol{\mu}\|_2^{-2}$, and $\|\boldsymbol{\xi}_i\|_2^{-2}$ ensure that $\gamma_{j,r}^{(t,b)} \approx \langle \mathbf{w}_{j,r}^{(t,b)}, \boldsymbol{\mu}\rangle$ and $\rho_{j,r}^{(t,b)} \approx (P-1)\langle \mathbf{w}_{j,r}^{(t,b)}, \boldsymbol{\xi}_i\rangle$. Through signal-noise decomposition, we characterize the learning progress of signal $\boldsymbol{\mu}$ using $\gamma_{j,r}^{(t,b)}$, and the learning progress of noise using $\rho_{j,r}^{(t,b)}$. This decomposition turns the analysis of SGD updates into the analysis of signal noise coefficients. Kou et al. (2023) extend this technique to the ReLU activation function as well as in the presence of label flipping noise. However, mini-batch SGD updates amplify the complications introduced by label flipping noise, making it more difficult to ensure learning. We have developed advanced methods for coefficient balancing and activation pattern analysis. These techniques will be thoroughly discussed in the sequel. The progress of signal learning is characterized by $\gamma_{j,r}^{(t,b)}$, whose update rule is as follows:

$$\gamma_{j,r}^{(t,b+1)} = \gamma_{j,r}^{(t,b)} - \frac{\eta}{Bm} \cdot \left[ \sum_{i \in \mathcal{I}_{t,b} \cap S_+} \ell_i'^{(t,b)} \sigma'(\langle \mathbf{w}_{j,r}^{(t,b)}, \widehat{y}_i \cdot \boldsymbol{\mu}\rangle) \right.$$
$$\left. - \sum_{i \in \mathcal{I}_{t,b} \cap S_-} \ell_i'^{(t,b)} \sigma'(\langle \mathbf{w}_{j,r}^{(t,b)}, \widehat{y}_i \cdot \boldsymbol{\mu}\rangle) \right] \cdot \|\boldsymbol{\mu}\|_2^2. \tag{8}$$

Here, $\mathcal{I}_{t,b}$ represents the indices of samples in batch $b$ of epoch $t$, $S_+$ denotes the set of clean samples where $y_i = \widehat{y}_i$, and $S_-$ represents the set of noisy samples where $y_i = -\widehat{y}_i$. The updates of $\gamma_{j,r}^{(t,b)}$ comprise an increment arising from sample learning, counterbalanced by a decrement due to noisy sample learning. Both empirical and theoretical analyses have demonstrated that overparametrization allows the model to fit even random labels. This occurs when the negative term $\sum_{i \in \mathcal{I}_{t,b} \cap S_-} \ell_i'^{(t,b)} \sigma'(\langle \mathbf{w}_{j,r}^{(t,b)}, y_i \cdot \boldsymbol{\mu}\rangle)$ primarily drives model learning. Such unfavorable scenarios can be attributed to two possible factors. Firstly, the gradient of the loss $\ell_i'^{(t,b)}$ might be significantly larger for noisy samples compared to clean samples. Secondly, during certain epochs, the majority of samples may be noisy, meaning that $\mathcal{I}_{t,b} \cap S_-$ significantly outnumbers $\mathcal{I}_{t,b} \cap S_+$.

To deal with the first factor, we have to control the ratio of the loss gradient with regard to different samples, as depicted in (9). Given that noisy samples may overwhelm a single batch, we impose an additional requirement: the ratio of the loss gradient must be controllable across different batches within a single epoch, i.e.,

$$\ell_i'^{(t,b_1)}/\ell_k'^{(t,b_2)} \leq C_2. \tag{9}$$

As $\ell'(z_1)/\ell'(z_2) \approx \exp(z_2 - z_1)$, we can upper bound $\ell_i'^{(t,b_1)}/\ell_k'^{(t,b_2)}$ by $y_i \cdot f(\mathbf{W}^{(t,b_1)}, \mathbf{x}_i) - y_k \cdot f(\mathbf{W}^{(t,b_2)}, \mathbf{x}_k)$, which can be further upper bounded by $\sum_r \overline{\rho}_{y_i,r,i}^{(t,b_1)} - \sum_r \overline{\rho}_{y_i,r,k}^{(t,b_2)}$ with a small error. Therefore, the proof of (9) can be reduced to proving a uniform bound on the difference among $\overline{\rho}_{y_i,r,i}^{(t,b)}$, i.e., $\sum_{r=1}^m \overline{\rho}_{y_i,r,i}^{(t,b_1)} - \sum_{r=1}^m \overline{\rho}_{y_k,r,k}^{(t,b_2)} \leq \kappa$, $\forall i, k$.

However, achieving this uniform upper bound turns out to be challenging, since the updates of $\overline{\rho}_{j,r,i}^{(t,b)}$'s are not evenly distributed across different batches within an epoch. Each mini-batch update utilizes only a portion of the samples, meaning that some $\overline{\rho}_{y_i,r,i}^{(t,b)}$ can increase or decrease much more than

the others. Therefore, the uniformly bounded difference can only be achieved after the entire epoch is processed. Consequently, we have to first bound the difference among $\overline{\rho}_{y_i,r,i}^{(t,b)}$'s after each entire epoch, and then control the maximal difference within one epoch. The full batch (epoch) update rule is established as follows:

$$
\sum_{r=1}^{m} \left[ \overline{\rho}_{y_i,r,i}^{(t+1,0)} - \overline{\rho}_{y_k,r,k}^{(t+1,0)} \right] = \sum_{r=1}^{m} \left[ \overline{\rho}_{y_i,r,i}^{(t,0)} - \overline{\rho}_{y_k,r,k}^{(t,0)} \right] - \frac{\eta(P-1)^2}{Bm} \cdot \left( |\widetilde{S}_i^{(t,b_i^{(t)})}| \ell_i'^{(t,b_i^{(t)})} \cdot \|\boldsymbol{\xi}_i\|_2^2 \right.
$$
$$
\left. - |\widetilde{S}_k^{(t,b_k^{(t)})}| \ell_k'^{(t,b_k^{(t)})} \cdot \|\boldsymbol{\xi}_k\|_2^2 \right). \tag{10}
$$

Here, $b_i^{(t)}$ denotes the batch to which sample $i$ belongs in epoch $t$, and $\widetilde{S}_i^{(t,b_i^{(t)})}$ represents the parameters that learn $\boldsymbol{\xi}_i$ at epoch $t$ defined as

$$
\widetilde{S}_i^{(t,b)} := \{ r : \langle \mathbf{w}_{y_i,r}^{(t,b)}, \boldsymbol{\xi}_i \rangle > 0 \}. \tag{11}
$$

Therefore, the update of $\sum_{r=1}^{m} \left[ \overline{\rho}_{y_i,r,i}^{(t,0)} - \overline{\rho}_{y_k,r,k}^{(t,0)} \right]$ is indeed characterized by the activation pattern of parameters, which serves as the key technique for analyzing the full batch update of $\sum_{r=1}^{m} \left[ \overline{\rho}_{y_i,r,i}^{(t,0)} - \overline{\rho}_{y_k,r,k}^{(t,0)} \right]$. However, analyzing the pattern of $S_i^{(t,b)}$ directly is challenging since $\langle \mathbf{w}_{y_i,r}^{(t,b)}, \boldsymbol{\xi}_i \rangle$ fluctuates in mini-batches without sample $i$. Therefore, we introduce the set series $S_i^{(t,b)}$ as the activation pattern with certain threshold as follows:

$$
S_i^{(t,b)} := \{ r : \langle \mathbf{w}_{y_i,r}^{(t,b)}, \boldsymbol{\xi}_i \rangle > \sigma_0 \sigma_p \sqrt{d}/\sqrt{2} \}. \tag{12}
$$

The following lemma suggests that the set of activated parameters $S_i^{(t,0)}$ is a non-decreasing sequence with regards to $t$, and the set of plain activated parameters $\widetilde{S}_i^{(t,b)}$ always include $S_i^{(t,0)}$. Consequently, $S_i^{(0,0)}$ is always included in $\widetilde{S}_i^{(t,b)}$, guaranteeing that $\boldsymbol{\xi}_i$ can always be learned by some parameter. And this further makes sure the difference among $\overline{\rho}_{y_i,r,i}^{(t,b)}$ is bounded, as well as $\ell_i'^{(t,b_1)}/\ell_k'^{(t,b_2)} \leq C_2$. In the proof for SGD, we consider the learning period $0 \leq t \leq T^*$, where $T^* = \eta^{-1}\mathrm{poly}(\epsilon^{-1}, d, n, m)$ is the maximum number of admissible iterations.

**Lemma 3.4** (Informal Statement of Lemma C.8). *For all $t \in [0, T^*]$ and $b \in \overline{[H]}$, we have*

$$
S_i^{(t-1,0)} \subseteq S_i^{(t,0)} \subseteq \widetilde{S}_i^{(t,b)}. \tag{13}
$$

As we have mentioned above, if noisy samples outnumber clean samples, $\gamma_{j,r}^{(t,b)}$ may also decrease. To deal with such a scenario, we establish a two-stage analysis of $\gamma_{j,r}^{(t,b)}$ progress. In the first stage, when $-\ell_i'$ is lower bound by a positive constant, we prove that there are enough batches containing sufficient clear samples. This is characterized by the following high-probability event.

**Lemma 3.5** (Infomal Statement of Lemma B.6). *With high probability, for all $T \in [\widetilde{O}(1), T^*]$, there exist at least $c_1 \cdot T$ epochs among $[0, T]$, such that at least $c_2 \cdot H$ batches in each of these epochs satisfying the following condition:*

$$
|S_+ \cap S_y \cap \mathcal{I}_{t,b}| \in [0.25B, 0.75B]. \tag{14}
$$

After the first stage of $T = \Theta(\eta^{-1}m(P-1)^{-2}\sigma_p^{-2}d^{-1})$ epochs, we have $\gamma_{j,r}^{(T,0)} = \Omega\left( n \frac{\|\boldsymbol{\mu}\|_2^2}{(P-1)^2\sigma_p^2 d} \right)$. The scale of $\gamma_{j,r}^{(T,0)}$ guarantees that $\langle \mathbf{w}_{j,r}^{(t,b)}, \boldsymbol{\mu} \rangle$ remains resistant to intra-epoch fluctuations. Consequently, this implies the sign of $\langle \mathbf{w}_{j,r}^{(t,b)}, \boldsymbol{\mu} \rangle$ will persist unchanged throughout the entire epoch. Without loss of generality, we suppose that $\langle \mathbf{w}_{j,r}^{(t,b)}, \boldsymbol{\mu} \rangle > 0$, then the update of $\gamma_{j,r}^{(t,b)}$ can be written as follows:

$$
\gamma_{j,r}^{(t+1,0)} = \gamma_{j,r}^{(t,0)} + \frac{\eta}{Bm} \cdot \left[ \min_{i \in \mathcal{I}_{t,b,b}} |\ell_i'^{(t,b)}| |S_+ \cap S_1| - \max_{i \in \mathcal{I}_{t,b,b}} |\ell_i'^{(t,b)}| |S_- \cap S_{-1}| \right] \cdot \|\boldsymbol{\mu}\|_2^2. \tag{15}
$$

As we have proved the balancing of logits $\ell_i'^{(t,b)}$ across batches, the progress analysis of $\gamma_{j,r}^{(t+1,0)}$ is established to characterize the signal learning of SGD.

# 4 Result for SAM

In this section, we present the positive results for SAM in the following theorem.

**Theorem 4.1.** *For any $\epsilon > 0$, under Condition 3.1 with $\sigma_0 = \widetilde{\Theta}(P^{-1}\sigma_p^{-1}d^{-1/2})$, choose $\tau = \Theta\left(\frac{m\sqrt{B}}{P\sigma_p\sqrt{d}}\right)$. With probability at least $1 - \delta$, neural networks first trained with SAM with $O\left(\eta^{-1}\epsilon^{-1}n^{-1}mB\|\boldsymbol{\mu}\|_2^{-2}\right)$ iterations, then trained with SGD with $\widetilde{O}\left(\eta^{-1}\epsilon^{-1}mnd^{-1}P^{-2}\sigma_p^{-2}\right)$ iterations can find $\mathbf{W}^{(t)}$ such that,*

1. *The training loss satisfies $L_S(\mathbf{W}^{(t)}) \leq \epsilon$.*

2. *The test error $L_{\mathcal{D}}^{0-1}(\mathbf{W}^{(t)}) \leq p + \epsilon$.*

In contrast to Theorem 3.2, Theorem 4.1 demonstrates that CNNs trained by SAM exhibit benign overfitting under much milder conditions. This condition is almost dimension-free, as opposed to the threshold of $\|\boldsymbol{\mu}\|_2^4 \geq \widetilde{\Omega}((d/n)P^4\sigma_p^4)$ for CNNs trained by SGD. The discrepancy in the thresholds can be observed in Figure 1. This difference is because SAM introduces a perturbation during the model parameter update process, which effectively prevents the early-stage memorization of noise by deactivating the corresponding neurons.

## 4.1 Noise Memorization Prevention

In this subsection, we will show how SAM can prevent noise memorization by changing the activation pattern of the neurons. For SAM, we have the following update rule of decomposition coefficients $\gamma_{j,r}^{(t,b)}, \overline{\rho}_{j,r,i}^{(t,b)}, \underline{\rho}_{j,r,i}^{(t,b)}$.

**Lemma 4.2.** *The coefficients $\gamma_{j,r}^{(t,b)}, \overline{\rho}_{j,r,i}^{(t,b)}, \underline{\rho}_{j,r,i}^{(t,b)}$ defined in Lemma 3.3 satisfy the following iterative equations for all $r \in [m]$, $j \in \{\pm 1\}$ and $i \in [n]$:*

$$\gamma_{j,r}^{(0,0)}, \overline{\rho}_{j,r,i}^{(0,0)}, \underline{\rho}_{j,r,i}^{(0,0)} = 0,$$

$$\gamma_{j,r}^{(t,b+1)} = \gamma_{j,r}^{(t,b)} - \frac{\eta}{Bm} \cdot \left[ \sum_{i \in \mathcal{I}_{t,b} \cap S_+} \ell_i'^{(t,b)} \sigma'(\langle \mathbf{w}_{j,r}^{(t,b)} + \widehat{\boldsymbol{\epsilon}}_{j,r}^{(t,b)}, \widehat{y}_i \cdot \boldsymbol{\mu} \rangle) \right.$$
$$\left. - \sum_{i \in \mathcal{I}_{t,b} \cap S_-} \ell_i'^{(t,b)} \sigma'(\langle \mathbf{w}_{j,r}^{(t,b)} + \widehat{\boldsymbol{\epsilon}}_{j,r}^{(t,b)}, \widehat{y}_i \cdot \boldsymbol{\mu} \rangle) \right] \cdot \|\boldsymbol{\mu}\|_2^2,$$

$$\overline{\rho}_{j,r,i}^{(t,b+1)} = \overline{\rho}_{j,r,i}^{(t,b)} - \frac{\eta(P-1)^2}{Bm} \cdot \ell_i'^{(t,b)} \cdot \sigma'(\langle \mathbf{w}_{j,r}^{(t,b)} + \widehat{\boldsymbol{\epsilon}}_{j,r}^{(t,b)}, \boldsymbol{\xi}_i \rangle) \cdot \|\boldsymbol{\xi}_i\|_2^2 \cdot \mathbb{1}(y_i = j)\,\mathbb{1}(i \in \mathcal{I}_{t,b}),$$

$$\underline{\rho}_{j,r,i}^{(t,b+1)} = \underline{\rho}_{j,r,i}^{(t,b)} + \frac{\eta(P-1)^2}{Bm} \cdot \ell_i'^{(t,b)} \cdot \sigma'(\langle \mathbf{w}_{j,r}^{(t,b)} + \widehat{\boldsymbol{\epsilon}}_{j,r}^{(t,b)}, \boldsymbol{\xi}_i \rangle) \cdot \|\boldsymbol{\xi}_i\|_2^2 \cdot \mathbb{1}(y_i = -j)\,\mathbb{1}(i \in \mathcal{I}_{t,b}),$$

*where $\mathcal{I}_{t,b}$ denotes the sample index set of the $b$-th batch in the $t$-th epoch.*

The primary distinction between SGD and SAM is how neuron activation is determined. In SAM, the activation is based on the perturbed weight $\mathbf{w}_{j,r}^{(t,b)} + \widehat{\boldsymbol{\epsilon}}_{j,r}^{(t,b)}$, whereas in SGD, it is determined by the unperturbed weight $\mathbf{w}_{j,r}^{(t,b)}$. This perturbation to the weight update process at each iteration gives SAM an intriguing denoising property. Specifically, if a neuron is activated by noise in the SGD update, it will subsequently become deactivated after the perturbation, as stated in the following lemma.

**Lemma 4.3** (Informal Statement of Lemma D.5). *Suppose the Condition 3.1 holds with parameter choices in Theorem 4.1, if $\langle \mathbf{w}_{j,r}^{(t,b)}, \boldsymbol{\xi}_k \rangle \geq 0$, $k \in \mathcal{I}_{t,b}$ and $j = y_k$, then $\langle \mathbf{w}_{j,r}^{(t,b)} + \widehat{\boldsymbol{\epsilon}}_{j,r}^{(t,b)}, \boldsymbol{\xi}_k \rangle < 0$.*

By leveraging this intriguing property, we can derive a constant upper bound for the noise coefficients $\overline{\rho}_{j,r,i}^{(t,b)}$ by considering the following cases:

1. If $\boldsymbol{\xi}_i$ is not in the current batch, then $\overline{\rho}_{j,r,i}^{(t,b)}$ will not be updated in the current iteration.

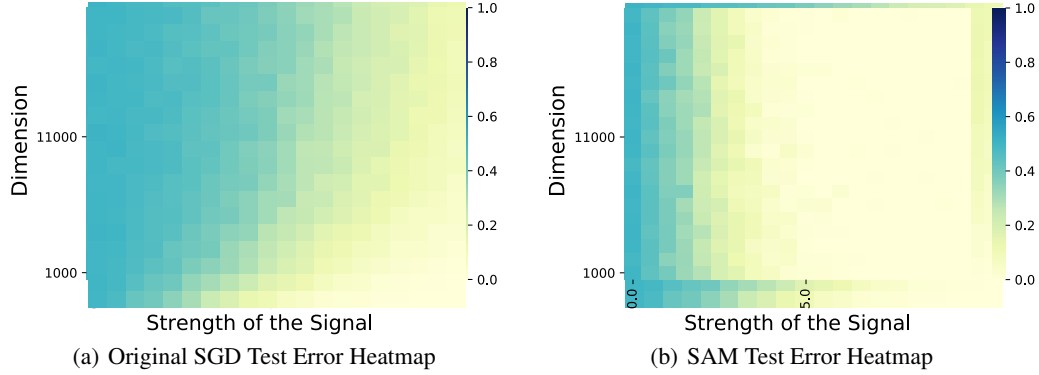

| (a) Original SGD Test Error Heatmap | (b) SAM Test Error Heatmap |

Figure 2: (a) is a heatmap illustrating test error on synthetic data for various dimensions $d$ and signal strengths $\boldsymbol{\mu}$ when trained using Vanilla Gradient Descent. High test errors are represented in blue, while low test errors are shown in yellow. (b) displays a heatmap of test errors on the synthetic data under the same conditions as in (a), but trained using SAM instead with $\tau = 0.03$. The y-axis represents a normal scale with a range of $1000 \sim 21000$.

2. If $\boldsymbol{\xi}_i$ is in the current batch, we discuss two cases:

   (a) If $\langle \mathbf{w}_{j,r}^{(t,b)}, \boldsymbol{\xi}_i \rangle \geq 0$, then by Lemma 4.3, one can know that $\sigma'(\langle \mathbf{w}_{j,r}^{(t,b)} + \widehat{\boldsymbol{\epsilon}}_{j,r}^{(t,b)}, \boldsymbol{\xi}_i \rangle) = 0$ and thus $\overline{\rho}_{j,r,i}^{(t,b)}$ will not be updated in the current iteration.

   (b) If $\langle \mathbf{w}_{j,r}^{(t,b)}, \boldsymbol{\xi}_i \rangle \leq 0$, then given that $\langle \mathbf{w}_{j,r}^{(t,b)}, \boldsymbol{\xi}_i \rangle \approx \overline{\rho}_{j,r,i}^{(t,b)}$ and $\overline{\rho}_{j,r,i}^{(t,b+1)} \leq \overline{\rho}_{j,r,i}^{(t,b)} + \frac{\eta(P-1)^2 \|\boldsymbol{\xi}_i\|_2^2}{Bm}$, we can assert that, provided $\eta$ is sufficiently small, the term $\overline{\rho}_{j,r,i}^{(t,b)}$ can be upper bounded by a small constant.

In contrast to the analysis of SGD, which provides an upper bound for $\overline{\rho}_{j,r,i}^{(t,b)}$ of order $O(\log d)$, the noise memorization prevention property described in Lemma 4.3 allows us to obtain an upper bound for $\overline{\rho}_{j,r,i}^{(t,b)}$ of order $O(1)$ throughout $[0, T_1]$. This indicates that SAM memorizes less noise compared to SGD. On the other hand, the signal coefficient $\gamma_{j,r,i}^{(t)}$ also increases to $\Omega(1)$ for SAM, following the same argument as in SGD. This property ensures that training with SAM does not exhibit harmful overfitting for the same signal-to-noise ratio at which training with SGD suffers from harmful overfitting.

## 5  Experiments

In this section, we conduct synthetic experiments to validate our theory. Additional experiments on real data sets can be found in Appendix A.

We set training data size $n = 20$ without label-flipping noise. Since the learning problem is rotation-invariant, without loss of generality, we set $\boldsymbol{\mu} = \|\boldsymbol{\mu}\|_2 \cdot [1, 0, \ldots, 0]^\top$. We then generate the noise vector $\boldsymbol{\xi}$ from the Gaussian distribution $\mathcal{N}(\mathbf{0}, \sigma_p^2 \mathbf{I})$ with fixed standard deviation $\sigma_p = 1$. We train a two-layer CNN model defined in Section 2 with the ReLU activation function. The number of filters is set as $m = 10$. We use the default initialization method in PyTorch to initialize the CNN parameters and train the CNN with full-batch gradient descent with a learning rate of $0.01$ for $100$ iterations. We consider different dimensions $d$ ranging from $1000$ to $20000$, and different signal strengths $\|\boldsymbol{\mu}\|_2$ ranging from $0$ to $10$. Based on our results, for any dimension $d$ and signal strength $\mu$ setting we consider, our training setup can guarantee a training loss smaller than $0.05$. After training, we estimate the test error for each case using $1000$ test data points. We report the test error heat map with average results over $10$ runs in Figure 2.

## 6  Related Work

**Sharpness Aware Minimization.** Foret et al. (2020), and Zheng et al. (2021) concurrently introduced methods to enhance generalization by minimizing the loss in the worst direction, perturbed from the current parameter. Kwon et al. (2021) introduced ASAM, a variant of SAM, designed to address parameter re-scaling. Subsequently, Liu et al. (2022b) presented LookSAM, a more computationally

efficient alternative. Zhuang et al. (2022) highlighted that SAM did not consistently favor the flat minima and proposed GSAM to improve generalization by minimizing the surrogate. Recently, Zhao et al. (2022) showed that the SAM algorithm is related to the gradient regularization (GR) method when the loss is smooth and proposed an algorithm that can be viewed as a generalization of the SAM algorithm. Meng et al. (2023) further studied the mechanism of Per-Example Gradient Regularization (PEGR) on the CNN training and revealed that PEGR penalizes the variance of pattern learning.

**Benign Overfitting in Neural Networks.** Since the pioneering work by Bartlett et al. (2020) on benign overfitting in linear regression, there has been a surge of research studying benign overfitting in linear models, kernel methods, and neural networks. Li et al. (2021b); Montanari and Zhong (2022) examined benign overfitting in random feature or neural tangent kernel models defined in two-layer neural networks. Chatterji and Long (2022) studied the excess risk of interpolating deep linear networks trained by gradient flow. Understanding benign overfitting in neural networks beyond the linear/kernel regime is much more challenging because of the non-convexity of the problem. Recently, Frei et al. (2022) studied benign overfitting in fully-connected two-layer neural networks with smoothed leaky ReLU activation. Cao et al. (2022) provided an analysis for learning two-layer convolutional neural networks (CNNs) with polynomial ReLU activation function (ReLU$^q$, $q > 2$). Kou et al. (2023) further investigates the phenomenon of benign overfitting in learning two-layer ReLU CNNs. Kou et al. (2023) is most related to our paper. However, our work studied SGD rather than GD, which requires advanced techniques to control the update of coefficients at both batch-level and epoch-level. We also provide a novel analysis for SAM, which differs from the analysis of GD/SGD.

# 7 Conclusion

In this work, we rigorously analyze the training behavior of two-layer convolutional ReLU networks for both SGD and SAM. In particular, we precisely outlined the conditions under which benign overfitting can occur during SGD training, marking the first such finding for neural networks trained with mini-batch SGD. We also proved that SAM can lead to benign overfitting under circumstances that prompt harmful overfitting via SGD, which demonstrates the clear theoretical superiority of SAM over SGD. Our results provide a deeper comprehension of SAM, particularly when it comes to its utilization with non-smooth neural networks. An interesting future work is to consider other modern deep learning techniques, such as weight normalization, momentum, and weight decay, in our analysis.

## Acknowledgements

We thank the anonymous reviewers for their helpful comments. ZC, JZ, YK, and QG are supported in part by the National Science Foundation CAREER Award 1906169 and IIS-2008981, and the Sloan Research Fellowship. XC and CJH are supported in part by NSF under IIS-2008173, IIS-2048280, CISCO and Sony. The views and conclusions contained in this paper are those of the authors and should not be interpreted as representing any funding agencies.

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

Table 1: Top-1 accuracy (%) of ResNet-50 on the ImageNet dataset when we vary the starting point of using the SAM update rule, baseline result is 76.4%.

| $\tau$ | 10% | 30% | 50% | 70% | 90% |
|--------|------|------|------|------|------|
| 0.01 | 76.9 | 76.9 | 76.9 | 76.7 | 76.7 |
| 0.02 | 77.1 | 77.0 | 76.9 | 76.8 | 76.6 |
| 0.05 | 76.2 | 76.4 | 76.3 | 76.3 | 76.2 |

Table 2: Top-1 accuracy (%) of wide ResNet on the CIFAR datasets when adding different levels of Gaussian noise.

| Model | Noise | Dataset | Optimizer | Accuracy |
|-------|-------|---------|-----------|----------|
| WRN-16-8 | - | CIFAR-10 | SGD | 96.69 |
| WRN-16-8 | - | CIFAR-10 | SAM | 97.19 |
| WRN-16-8 | $\mathcal{N}(0, 0.1)$ | CIFAR-10 | SGD | 95.87 |
| WRN-16-8 | $\mathcal{N}(0, 0.1)$ | CIFAR-10 | SAM | 96.57 |
| WRN-16-8 | $\mathcal{N}(0, 0.3)$ | CIFAR-10 | SGD | 92.40 |
| WRN-16-8 | $\mathcal{N}(0, 0.3)$ | CIFAR-10 | SAM | 93.37 |
| WRN-16-8 | $\mathcal{N}(0, 1)$ | CIFAR-10 | SGD | 79.50 |
| WRN-16-8 | $\mathcal{N}(0, 1)$ | CIFAR-10 | SAM | 80.37 |
| WRN-16-8 | - | CIFAR-100 | SGD | 81.93 |
| WRN-16-8 | - | CIFAR-100 | SAM | 83.68 |

# A  Additional Experiments

## A.1  Real Experiments on CIFAR

In this section, we provide the experiments on real data sets.

**Varying different starting points for SAM.** In Section 4, we show that the SAM algorithm can effectively prevent noise memorization and thus improve feature learning in the early stage of training. Is SAM also effective if we add the algorithm in the middle of the training process? We conduct experiments on the ImageNet dataset with ResNet50. We choose the batch size as $1024$, and the model is trained for 90 epochs with the best learning rate in grid search $\{0.01, 0.03, 0.1, 0.3\}$. The learning rate schedule is 10k steps linear warmup, then cosine decay. As shown in Table 1, the earlier SAM is introduced, the more pronounced its effectiveness becomes.

**SAM with additive noises.** Here, we conduct experiments on the CIFAR dataset with WRN-16-8. We add Gaussian random noises to the image data with variance $\{0.1, 0.3, 1\}$. We choose the batch size as $128$ and train the model over 200 epochs using a learning rate of $0.1$, a momentum of $0.9$, and a weight decay of $5e - 4$. The SAM hyperparameter is chosen as $\tau = 2.0$. As we can see from Table 2, SAM can consistently prevent noise learning and get better performance, compared to the SGD, which varies from different additive noise levels.

## A.2  Discussion on the Stochastic Gradient Descent

Here, we include additional empirical results to show how the batch size and step size influence the transition phase of the benign/harmful regimes.

Our synthetic experiment is only performed with gradient descent instead of SGD in Figure 2. We add an experiment of SGD with a mini-batch size of 10 on the synthetic data. If we compare Figure 3 and Figure 2, the comparison results should remain the same, and both support our main Theorems 3.2 and 4.1.

In Figure 4, we also conducted an extended study on the learning rate to check whether tuning the learning rate can get better generalization performance and achieve SAM's performance. Specifically, we experimented with learning rates of $0.001, 0.01, 0.1$, and $1$ under the same conditions described

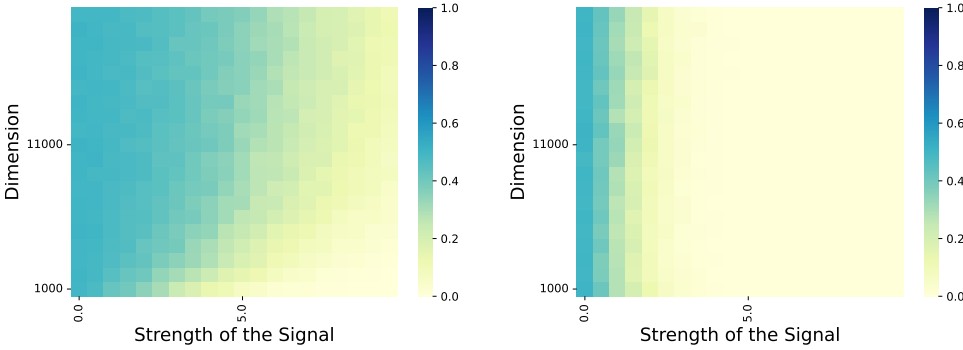

(a) SGD Test Error Heatmap with minibatch 10      (b) SAM Test Error Heatmap with minibatch 10

Figure 3: (a) is a heatmap illustrating test error on synthetic data for various dimensions $d$ and signal strengths $\boldsymbol{\mu}$ when trained using stochastic gradient descent. High test errors are represented in blue, while low test errors are shown in yellow. (b) displays a heatmap of test errors on the synthetic data under the same conditions as in (a), but trained using SAM instead. The y-axis represents a normal scale with a range of 1000-21000.

in Section 5. The results show that for all learning rates, the harmful and benign overfitting patterns are quite similar and consistent with our Theorem 3.2.

We observed that larger learning rates, such as $0.1$ and $1$, can improve SGD's generalization performance. The benign overfitting region is enlarged for learning rates of $0.1$ and $1$ when contrasted with $0.01$. This trend resonates with recent findings that large LR helps SGD find better minima (Li et al., 2019, 2021a; Ahn et al., 2022; Andriushchenko et al., 2023). Importantly, even with this expansion, the benign overfitting region remains smaller than what is empirically observed with SAM. Our conclusion is that while SGD, with a more considerable learning rate, exhibits improved generalization, it still falls short of matching SAM's performance.

## B    Preliminary Lemmas

**Lemma B.1** (Lemma B.4 in Kou et al. (2023)). *Suppose that $\delta > 0$ and $d = \Omega(\log(6n/\delta))$. Then with probability at least $1 - \delta$,*

$$\sigma_p^2 d/2 \leq \|\boldsymbol{\xi}_i\|_2^2 \leq 3\sigma_p^2 d/2,$$
$$|\langle \boldsymbol{\xi}_i, \boldsymbol{\xi}_{i'} \rangle| \leq 2\sigma_p^2 \cdot \sqrt{d \log(6n^2/\delta)},$$
$$|\langle \boldsymbol{\xi}_i, \boldsymbol{\mu} \rangle| \leq \|\boldsymbol{\mu}\|_2 \sigma_p \cdot \sqrt{2 \log(6n/\delta)}$$

*for all $i, i' \in [n]$.*

**Lemma B.2** (Lemma B.5 in Kou et al. (2023)). *Suppose that $d = \Omega(\log(mn/\delta))$, $m = \Omega(\log(1/\delta))$. Then with probability at least $1 - \delta$,*

$$\sigma_0^2 d/2 \leq \|\mathbf{w}_{j,r}^{(0,0)}\|_2^2 \leq 3\sigma_0^2 d/2,$$
$$|\langle \mathbf{w}_{j,r}^{(0,0)}, \boldsymbol{\mu} \rangle| \leq \sqrt{2 \log(12m/\delta)} \cdot \sigma_0 \|\boldsymbol{\mu}\|_2,$$
$$|\langle \mathbf{w}_{j,r}^{(0,0)}, \boldsymbol{\xi}_i \rangle| \leq 2\sqrt{\log(12mn/\delta)} \cdot \sigma_0 \sigma_p \sqrt{d}$$

*for all $r \in [m]$, $j \in \{\pm 1\}$ and $i \in [n]$. Moreover,*

$$\sigma_0 \|\boldsymbol{\mu}\|_2/2 \leq \max_{r \in [m]} j \cdot \langle \mathbf{w}_{j,r}^{(0,0)}, \boldsymbol{\mu} \rangle \leq \sqrt{2 \log(12m/\delta)} \cdot \sigma_0 \|\boldsymbol{\mu}\|_2,$$
$$\sigma_0 \sigma_p \sqrt{d}/4 \leq \max_{r \in [m]} j \cdot \langle \mathbf{w}_{j,r}^{(0,0)}, \boldsymbol{\xi}_i \rangle \leq 2\sqrt{\log(12mn/\delta)} \cdot \sigma_0 \sigma_p \sqrt{d}$$

*for all $j \in \{\pm 1\}$ and $i \in [n]$.*

**Lemma B.3.** *Let $S_i^{(t,b)}$ denote $\{r : \langle \mathbf{w}_{y_i,r}^{(t,b)}, \boldsymbol{\xi}_i \rangle > \sigma_0 \sigma_p \sqrt{d}/\sqrt{2}\}$. Suppose that $\delta > 0$ and $m \geq 50 \log(2n/\delta)$. Then with probability at least $1 - \delta$,*

$$|S_i^{(0,0)}| \geq 0.8\Phi(-1)m, \ \forall i \in [n].$$

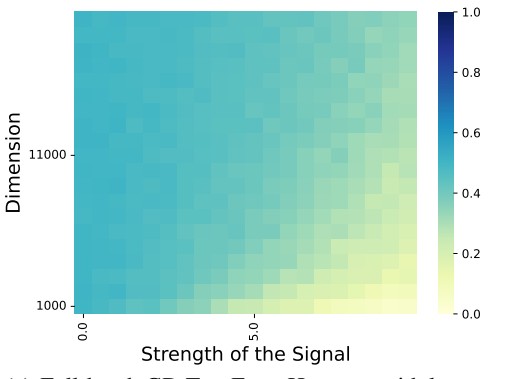 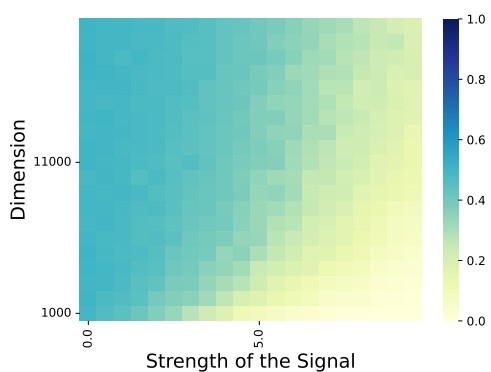

(a) Full-batch GD Test Error Heatmap with learning rate $\eta = 0.001$

(b) Full-batch GD Test Error Heatmap with learning rate $\eta = 0.01$

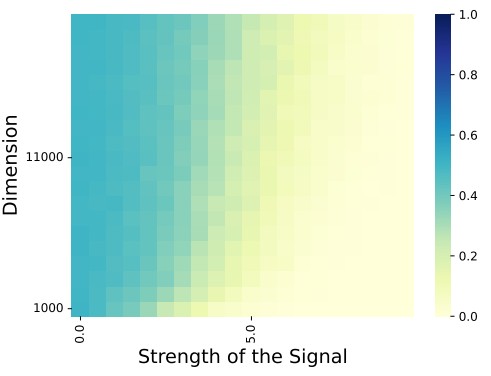 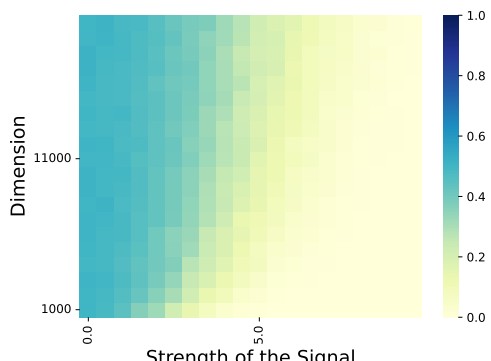

(c) Full-batch GD Test Error Heatmap with learning rate $\eta = 0.1$

(d) Full-batch GD Test Error Heatmap with learning rate $\eta = 1$

Figure 4: (a) is a heatmap illustrating test error on synthetic data for various dimensions $d$ and signal strengths $\boldsymbol{\mu}$ when trained using learning rate $0.001$. High test errors are represented in blue, while low test errors are shown in yellow. (b)(c)(d) displays a heatmap of test errors on the synthetic data under the same conditions as in (a), but trained using GD with different learning rates $0.01, 0.1, 1$. The y-axis represents a normal scale with a range of 1000-21000.

*Proof of Lemma B.3.* Since $\langle \mathbf{w}_{y_i,r}^{(0,0)}, \boldsymbol{\xi}_i \rangle \sim \mathcal{N}(0, \sigma_0^2 \|\boldsymbol{\xi}_i\|_2^2)$, we have

$$P(\langle \mathbf{w}_{y_i,r}^{(0,0)}, \boldsymbol{\xi}_i \rangle > \sigma_0 \sigma_p \sqrt{d}/\sqrt{2}) \geq P(\langle \mathbf{w}_{y_i,r}^{(0,0)}, \boldsymbol{\xi}_i \rangle > \sigma_0 \|\boldsymbol{\xi}_i\|_2) = \Phi(-1),$$

where $\Phi(\cdot)$ is CDF of the standard normal distribution. Note that $|S_i^{(0,0)}| = \sum_{r=1}^m \mathbb{1}[\langle \mathbf{w}_{y_i,r}^{(0,0)}, \boldsymbol{\xi}_i \rangle > \sigma_0 \sigma_p \sqrt{d}/\sqrt{2}]$ and $P(\langle \mathbf{w}_{y_i,r}^{(0,0)}, \boldsymbol{\xi}_i \rangle > \sigma_0 \sigma_p \sqrt{d}/\sqrt{2}) \geq \Phi(-1)$, then by Hoeffding's inequality, with probability at least $1 - \delta/n$, we have

$$\frac{|S_i^{(0,0)}|}{m} \geq \Phi(-1) - \sqrt{\frac{\log(2n/\delta)}{2m}}.$$

Therefore, as long as $0.2\sqrt{m}\Phi(-1) \geq \sqrt{\frac{\log(2n/\delta)}{2}}$, by applying union bound, with probability at least $1 - \delta$, we have

$$|S_i^{(0)}| \geq 0.8\Phi(-1)m, \ \forall i \in [n].$$

$\square$

**Lemma B.4.** *Let $S_{j,r}^{(t,b)}$ denote $\{i \in [n] : y_i = j, \ \langle \mathbf{w}_{y_i,r}^{(t,b)}, \boldsymbol{\xi}_i \rangle > \sigma_0 \sigma_p \sqrt{d}/\sqrt{2}\}$. Suppose that $\delta > 0$ and $n \geq 32 \log(4m/\delta)$. Then with probability at least $1 - \delta$,*

$$|S_{j,r}^{(0)}| \geq n\Phi(-1)/4, \ \forall j \in \{\pm 1\}, r \in [m].$$

*Proof of Lemma B.4.* Since $\langle \mathbf{w}_{j,r}^{(0,0)}, \boldsymbol{\xi}_i \rangle \sim \mathcal{N}(0, \sigma_0^2 \|\boldsymbol{\xi}_i\|_2^2)$, we have

$$P(\langle \mathbf{w}_{j,r}^{(0,0)}, \boldsymbol{\xi}_i \rangle > \sigma_0 \sigma_p \sqrt{d}/\sqrt{2}) \geq P(\langle \mathbf{w}_{j,r}^{(0,0)}, \boldsymbol{\xi}_i \rangle > \sigma_0 \|\boldsymbol{\xi}_i\|_2) = \Phi(-1),$$

where $\Phi(\cdot)$ is CDF of the standard normal distribution.

Note that $|S_{j,r}^{(0,0)}| = \sum_{i=1}^n \mathbb{1}[y_i = j]\,\mathbb{1}[\langle \mathbf{w}_{j,r}^{(0)}, \boldsymbol{\xi}_i \rangle > \sigma_0 \sigma_p \sqrt{d}/\sqrt{2}]$ and $\mathbb{P}(y_i = j, \langle \mathbf{w}_{j,r}^{(0)}, \boldsymbol{\xi}_i \rangle > \sigma_0 \sigma_p \sqrt{d}/\sqrt{2}) \geq \Phi(-1)/2$, then by Hoeffding's inequality, with probability at least $1 - \delta/2m$, we have

$$\frac{|S_{j,r}^{(0)}|}{n} \geq \Phi(-1)/2 + \sqrt{\frac{\log(4m/\delta)}{2n}}.$$

Therefore, as long as $\Phi(-1)/4 \geq \sqrt{\frac{\log(4m/\delta)}{2n}}$, by applying union bound, we have with probability at least $1 - \delta$,

$$|S_{j,r}^{(0)}| \geq n\Phi(-1)/4, \ \forall j \in \{\pm 1\}, r \in [m].$$

$\square$

**Lemma B.5** (Lemma B.3 in Kou et al. (2023))**.** *For $|S_+ \cap S_y|$ and $|S_- \cap S_y|$ where $y \in \{\pm 1\}$, it holds with probability at least $1 - \delta(\delta > 0)$ that*

$$\left| |S_+ \cap S_y| - \frac{(1-p)n}{2} \right| \leq \sqrt{\frac{n}{2}\log\left(\frac{8}{\delta}\right)}, \left| |S_- \cap S_y| - \frac{pn}{2} \right| \leq \sqrt{\frac{n}{2}\log\left(\frac{8}{\delta}\right)}, \forall y \in \{\pm 1\}.$$

**Lemma B.6.** *It holds with probability at least $1 - \delta$, for all $T \in [\frac{\log(2T^*/\delta)}{c_3^2}, T^*]$ and $y \in \{\pm 1\}$, there exist at least $c_3 \cdot T$ epochs among $[0, T]$, such that at least $c_4 \cdot H$ batches in these epochs, satisfy*

$$|S_+ \cap S_y \cap \mathcal{I}_{t,b}| \in \left[\frac{B}{4}, \frac{3B}{4}\right]. \tag{16}$$

*Proof.* Let

$$\mathcal{E}_{1,t} := \{\text{In epoch } t, \text{ there are at least } c_2 \cdot \frac{n}{B} \text{ batches such that 16 holds for } y = 1\},$$

$$\mathcal{E}_{1,t,b} := \{\text{In epoch } t \text{ natch } b, \text{ 16 holds for } y = 1\}.$$

First let $n$ big enough, then we have $S_+ \cap S_y \in \left[\frac{3(1-p)n}{8}, \frac{5(1-p)n}{8}\right]$. We consider the first $c_1 H$ batches. At the time we are starting to sample $h$-th batch in the first $c_1 H$ batches, suppose there are $n_1$ samples that belong to $S_+ \cap S_y$, and there are $n_2$ samples that don't belong to $S_+ \cap S_y$. Then $n_1 \geq \frac{3(1-p)n}{8} - c_1 n \geq \frac{5(1-p)n}{16}$ and $n_2 \geq \frac{3(1-p)n}{8} - c_1 n \geq \frac{5(1-p)n}{16}$.

$$\begin{aligned}
\mathbb{P}(\mathcal{E}_{1,t,h}) &= \frac{\sum_{l=B/4}^{3B/4} C_B^l C_{n_1}^l C_{n_2}^{B-l}}{C_n^B} \\
&\geq \frac{\sum_{l=1B/4}^{3B/4} C_B^l C_{\frac{5(1-p)n}{16}}^l C_{\frac{5(1-p)n}{16}}^{B-l}}{C_n^B} \\
&\geq \frac{\frac{B}{2} C_B^{B/4} \left(\frac{9(1-p)n}{32}\right)^B / B!}{n^B / B!} \\
&= \frac{B}{2} C_B^{B/4} \left(\frac{9(1-p)}{32}\right)^B := 2c_2.
\end{aligned}$$

Then, the probability that there are less than $c_1 c_2 H$ batches in first $c_1 H$ batches such that 16 holds is:

$$\sum_{i=0}^{c_1 c_2 H - 1} \sum_{\sum l_h = i} \mathbb{P}[\mathbb{1}(\mathcal{E}_{1,t,0}) = l_0]\mathbb{P}[\mathbb{1}(\mathcal{E}_{1,t,1}) = l_1 | \mathbb{1}(\mathcal{E}_{1,t,0}) = l_0] \cdots$$

$$\mathbb{P}[\mathbb{1}(\mathcal{E}_{1,t,c_1 H-1}) = l_{c_1 H-1} | \mathbb{1}(\mathcal{E}_{1,t,0}) = l_0, \cdots, \mathbb{1}(\mathcal{E}_{1,t,c_1 H-2}) = l_{c_1 H-2}]$$

$$\leq \sum_{i=0}^{c_1 c_2 H} C_{c_1 H}^i (1 - 2c_2)^{c_1 H - i}$$

$$\leq c_1 c_2 H \cdot (2c_2)^{c_1 c_2 H} (1 - 2c_2)^{c_1 H - c_1 c_2 H}.$$

Choose $H_0$ such that $c_1 c_2 H_0 \cdot (2c_2)^{c_1 c_2 H_0} (1 - 2c_2)^{c_1 H_0 - c_1 c_2 H_0} = 1 - 2c_3$, then as long as $H \geq H_0$, with probability $c_3$, there are at least $c_1 c_2 H$ batches in first $c_1 H$ batches such that 16 holds. Then $\mathbb{P}[\mathcal{E}_{1,t}] \geq 2c_3$. Therefore,

$$\mathbb{P}(\sum_t \mathbb{1}(\mathcal{E}_t) - 2Tc_3 \leq -t) \leq \exp(-\frac{2t^2}{T}).$$

Let $T \geq \frac{\log(2T^*/\delta)}{2c_3^2}$. Then, with probability at least $1 - \delta/(2T^*)$,

$$\sum_t \mathbb{1}(\mathcal{E}_{1,t}) \geq c_3 T.$$

Let $c_4 = c_1 c_2$. Thus, there are at least $c_3 T^*$ epochs, such that they have at least $c_4 H$ batches satisfying EquationB.6. This also holds for $y = -1$. Taking a union bound to get the result. $\qquad\square$

## C    Proofs for SGD

In this section, we prove the results for SGD. We first define some notations. Define $H = n/B$ as the number of batches within an epoch. For any $t_1, t_2$ and $b_1, b_2 \in \overline{[H]}$, we write $(t_1, b_1) \leq (t, b) \leq (t_2, b_2)$ to denote all iterations from $t_1$-th epoch's $b_1$-th batch (included) to $t_2$-th epoch's $b_2$-th batch (included). And the meanings change accordingly if we replace $\leq$ with $<$.

### C.1    Signal-noise Decomposition Coefficient Analysis

This part is dedicated to analyzing the update rule of Signal-noise Decomposition Coefficients. It is worth noting that

$$F_j(\mathbf{W}, \mathbf{X}) = \frac{1}{m} \sum_{r=1}^m \sum_{p=1}^P \sigma(\langle \mathbf{w}_{j,r}, \mathbf{x}_p \rangle) = \frac{1}{m} \sum_{r=1}^m \sigma(\langle \mathbf{w}_{j,r}, \widehat{y}\boldsymbol{\mu} \rangle) + (P-1)\sigma(\langle \mathbf{w}_{j,r}, \boldsymbol{\xi} \rangle).$$

Let $\mathcal{I}_{t,b}$ denote the set of indices of randomly chosen samples at epoch $t$ batch $b$, and $|\mathcal{I}_{t,b}| = B$, then the update rule is:

$$\text{for } b \in \overline{[H]} \qquad \mathbf{w}_{j,r}^{(t,b+1)} = \mathbf{w}_{j,r}^{(t,b)} - \eta \cdot \nabla_{\mathbf{w}_{j,r}} L_{\mathcal{I}_{t,b}}(\mathbf{W}^{(t,b)})$$

$$= \mathbf{w}_{j,r}^{(t,b)} - \frac{\eta(P-1)}{Bm} \sum_{i \in \mathcal{I}_{t,b}} \ell_i^{\prime(t,b)} \cdot \sigma'(\langle \mathbf{w}_{j,r}^{(t,b)}, \boldsymbol{\xi}_i \rangle) \cdot j y_i \boldsymbol{\xi}_i$$

$$- \frac{\eta}{Bm} \sum_{i \in \mathcal{I}_{t,b}} \ell_i^{\prime(t,b)} \cdot \sigma'(\langle \mathbf{w}_{j,r}^{(t,b)}, \widehat{y}_i \boldsymbol{\mu} \rangle) \cdot j y_i \widehat{y}_i \boldsymbol{\mu},$$

$$\text{and} \qquad \mathbf{w}_{j,r}^{(t+1,0)} = \mathbf{w}_{j,r}^{(t,H)}. \tag{17}$$

#### C.1.1    Iterative Expression for Decomposition Coefficient Analysis

**Lemma C.1.** *The coefficients $\gamma_{j,r}^{(t,b)}, \overline{\rho}_{j,r,i}^{(t,b)}, \underline{\rho}_{j,r,i}^{(t,b)}$ defined in Lemma 3.3 satisfy the following iterative equations:*

$$\gamma_{j,r}^{(0,0)}, \overline{\rho}_{j,r,i}^{(0,0)}, \underline{\rho}_{j,r,i}^{(0,0)} = 0, \tag{18}$$

$$\gamma_{j,r}^{(t,b+1)} = \gamma_{j,r}^{(t,b)} - \frac{\eta}{Bm} \cdot \left[ \sum_{i \in \mathcal{I}_{t,b} \cap S_+} \ell_i^{\prime(t,b)} \sigma'(\langle \mathbf{w}_{j,r}^{(t,b)}, \widehat{y}_i \cdot \boldsymbol{\mu} \rangle) \right.$$

$$-\sum_{i\in\mathcal{I}_{t,b}\cap S_-}\ell_i'^{(t,b)}\sigma'(\langle\mathbf{w}_{j,r}^{(t,b)},\widehat{y}_i\cdot\boldsymbol{\mu}\rangle)\Bigg]\cdot\|\boldsymbol{\mu}\|_2^2,\tag{19}$$

$$\overline{\rho}_{j,r,i}^{(t,b+1)}=\overline{\rho}_{j,r,i}^{(t,b)}-\frac{\eta(P-1)^2}{Bm}\cdot\ell_i'^{(t,b)}\cdot\sigma'(\langle\mathbf{w}_{j,r}^{(t,b)},\boldsymbol{\xi}_i\rangle)\cdot\|\boldsymbol{\xi}_i\|_2^2\cdot\mathbb{1}(y_i=j)\,\mathbb{1}(i\in\mathcal{I}_{t,b}),\tag{20}$$

$$\underline{\rho}_{j,r,i}^{(t,b+1)}=\underline{\rho}_{j,r,i}^{(t,b)}+\frac{\eta(P-1)^2}{Bm}\cdot\ell_i'^{(t,b)}\cdot\sigma'(\langle\mathbf{w}_{j,r}^{(t,b)},\boldsymbol{\xi}_i\rangle)\cdot\|\boldsymbol{\xi}_i\|_2^2\cdot\mathbb{1}(y_i=-j)\,\mathbb{1}(i\in\mathcal{I}_{t,b}),\tag{21}$$

*for all* $r\in[m]$, $j\in\{\pm1\}$ *and* $i\in[n]$.

*Proof.* First, we iterate the gradient descent update rule $t$ epochs plus $b$ batches and get

$$\mathbf{w}_{j,r}^{(t,b)}=\mathbf{w}_{j,r}^{(0,0)}-\frac{\eta(P-1)}{Bm}\sum_{(t',b')<(t,b)}\sum_{i\in\mathcal{I}_{t',b'}}\ell_i'^{(t',b')}\cdot\sigma'(\langle\mathbf{w}_{j,r}^{(t',b')},\boldsymbol{\xi}_i\rangle)\cdot jy_i(P-1)\boldsymbol{\xi}_i$$
$$-\frac{\eta}{Bm}\sum_{(t',b')<(t,b)}\sum_{i\in\mathcal{I}_{s,k}}\ell_i'^{(t',b')}\cdot\sigma'(\langle\mathbf{w}_{j,r}^{(t',b')},\widehat{y}_i\boldsymbol{\mu}\rangle)\cdot y_i\widehat{y}_ij\boldsymbol{\mu}.$$

According to the definition of $\gamma_{j,r}^{(t)}$ and $\rho_{j,r,i}^{(t)}$,

$$\mathbf{w}_{j,r}^{(t,b)}=\mathbf{w}_{j,r}^{(0,0)}+j\cdot\gamma_{j,r}^{(t,b)}\cdot\|\boldsymbol{\mu}\|_2^{-2}\cdot\boldsymbol{\mu}+\frac{1}{P-1}\sum_{i=1}^n\rho_{j,r,i}^{(t,b)}\cdot\|\boldsymbol{\xi}_i\|_2^{-2}\cdot\boldsymbol{\xi}_i.$$

Since $\boldsymbol{\xi}_i$ and $\boldsymbol{\mu}$ are linearly independent with probability 1, we have the unique representation as follows:

$$\rho_{j,r,i}^{(t,b)}=-\frac{\eta(P-1)^2}{Bm}\sum_{(t',b')<(t,b)}\ell_i'^{(t',b')}\cdot\sigma'(\langle\mathbf{w}_{j,r}^{(t',b')},\boldsymbol{\xi}_i\rangle)\cdot\|\boldsymbol{\xi}_i\|_2^2\mathbb{1}(i\in\mathcal{I}_{s,k})y_ij,$$

$$\gamma_{j,r}^{(t,b)}=-\frac{\eta}{Bm}\sum_{(t',b')<(t,b)}\Bigg[\sum_{i\in\mathcal{I}_{t',b'}\cap S_+}\ell_i'^{(t',b')}\sigma'(\langle\mathbf{w}_{j,r}^{(t',b')},y_i\cdot\boldsymbol{\mu}\rangle)$$
$$-\sum_{i\in\mathcal{I}_{t',b'}\cap S_-}\ell_i'^{(t',b')}\sigma'(\langle\mathbf{w}_{j,r}^{(t',b')},y_i\cdot\boldsymbol{\mu}\rangle)\Bigg]\|\boldsymbol{\mu}\|_2^2.$$

Since we define $\overline{\rho}_{j,r,i}^{(t,b)}:=\rho_{j,r,i}^{(t,b)}\mathbb{1}(\rho_{j,r,i}^{(t,b)}\geq0)$, $\underline{\rho}_{j,r,i}^{(t)}:=\rho_{j,r,i}^{(t,b)}\mathbb{1}(\rho_{j,r,i}^{(t,b)}\leq0)$, we obtain

$$\overline{\rho}_{j,r,i}^{(t,b)}=-\frac{\eta(P-1)^2}{Bm}\sum_{(t',b')<(t,b)}\ell_i'^{(t',b')}\cdot\sigma'(\langle\mathbf{w}_{j,r}^{(t',b')},\boldsymbol{\xi}_i\rangle)\cdot\|\boldsymbol{\xi}_i\|_2^2\mathbb{1}(i\in\mathcal{I}_{t',b'})\mathbb{1}(y_i=j),$$

$$\underline{\rho}_{j,r,i}^{(t,b)}=\frac{\eta(P-1)^2}{Bm}\sum_{(t',b')<(t,b)}\ell_i'^{(t',b')}\cdot\sigma'(\langle\mathbf{w}_{j,r}^{(t',b')},\boldsymbol{\xi}_i\rangle)\cdot\|\boldsymbol{\xi}_i\|_2^2\mathbb{1}(i\in\mathcal{I}_{t',b'})\mathbb{1}(y_i=-j).$$

And the iterative update equations (19), (20), and (21) follow directly. $\qquad\square$

### C.1.2 Scale of Decomposition Coefficients

We first define $T^*=\eta^{-1}\text{poly}(\epsilon^{-1},d,n,m)$ and

$$\alpha:=4\log(T^*),\tag{22}$$

$$\beta:=2\max_{i,j,r}\{|\langle\mathbf{w}_{j,r}^{(0,0)},\boldsymbol{\mu}\rangle|,(P-1)|\langle\mathbf{w}_{j,r}^{(0,0)},\boldsymbol{\xi}_i\rangle|\},\tag{23}$$

$$\text{SNR}:=\frac{\|\boldsymbol{\mu}\|_2}{(P-1)\sigma_p\sqrt{d}}.\tag{24}$$

By Lemma B.2 and Condition 3.1, $\beta$ can be bounded as

$$\beta=2\max_{i,j,r}\{|\langle\mathbf{w}_{j,r}^{(0,0)},\boldsymbol{\mu}\rangle|,(P-1)|\langle\mathbf{w}_{j,r}^{(0,0)},\boldsymbol{\xi}_i\rangle|\}$$

$$\leq 2 \max\{\sqrt{2\log(12m/\delta)} \cdot \sigma_0 \|\boldsymbol{\mu}\|_2, 2\sqrt{\log(12mn/\delta)} \cdot \sigma_0(P-1)\sigma_p\sqrt{d}\}$$
$$= O\big(\sqrt{\log(mn/\delta)} \cdot \sigma_0(P-1)\sigma_p\sqrt{d}\big),$$

Then, by Condition 3.1, we have the following inequality:

$$\max\left\{\beta, \mathrm{SNR}\sqrt{\frac{32\log(6n/\delta)}{d}}n\alpha, 5\sqrt{\frac{\log(6n^2/\delta)}{d}}n\alpha\right\} \leq \frac{1}{12}. \tag{25}$$

We first prove the following bounds for signal-noise decomposition coefficients.

**Proposition C.2.** *Under Assumption 3.1, for $(0,0) \leq (t,b) \leq (T^*, 0)$, we have that*

$$\gamma_{j,r}^{(0,0)}, \overline{\rho}_{j,r,i}^{(0,0)}, \underline{\rho}_{j,r,i}^{(0,0)} = 0 \tag{26}$$

$$0 \leq \overline{\rho}_{j,r,i}^{(t,b)} \leq \alpha, \tag{27}$$

$$0 \geq \underline{\rho}_{j,r,i}^{(t,b)} \geq -\beta - 10\sqrt{\frac{\log(6n^2/\delta)}{d}}n\alpha \geq -\alpha, \tag{28}$$

*and there exists a positive constant $C'$ such that*

$$-\frac{1}{12} \leq \gamma_{j,r}^{(t,b)} \leq C'\widehat{\gamma}\alpha, \tag{29}$$

*for all $r \in [m]$, $j \in \{\pm 1\}$ and $i \in [n]$, where $\widehat{\gamma} := n \cdot \mathrm{SNR}^2$.*

We will prove Proposition C.2 by induction. We first approximate the change of the inner product by corresponding decomposition coefficients when Proposition C.2 holds.

**Lemma C.3.** *Under Assumption 3.1, suppose* (27), (28) *and* (29) *hold after $b$-th batch of $t$-th epoch. Then, for all $r \in [m]$, $j \in \{\pm 1\}$ and $i \in [n]$,*

$$\left|\langle \mathbf{w}_{j,r}^{(t,b)} - \mathbf{w}_{j,r}^{(0,0)}, \boldsymbol{\mu}\rangle - j \cdot \gamma_{j,r}^{(t,b)}\right| \leq \mathrm{SNR}\sqrt{\frac{32\log(6n/\delta)}{d}}n\alpha, \tag{30}$$

$$\left|\langle \mathbf{w}_{j,r}^{(t,b)} - \mathbf{w}_{j,r}^{(0,0)}, \boldsymbol{\xi}_i\rangle - \frac{1}{P-1}\underline{\rho}_{j,r,i}^{(t,b)}\right| \leq \frac{5}{P-1}\sqrt{\frac{\log(6n^2/\delta)}{d}}n\alpha, \ j \neq y_i, \tag{31}$$

$$\left|\langle \mathbf{w}_{j,r}^{(t,b)} - \mathbf{w}_{j,r}^{(0,0)}, \boldsymbol{\xi}_i\rangle - \frac{1}{P-1}\overline{\rho}_{j,r,i}^{(t,b)}\right| \leq \frac{5}{P-1}\sqrt{\frac{\log(6n^2/\delta)}{d}}n\alpha, \ j = y_i. \tag{32}$$

*Proof of Lemma C.3.* First, for any time $(t,b) \geq (0,0)$, we have from the following decomposition by definitions,

$$\langle \mathbf{w}_{j,r}^{(t,b)} - \mathbf{w}_{j,r}^{(0,0)}, \boldsymbol{\mu}\rangle = j \cdot \gamma_{j,r}^{(t,b)} + \frac{1}{P-1}\sum_{i'=1}^{n}\overline{\rho}_{j,r,i'}^{(t,b)}\|\boldsymbol{\xi}_{i'}\|_2^{-2} \cdot \langle \boldsymbol{\xi}_{i'}, \boldsymbol{\mu}\rangle$$

$$+ \frac{1}{P-1}\sum_{i'=1}^{n}\underline{\rho}_{j,r,i'}^{(t,b)}\|\boldsymbol{\xi}_{i'}\|_2^{-2} \cdot \langle \boldsymbol{\xi}_{i'}, \boldsymbol{\mu}\rangle.$$

According to Lemma B.1, we have

$$\left|\frac{1}{P-1}\sum_{i'=1}^{n}\overline{\rho}_{j,r,i'}^{(t,b)}\|\boldsymbol{\xi}_{i'}\|_2^{-2} \cdot \langle \boldsymbol{\xi}_{i'}, \boldsymbol{\mu}\rangle + \frac{1}{P-1}\sum_{i'=1}^{n}\underline{\rho}_{j,r,i'}^{(t,b)}\|\boldsymbol{\xi}_{i'}\|_2^{-2} \cdot \langle \boldsymbol{\xi}_{i'}, \boldsymbol{\mu}\rangle\right|$$

$$\leq \frac{1}{P-1}\sum_{i'=1}^{n}|\overline{\rho}_{j,r,i'}^{(t,b)}|\|\boldsymbol{\xi}_{i'}\|_2^{-2} \cdot |\langle \boldsymbol{\xi}_{i'}, \boldsymbol{\mu}\rangle| + \frac{1}{P-1}\sum_{i'=1}^{n}|\underline{\rho}_{j,r,i'}^{(t,b)}|\|\boldsymbol{\xi}_{i'}\|_2^{-2} \cdot |\langle \boldsymbol{\xi}_{i'}, \boldsymbol{\mu}\rangle|$$

$$\leq \frac{2\|\boldsymbol{\mu}\|_2\sqrt{2\log(6n/\delta)}}{(P-1)\sigma_p d}\left(\sum_{i'=1}^{n}|\overline{\rho}_{j,r,i'}^{(t,b)}| + \sum_{i'=1}^{n}|\underline{\rho}_{j,r,i'}^{(t,b)}|\right)$$

$$= \text{SNR}\sqrt{\frac{8\log(6n/\delta)}{d}}\left(\sum_{i'=1}^{n}|\overline{\rho}_{j,r,i'}^{(t,b)}| + \sum_{i'=1}^{n}|\underline{\rho}_{j,r,i'}^{(t,b)}|\right)$$

$$\leq \text{SNR}\sqrt{\frac{32\log(6n/\delta)}{d}}n\alpha,$$

where the first inequality is by triangle inequality, the second inequality is by Lemma B.1, the equality is by $\text{SNR} = \|\boldsymbol{\mu}\|_2/((P-1)\sigma_p\sqrt{d})$, and the last inequality is by (27), (28). It follows that

$$|\langle \mathbf{w}_{j,r}^{(t,b)} - \mathbf{w}_{j,r}^{(0,0)}, \boldsymbol{\mu}\rangle - j\cdot\gamma_{j,r}^{(t,b)}| \leq \text{SNR}\sqrt{\frac{32\log(6n/\delta)}{d}}n\alpha.$$

Then, for $j \neq y_i$ and any $t \geq 0$, we have

$$\langle \mathbf{w}_{j,r}^{(t,b)} - \mathbf{w}_{j,r}^{(0,0)}, \boldsymbol{\xi}_i\rangle$$

$$= j\cdot\gamma_{j,r}^{(t,b)}\|\boldsymbol{\mu}\|_2^{-2}\cdot\langle\boldsymbol{\mu},\boldsymbol{\xi}_i\rangle + \frac{1}{P-1}\sum_{i'=1}^{n}\overline{\rho}_{j,r,i'}^{(t,b)}\|\boldsymbol{\xi}_{i'}\|_2^{-2}\cdot\langle\boldsymbol{\xi}_{i'},\boldsymbol{\xi}_i\rangle$$

$$+ \frac{1}{P-1}\sum_{i'=1}^{n}\underline{\rho}_{j,r,i'}^{(t,b)}\|\boldsymbol{\xi}_{i'}\|_2^{-2}\cdot\langle\boldsymbol{\xi}_{i'},\boldsymbol{\xi}_i\rangle$$

$$= j\cdot\gamma_{j,r}^{(t,b)}\|\boldsymbol{\mu}\|_2^{-2}\cdot\langle\boldsymbol{\mu},\boldsymbol{\xi}_i\rangle + \frac{1}{P-1}\sum_{i'=1}^{n}\underline{\rho}_{j,r,i'}^{(t,b)}\|\boldsymbol{\xi}_{i'}\|_2^{-2}\cdot\langle\boldsymbol{\xi}_{i'},\boldsymbol{\xi}_i\rangle$$

$$= \frac{1}{P-1}\underline{\rho}_{j,r,i}^{(t,b)} + j\cdot\gamma_{j,r}^{(t,b)}\|\boldsymbol{\mu}\|_2^{-2}\cdot\langle\boldsymbol{\mu},\boldsymbol{\xi}_i\rangle + \frac{1}{P-1}\sum_{i'\neq i}\underline{\rho}_{j,r,i'}^{(t,b)}\|\boldsymbol{\xi}_{i'}\|_2^{-2}\cdot\langle\boldsymbol{\xi}_{i'},\boldsymbol{\xi}_i\rangle,$$

where the second equality is due to $\overline{\rho}_{j,r,i}^{(t,b)} = 0$ for $j \neq y_i$. Next, we have

$$\left|j\cdot\gamma_{j,r}^{(t,b)}\|\boldsymbol{\mu}\|_2^{-2}\cdot\langle\boldsymbol{\mu},\boldsymbol{\xi}_i\rangle + \frac{1}{P-1}\sum_{i'\neq i}\underline{\rho}_{j,r,i'}^{(t,b)}\|\boldsymbol{\xi}_{i'}\|_2^{-2}\cdot\langle\boldsymbol{\xi}_{i'},\boldsymbol{\xi}_i\rangle\right|$$

$$\leq |\gamma_{j,r}^{(t,b)}|\|\boldsymbol{\mu}\|_2^{-2}\cdot|\langle\boldsymbol{\mu},\boldsymbol{\xi}_i\rangle| + \frac{1}{P-1}\sum_{i'\neq i}|\underline{\rho}_{j,r,i'}^{(t,b)}|\|\boldsymbol{\xi}_{i'}\|_2^{-2}\cdot|\langle\boldsymbol{\xi}_{i'},\boldsymbol{\xi}_i\rangle|$$

$$\leq |\gamma_{j,r}^{(t,b)}|\|\boldsymbol{\mu}\|_2^{-1}\sigma_p\sqrt{2\log(6n/\delta)} + \frac{4}{P-1}\sqrt{\frac{\log(6n^2/\delta)}{d}}\sum_{i'\neq i}|\underline{\rho}_{j,r,i'}^{(t,b)}|$$

$$= \frac{\text{SNR}^{-1}}{P-1}\sqrt{\frac{2\log(6n/\delta)}{d}}|\gamma_{j,r}^{(t,b)}| + \frac{4}{P-1}\sqrt{\frac{\log(6n^2/\delta)}{d}}\sum_{i'\neq i}|\underline{\rho}_{j,r,i'}^{(t,b)}|$$

$$\leq \frac{\text{SNR}}{P-1}\sqrt{\frac{8C^2\log(6n/\delta)}{d}}n\alpha + \frac{4}{P-1}\sqrt{\frac{\log(6n^2/\delta)}{d}}n\alpha$$

$$\leq \frac{5}{P-1}\sqrt{\frac{\log(6n^2/\delta)}{d}}n\alpha,$$

where the first inequality is by triangle inequality; the second inequality is by Lemma B.1; the equality is by $\text{SNR} = \|\boldsymbol{\mu}\|_2/\sigma_p\sqrt{d}$; the third inequality is by (28) and (29); the forth inequality is by $\text{SNR} \leq 1/\sqrt{8C'^2}$. Therefore, for $j \neq y_i$, we have

$$|\langle\mathbf{w}_{j,r}^{(t,b)} - \mathbf{w}_{j,r}^{(0,0)}, \boldsymbol{\xi}_i\rangle - \frac{1}{P-1}\underline{\rho}_{j,r,i}^{(t,b)}| \leq \frac{5}{P-1}\sqrt{\frac{\log(6n^2/\delta)}{d}}n\alpha.$$

Similarly, we have for $y_i = j$ that

$$\langle\mathbf{w}_{j,r}^{(t,b)} - \mathbf{w}_{j,r}^{(0,0)}, \boldsymbol{\xi}_i\rangle$$

$$= j\cdot\gamma_{j,r}^{(t,b)}\|\boldsymbol{\mu}\|_2^{-2}\cdot\langle\boldsymbol{\mu},\boldsymbol{\xi}_i\rangle + \frac{1}{P-1}\sum_{i'=1}^{n}\overline{\rho}_{j,r,i'}^{(t,b)}\|\boldsymbol{\xi}_{i'}\|_2^{-2}\cdot\langle\boldsymbol{\xi}_{i'},\boldsymbol{\xi}_i\rangle$$

$$+ \frac{1}{P-1} \sum_{i'=1}^{n} \underline{\rho}_{j,r,i'}^{(t,b)} \|\boldsymbol{\xi}_{i'}\|_2^{-2} \cdot \langle \boldsymbol{\xi}_{i'}, \boldsymbol{\xi}_i \rangle$$

$$= j \cdot \gamma_{j,r}^{(t,b)} \|\boldsymbol{\mu}\|_2^{-2} \cdot \langle \boldsymbol{\mu}, \boldsymbol{\xi}_i \rangle + \frac{1}{P-1} \sum_{i'=1}^{n} \overline{\rho}_{j,r,i'}^{(t,b)} \|\boldsymbol{\xi}_{i'}\|_2^{-2} \cdot \langle \boldsymbol{\xi}_{i'}, \boldsymbol{\xi}_i \rangle$$

$$= \frac{1}{P-1} \overline{\rho}_{j,r,i}^{(t,b)} + j \cdot \gamma_{j,r}^{(t,b)} \|\boldsymbol{\mu}\|_2^{-2} \cdot \langle \boldsymbol{\mu}, \boldsymbol{\xi}_i \rangle + \frac{1}{P-1} \sum_{i' \neq i} \overline{\rho}_{j,r,i'}^{(t,b)} \|\boldsymbol{\xi}_{i'}\|_2^{-2} \cdot \langle \boldsymbol{\xi}_{i'}, \boldsymbol{\xi}_i \rangle,$$

and

$$\left| j \cdot \gamma_{j,r}^{(t,b)} \|\boldsymbol{\mu}\|_2^{-2} \cdot \langle \boldsymbol{\mu}, \boldsymbol{\xi}_i \rangle + \sum_{i' \neq i} \overline{\rho}_{j,r,i'}^{(t,b)} \|\boldsymbol{\xi}_{i'}\|_2^{-2} \cdot \langle \boldsymbol{\xi}_{i'}, \boldsymbol{\xi}_i \rangle \right|$$

$$\leq \frac{\mathrm{SNR}^{-1}}{P-1} \sqrt{\frac{2\log(6n/\delta)}{d}} |\gamma_{j,r}^{(t,b)}| + \frac{4}{P-1} \sqrt{\frac{\log(6n^2/\delta)}{d}} \sum_{i' \neq i} |\overline{\rho}_{j,r,i'}^{(t,b)}|$$

$$\leq \frac{\mathrm{SNR}}{P-1} \sqrt{\frac{8C^2 \log(6n/\delta)}{d}} n\alpha + \frac{4}{P-1} \sqrt{\frac{\log(6n^2/\delta)}{d}} n\alpha$$

$$\leq \frac{5}{P-1} \sqrt{\frac{\log(6n^2/\delta)}{d}} n\alpha,$$

where the second inequality is by (27) and (29), and the third inequality is by $\mathrm{SNR} \leq 1/\sqrt{8C'^2}$. Therefore, for $j = y_i$, we have

$$\left| \langle \mathbf{w}_{j,r}^{(t,b)} - \mathbf{w}_{j,r}^{(0,0)}, \boldsymbol{\xi}_i \rangle - \frac{1}{P-1} \overline{\rho}_{j,r,i}^{(t,b)} \right| \leq \frac{5}{P-1} \sqrt{\frac{\log(6n^2/\delta)}{d}} n\alpha.$$

$\square$

**Lemma C.4.** *Under Condition 3.1, suppose* (27), (28) *and* (29) *hold after $b$-th batch of $t$-th epoch. Then, for all $j \neq y_i$, $j \in \{\pm 1\}$ and $i \in [n]$, $F_j(\mathbf{W}_j^{(t,b)}, \mathbf{x}_i) \leq 0.5$.*

*Proof of Lemma C.4.* According to Lemma C.3, we have

$$F_j(\mathbf{W}_j^{(t,b)}, \mathbf{x}_i) = \frac{1}{m} \sum_{r=1}^{m} [\sigma(\langle \mathbf{w}_{j,r}^{(t,b)}, y_i \boldsymbol{\mu} \rangle) + (P-1)\sigma(\langle \mathbf{w}_{j,r}^{(t,b)}, \boldsymbol{\xi}_i \rangle)]$$

$$\leq 2 \max\{\langle \mathbf{w}_{j,r}^{(t,b)}, y_i \boldsymbol{\mu} \rangle, (P-1)\langle \mathbf{w}_{j,r}^{(t,b)}, \boldsymbol{\xi}_i \rangle, 0\}$$

$$\leq 6 \max \left\{ \langle \mathbf{w}_{j,r}^{(0)}, y_i \boldsymbol{\mu} \rangle, (P-1)\langle \mathbf{w}_{j,r}^{(0)}, \boldsymbol{\xi}_i \rangle, \mathrm{SNR}\sqrt{\frac{32\log(6n/\delta)}{d}} n\alpha, y_i j \gamma_{j,r}^{(t,b)}, \right.$$

$$\left. 5\sqrt{\frac{\log(6n^2/\delta)}{d}} n\alpha + \underline{\rho}_{j,r,i}^{(t,b)} \right\}$$

$$\leq 6 \max \left\{ \beta/2, \mathrm{SNR}\sqrt{\frac{32\log(6n/\delta)}{d}} n\alpha, -\gamma_{j,r}^{(t,b)}, 5\sqrt{\frac{\log(6n^2/\delta)}{d}} n\alpha \right\}$$

$$\leq 0.5,$$

where the second inequality is by (30), (31) and (32); the third inequality is due to the definition of $\beta$ and $\underline{\rho}_{j,r,i}^{(t,b)} < 0$; the third inequality is by (25) and $-\gamma_{j,r}^{(t,b)} \leq \frac{1}{12}$.

$\square$

**Lemma C.5.** *Under Condition 3.1, suppose* (27), (28) *and* (29) *hold at $b$-th batch of $t$-th epoch. Then, it holds that*

$$(P-1)\langle \mathbf{w}_{y_i,r}^{(t,b)}, \boldsymbol{\xi}_i \rangle \geq -0.25,$$

$$(P-1)\langle \mathbf{w}_{y_i,r}^{(t,b)}, \boldsymbol{\xi}_i \rangle \leq (P-1)\sigma(\langle \mathbf{w}_{y_i,r}^{(t,b)}, \boldsymbol{\xi}_i \rangle) \leq (P-1)\langle \mathbf{w}_{y_i,r}^{(t,b)}, \boldsymbol{\xi}_i \rangle + 0.25,$$

*for any $i \in [n]$.*

*Proof of Lemma C.5.* According to (32) in Lemma C.3, we have

$$(P-1)\langle \mathbf{w}_{y_i,r}^{(t,b)}, \boldsymbol{\xi}_i\rangle \geq (P-1)\langle \mathbf{w}_{y_i,r}^{(0,0)}, \boldsymbol{\xi}_i\rangle + \overline{\rho}_{y_i,r,i}^{(t,b)} - 5n\sqrt{\frac{\log(6n^2/\delta)}{d}}\alpha$$

$$\geq -\beta - 5n\sqrt{\frac{\log(6n^2/\delta)}{d}}\alpha$$

$$\geq -0.25,$$

where the second inequality is due to $\overline{\rho}_{y_i,r,i}^{(t,b)} \geq 0$, the third inequality is due to $\beta < 1/8$ and $5n\sqrt{\log(6n^2/\delta)/d} \cdot \alpha < 1/8$ by Condition 3.1.

For the second equation, the first inequality holds naturally since $z \leq \sigma(z)$. For the inequality, if $\langle \mathbf{w}_{y_i,r}^{(t)}, \boldsymbol{\xi}_i\rangle \leq 0$, we have

$$(P-1)\sigma(\langle \mathbf{w}_{y_i,r}^{(t,b)}, \boldsymbol{\xi}_i\rangle) = 0 \leq (P-1)\langle \mathbf{w}_{y_i,r}^{(t,b)}, \boldsymbol{\xi}_i\rangle + 0.25.$$

And if $\langle \mathbf{w}_{y_i,r}^{(t,b)}, \boldsymbol{\xi}_i\rangle > 0$, we have

$$(P-1)\sigma(\langle \mathbf{w}_{y_i,r}^{(t,b)}, \boldsymbol{\xi}_i\rangle) = (P-1)\langle \mathbf{w}_{y_i,r}^{(t,b)}, \boldsymbol{\xi}_i\rangle < (P-1)\langle \mathbf{w}_{y_i,r}^{(t,b)}, \boldsymbol{\xi}_i\rangle + 0.25.$$

$\square$

**Lemma C.6** (Lemma C.6 in Kou et al. (2023)). *Let $g(z) = \ell'(z) = -1/(1+\exp(z))$, then for all $z_2 - c \geq z_1 \geq -1$ where $c \geq 0$ we have that*

$$\frac{\exp(c)}{4} \leq \frac{g(z_1)}{g(z_2)} \leq \exp(c).$$

**Lemma C.7.** *For any iteration $t \in [0, T^*)$ and $b, b_1, b_2 \in \overline{[H]}$, we have the following statements hold:*

1. $\left| \sum_{r=1}^m \left[\overline{\rho}_{y_i,r,i}^{(t,0)} - \overline{\rho}_{y_k,r,k}^{(t,0)}\right] - \sum_{r=1}^m \left[\overline{\rho}_{y_i,r,i}^{(t,b_1)} - \overline{\rho}_{y_k,r,k}^{(t,b_2)}\right]\right| \leq 0.1\kappa$.

2. $\langle \mathbf{w}_{y_i,r}^{(t,b)}, \boldsymbol{\xi}_i\rangle \geq \langle \mathbf{w}_{y_i,r}^{(t,0)}, \boldsymbol{\xi}_i\rangle - \sigma_0\sigma_p\sqrt{d}/\sqrt{2}$.

3. *Let $\widetilde{S}_i^{(t,b)} = \{r \in [m] : \langle \mathbf{w}_{y_i,r}^{(t,b)}, \boldsymbol{\xi}_i\rangle > 0\}$, then we have*
$$S_i^{(t,0)} \subseteq \widetilde{S}_i^{(t,b)}.$$

4. *Let $\widetilde{S}_{j,r}^{(t,b)} = \{i \in [n] : y_i = j, \langle \mathbf{w}_{j,r}^{(t,b)}, \boldsymbol{\xi}_i\rangle > 0\}$, then we have*
$$S_{j,r}^{(t,0)} \subseteq \widetilde{S}_{j,r}^{(t,b)}.$$

*Proof.* For the first statement,

$$\left| \sum_{r=1}^m \left[\overline{\rho}_{y_i,r,i}^{(t,0)} - \overline{\rho}_{y_k,r,k}^{(t,0)}\right] - \sum_{r=1}^m \left[\overline{\rho}_{y_i,r,i}^{(t,b_1)} - \overline{\rho}_{y_k,r,k}^{(t,b_2)}\right]\right|$$

$$\leq \frac{\eta(P-1)^2}{Bm} \max\left\{|S_i^{(\widetilde{t}-1,b_1)}||\ell_i'^{(\widetilde{t}-1,b_1)}| \cdot \|\boldsymbol{\xi}_i\|_2^2, |S_k^{(\widetilde{t}-1,b_2)}||\ell_k'^{(\widetilde{t}-1,b_2)}| \cdot \|\boldsymbol{\xi}_k\|_2^2\right\}$$

$$\leq \frac{\eta(P-1)^2}{B}\frac{3\sigma_p^2 d}{2}$$

$$\leq 0.1\kappa,$$

where the first inequality follows from the iterative update rule of $\overline{\rho}_{j,r,i}^{(t,b)}$, the second inequality is due to Lemma B.2, and the last inequality is due to Condition 3.1.

For the second statement, recall that the stochastic gradient update rule is

$$\langle \mathbf{w}_{y_i,r}^{(t,b)}, \boldsymbol{\xi}_i\rangle = \langle \mathbf{w}_{y_i,r}^{(t,b-1)}, \boldsymbol{\xi}_i\rangle - \frac{\eta}{Bm} \cdot \sum_{i' \in \mathcal{I}_{t,b-1}} \ell_{i'}^{(t,b-1)} \cdot \sigma'(\langle \mathbf{w}_{y_i,r}^{(t,b-1)}, y_{i'}\boldsymbol{\mu}\rangle) \cdot \langle y_{i'}\boldsymbol{\mu}, \boldsymbol{\xi}_i\rangle y_{i'}$$

$$- \frac{\eta(P-1)}{Bm} \cdot \sum_{i' \in \mathcal{I}_{t,b-1}/i} \ell_{i'}^{(t,b-1)} \cdot \sigma'(\langle \mathbf{w}_{y_i,r}^{(t,b-1)}, \boldsymbol{\xi}_{i'} \rangle) \cdot \langle \boldsymbol{\xi}_{i'}, \boldsymbol{\xi}_i \rangle.$$

Therefore,

$$\langle \mathbf{w}_{y_i,r}^{(t,b)}, \boldsymbol{\xi}_i \rangle \geq \langle \mathbf{w}_{y_i,r}^{(t,0)}, \boldsymbol{\xi}_i \rangle - \frac{\eta}{Bm} \cdot n \cdot \|\mu\|_2 \sigma_p \sqrt{2\log(6n/\delta)} - \frac{\eta(P-1)}{Bm} \cdot n \cdot 2\sigma_p^2 \sqrt{d\log(6n^2/\delta)}$$
$$\geq \langle \mathbf{w}_{y_i,r}^{(t,0)}, \boldsymbol{\xi}_i \rangle - \sigma_0 \sigma_p \sqrt{d}/\sqrt{2},$$

where the first inequality is due to Lemma B.1, and the second inequality is due to Condition 3.1.

For the third statement. Let $r^* \in S_i^{(t,0)}$, then

$$\langle \mathbf{w}_{y_i,r^*}^{(t,b)}, \boldsymbol{\xi}_i \rangle \geq \langle \mathbf{w}_{y_i,r^*}^{(t,0)}, \boldsymbol{\xi}_i \rangle - \sigma_0 \sigma_p \sqrt{d}/\sqrt{2} > 0,$$

where the first inequality is due to the second statement, and the second inequality is due to the definition of $\widetilde{S}_i^{t,0}$. Therefore, $r^* \in \widetilde{S}_i^{(t,b)}$ and $S_i^{(t,b)} \subseteq \widetilde{S}_i^{(t,b)}$. The fourth statement can be obtained similarly. $\qquad\square$

**Lemma C.8.** *Under Assumption 3.1, suppose* (27), (28) *and* (29) *hold for any iteration* $(t', b') \leq (t, 0)$. *Then, the following conditions also hold for* $\forall t' \leq t$ *and* $\forall b', b'_1, b'_2 \in [H]$:

1. $\sum_{r=1}^m \left[ \overline{\rho}_{y_i,r,i}^{(t',0)} - \overline{\rho}_{y_k,r,k}^{(t',0)} \right] \leq \kappa$ *for all* $i, k \in [n]$.

2. $y_i \cdot f(\mathbf{W}^{(t',b'_1)}, \mathbf{x}_i) - y_k \cdot f(\mathbf{W}^{(t',b'_2)}, \mathbf{x}_k) \leq C_1$ *for all* $i, k \in [n]$.

3. $\ell_i'^{(t',b'_1)} / \ell_k'^{(t',b'_2)} \leq C_2 = \exp(C_1)$ *for all* $i, k \in [n]$.

4. $S_i^{(0,0)} \subseteq S_i^{(t',0)}$, *where* $S_i^{(t',0)} := \{ r \in [m] : \langle \mathbf{w}_{y_i,r}^{(t',0)}, \boldsymbol{\xi}_i \rangle > \sigma_0 \sigma_p \sqrt{d}/\sqrt{2} \}$, *and hence* $|S_i^{(t',0)}| \geq 0.8m\Phi(-1)$ *for all* $i \in [n]$.

5. $S_{j,r}^{(0,0)} \subseteq S_{j,r}^{(t',0)}$, *where* $S_{j,r}^{(t',0)} := \{ i \in [n] : y_i = j, \langle \mathbf{w}_{j,r}^{(t',0)}, \boldsymbol{\xi}_i \rangle > \sigma_0 \sigma_p \sqrt{d}/\sqrt{2} \}$, *and hence* $|S_{j,r}^{(t',0)}| \geq \Phi(-1)n/4$ *for all* $j \in \{\pm 1\}, r \in [m]$.

*Here we take $\kappa$ and $C_1$ as $10$ and $5$ respectively.*

*Proof of Lemma C.8.* We prove Lemma C.8 by induction. When $t' = 0$, the fourth and fifth conditions hold naturally by Lemma B.3 and B.4.

For the first condition, since we have $\overline{\rho}_{j,r,i}^{(0,0)} = 0$ for any $j, r, i$ according to (26), it is straightforward that $\sum_{r=1}^m \left[ \overline{\rho}_{y_i,r,i}^{(0,0)} - \overline{\rho}_{y_k,r,k}^{(0,0)} \right] = 0$ for all $i, k \in [n]$. So the first condition holds for $t' = 0$.

For the second condition, we have

$$y_i \cdot f(\mathbf{W}^{(0,0)}, \mathbf{x}_i) - y_k \cdot f(\mathbf{W}^{(0,0)}, \mathbf{x}_k)$$
$$= F_{y_i}(\mathbf{W}_{y_i}^{(0,0)}, \mathbf{x}_i) - F_{-y_i}(\mathbf{W}_{-y_i}^{(0,0)}, \mathbf{x}_i) + F_{-y_k}(\mathbf{W}_{-y_k}^{(0,0)}, \mathbf{x}_i) - F_{y_k}(\mathbf{W}_{y_k}^{(0,0)}, \mathbf{x}_i)$$
$$\leq F_{y_i}(\mathbf{W}_{y_i}^{(0,0)}, \mathbf{x}_i) + F_{-y_k}(\mathbf{W}_{-y_k}^{(0,0)}, \mathbf{x}_i)$$
$$= \frac{1}{m} \sum_{r=1}^m [\sigma(\langle \mathbf{w}_{y_i,r}^{(0,0)}, y_i\boldsymbol{\mu} \rangle) + (P-1)\sigma(\langle \mathbf{w}_{y_i,r}^{(0,0)}, \boldsymbol{\xi}_i \rangle)]$$
$$+ \frac{1}{m} \sum_{r=1}^m [\sigma(\langle \mathbf{w}_{-y_k,r}^{(0,0)}, y_k\boldsymbol{\mu} \rangle) + (P-1)\sigma(\langle \mathbf{w}_{-y_k,r}^{(0,0)}, \boldsymbol{\xi}_i \rangle)]$$
$$\leq 4\beta \leq 1/3 \leq C_1,$$

where the first inequality is by $F_j(\mathbf{W}_j^{(0,0)}, \mathbf{x}_i) > 0$, the second inequality is due to (23), and the third inequality is due to (25).

By Lemma C.6 and the second condition, the third condition can be obtained directly as

$$\frac{\ell_i'^{(0,0)}}{\ell_k'^{(0,0)}} \le \exp\left(y_k \cdot f(\mathbf{W}^{(0,0)}, \mathbf{x}_k) - y_i \cdot f(\mathbf{W}^{(0,0)}, \mathbf{x}_i)\right) \le \exp(C_1).$$

Now suppose there exists $(\widetilde{t}, \widetilde{b}) \le (t, b)$ such that these five conditions hold for any $(0,0) \le (t', b') < (\widetilde{t}, \widetilde{b})$. We aim to prove that these conditions also hold for $(t', b') = (\widetilde{t}, \widetilde{b})$.

We first show that, for any $0 \le t' \le t$ and $0 \le b_1', b_2' \le b$, $y_i \cdot f(\mathbf{W}^{(t',b_1')}, \mathbf{x}_i) - y_k \cdot f(\mathbf{W}^{(t',b_2')}, \mathbf{x}_k)$ can be approximated by $\frac{1}{m}\sum_{r=1}^{m} \left[\overline{\rho}_{y_i,r,i}^{(t',b_1')} - \overline{\rho}_{y_k,r,k}^{(t',b_2')}\right]$ with a small constant approximation error. We begin by writing out

$$
\begin{aligned}
&y_i \cdot f(\mathbf{W}^{(t',b_1')}, \mathbf{x}_i) - y_k \cdot f(\mathbf{W}^{(t',b_2')}, \mathbf{x}_k) \\
&= y_i \sum_{j \in \{\pm 1\}} j \cdot F_j(\mathbf{W}_j^{(t',b_1')}, \mathbf{x}_i) - y_k \sum_{j \in \{\pm 1\}} j \cdot F_j(\mathbf{W}_j^{(t',b_2')}, \mathbf{x}_k) \\
&= F_{-y_k}(\mathbf{W}_{-y_k}^{(t',b_2')}, \mathbf{x}_k) - F_{-y_i}(\mathbf{W}_{-y_i}^{(t',b_1')}, \mathbf{x}_i) + F_{y_i}(\mathbf{W}_{y_i}^{(t',b_1')}, \mathbf{x}_i) - F_{y_k}(\mathbf{W}_{y_k}^{(t',b_2')}, \mathbf{x}_k) \\
&= F_{-y_k}(\mathbf{W}_{-y_k}^{(t',b_2')}, \mathbf{x}_k) - F_{-y_i}(\mathbf{W}_{-y_i}^{(t',b_1')}, \mathbf{x}_i) \\
&\quad + \frac{1}{m} \sum_{r=1}^{m} [\sigma(\langle \mathbf{w}_{y_i,r}^{(t',b_1')}, y_i \cdot \boldsymbol{\mu}\rangle) + (P-1)\sigma(\langle \mathbf{w}_{y_i,r}^{(t',b_1')}, \boldsymbol{\xi}_i\rangle)] \\
&\quad - \frac{1}{m} \sum_{r=1}^{m} [\sigma(\langle \mathbf{w}_{y_k,r}^{(t',b_2')}, y_k \cdot \boldsymbol{\mu}\rangle) + (P-1)\sigma(\langle \mathbf{w}_{y_k,r}^{(t',b_2')}, \boldsymbol{\xi}_k\rangle)] \\
&= \underbrace{F_{-y_k}(\mathbf{W}_{-y_k}^{(t',b_2')}, \mathbf{x}_k) - F_{-y_i}(\mathbf{W}_{-y_i}^{(t',b_1')}, \mathbf{x}_i)}_{I_1} \\
&\quad + \underbrace{\frac{1}{m} \sum_{r=1}^{m} [\sigma(\langle \mathbf{w}_{y_i,r}^{(t',b_1')}, y_i \cdot \boldsymbol{\mu}\rangle) - \sigma(\langle \mathbf{w}_{y_k,r}^{(t',b_2')}, y_k \cdot \boldsymbol{\mu}\rangle)]}_{I_2} \\
&\quad + \underbrace{\frac{1}{m} \sum_{r=1}^{m} [(P-1)\sigma(\langle \mathbf{w}_{y_i,r}^{(t',b_1')}, \boldsymbol{\xi}_i\rangle) - (P-1)\sigma(\langle \mathbf{w}_{y_k,r}^{(t',b_2')}, \boldsymbol{\xi}_k\rangle)]}_{I_3},
\end{aligned}
\tag{33}
$$

where all the equalities are due to the network definition. Then we bound $I_1$, $I_2$ and $I_3$.

For $|I_1|$, we have the following upper bound by Lemma C.4:

$$
\begin{aligned}
|I_1| &\le |F_{-y_k}(\mathbf{W}_{-y_k}^{(t',b_2')}, \mathbf{x}_k)| + |F_{-y_i}(\mathbf{W}_{-y_i}^{(t',b_1')}, \mathbf{x}_i)| \\
&= F_{-y_k}(\mathbf{W}_{-y_k}^{(t',b_2')}, \mathbf{x}_k) + F_{-y_i}(\mathbf{W}_{-y_i}^{(t',b_1')}, \mathbf{x}_i) \\
&\le 1.
\end{aligned}
\tag{34}
$$

For $|I_2|$, we have the following upper bound:

$$
\begin{aligned}
|I_2| &\le \max\left\{ \frac{1}{m}\sum_{r=1}^{m} \sigma(\langle \mathbf{w}_{y_i,r}^{(t',b_1')}, y_i \cdot \boldsymbol{\mu}\rangle), \frac{1}{m}\sum_{r=1}^{m} \sigma(\langle \mathbf{w}_{y_k,r}^{(t',b_2')}, y_k \cdot \boldsymbol{\mu}\rangle)\right\} \\
&\le 3\max\left\{ |\langle \mathbf{w}_{y_i,r}^{(0,0)}, y_i \cdot \boldsymbol{\mu}\rangle|, |\langle \mathbf{w}_{y_k,r}^{(0,0)}, y_k \cdot \boldsymbol{\mu}\rangle|, \gamma_{j,r}^{(t',b_1')}, \gamma_{j,r}^{(t',b_2')}, \mathrm{SNR}\sqrt{\frac{32\log(6n/\delta)}{d}}n\alpha\right\} \\
&\le 3\max\left\{ \beta, C'\widehat{\gamma}\alpha, \mathrm{SNR}\sqrt{\frac{32\log(6n/\delta)}{d}}n\alpha\right\} \\
&\le 0.25,
\end{aligned}
\tag{35}
$$

where the second inequality is due to (30), the second inequality is due to the definition of $\beta$ and (29), the third inequality is due to Condition 3.1 and (25).

For $I_3$, we have the following upper bound

$$I_3 = \frac{1}{m}\sum_{r=1}^{m}\left[(P-1)\sigma(\langle\mathbf{w}_{y_i,r}^{(t',b_1')},\boldsymbol{\xi}_i\rangle) - (P-1)\sigma(\langle\mathbf{w}_{y_k,r}^{(t',b_2')},\boldsymbol{\xi}_k\rangle)\right]$$

$$\leq \frac{1}{m}\sum_{r=1}^{m}\left[(P-1)\langle\mathbf{w}_{y_i,r}^{(t',b_1')},\boldsymbol{\xi}_i\rangle - (P-1)\langle\mathbf{w}_{y_k,r}^{(t',b_2')},\boldsymbol{\xi}_k\rangle\right] + 0.25$$

$$\leq \frac{1}{m}\sum_{r=1}^{m}\left[\overline{\rho}_{y_i,r,i}^{(t',b_1')} - \overline{\rho}_{y_k,r,k}^{(t',b_2')} + 10\sqrt{\frac{\log(6n^2/\delta)}{d}}n\alpha\right] + 0.25$$

$$\leq \frac{1}{m}\sum_{r=1}^{m}\left[\overline{\rho}_{y_i,r,i}^{(t',b_1')} - \overline{\rho}_{y_k,r,k}^{(t',b_2')}\right] + 0.5, \tag{36}$$

where the first inequality is due to Lemma C.5, the second inequality is due to Lemma C.3, the third inequality is due to $5\sqrt{\log(6n^2/\delta)/d}n\alpha \leq 1/8$ according to Condition 3.1.

Similarly, we have the following lower bound

$$I_3 = \frac{1}{m}\sum_{r=1}^{m}\left[(P-1)\sigma(\langle\mathbf{w}_{y_i,r}^{(t',b_1')},\boldsymbol{\xi}_i\rangle) - (P-1)\sigma(\langle\mathbf{w}_{y_k,r}^{(t',b_2')},\boldsymbol{\xi}_k\rangle)\right]$$

$$\geq \frac{1}{m}\sum_{r=1}^{m}\left[(P-1)\langle\mathbf{w}_{y_i,r}^{(t',b_1')},\boldsymbol{\xi}_i\rangle - (P-1)\langle\mathbf{w}_{y_k,r}^{(t',b_2')},\boldsymbol{\xi}_k\rangle\right] - 0.25$$

$$\geq \frac{1}{m}\sum_{r=1}^{m}\left[\overline{\rho}_{y_i,r,i}^{(t',b_1')} - \overline{\rho}_{y_k,r,k}^{(t',b_2')} - 10\sqrt{\frac{\log(6n^2/\delta)}{d}}n\alpha\right] - 0.25$$

$$\geq \frac{1}{m}\sum_{r=1}^{m}\left[\overline{\rho}_{y_i,r,i}^{(t',b_1')} - \overline{\rho}_{y_k,r,k}^{(t',b_2')}\right] - 0.5, \tag{37}$$

where the first inequality is due to Lemma C.5, the second inequality is due to Lemma C.3, the third inequality is due to $5\sqrt{\log(6n^2/\delta)/d}n\alpha \leq 1/8$ according to Condition 3.1.

By plugging (34)-(36) into (33), we have

$$y_i \cdot f(\mathbf{W}^{(t',b_1')},\mathbf{x}_i) - y_k \cdot f(\mathbf{W}^{(t',b_2')},\mathbf{x}_k) \leq |I_1| + |I_2| + I_3$$

$$\leq \frac{1}{m}\sum_{r=1}^{m}\left[\overline{\rho}_{y_i,r,i}^{(t',b_1')} - \overline{\rho}_{y_k,r,k}^{(t',b_2')}\right] + 1.75,$$

$$y_i \cdot f(\mathbf{W}^{(t',b_1')},\mathbf{x}_i) - y_k \cdot f(\mathbf{W}^{(t',b_2')},\mathbf{x}_k) \geq -|I_1| - |I_2| + I_3$$

$$\geq \frac{1}{m}\sum_{r=1}^{m}\left[\overline{\rho}_{y_i,r,i}^{(t',b_1')} - \overline{\rho}_{y_k,r,k}^{(t',b_2')}\right] - 1.75,$$

which is equivalent to

$$\left|y_i \cdot f(\mathbf{W}^{(t',b_1')},\mathbf{x}_i) - y_k \cdot f(\mathbf{W}^{(t',b_2')},\mathbf{x}_k) - \frac{1}{m}\sum_{r=1}^{m}\left[\overline{\rho}_{y_i,r,i}^{(t',b_1')} - \overline{\rho}_{y_k,r,k}^{(t',b_2')}\right]\right| \leq 1.75. \tag{38}$$

Therefore, the second condition immediately follows from the first condition.

Then, we prove the first condition holds for $(\widetilde{t},\widetilde{b})$. Recall from Lemma C.1 that

$$\overline{\rho}_{j,r,i}^{(t,b+1)} = \overline{\rho}_{j,r,i}^{(t,b)} - \frac{\eta(P-1)^2}{Bm}\cdot\ell_i'^{(t,b)}\cdot\sigma'(\langle\mathbf{w}_{j,r}^{(t,b)},\boldsymbol{\xi}_i\rangle)\cdot\|\boldsymbol{\xi}_i\|_2^2\cdot\mathbb{1}(y_i=j)\mathbb{1}(i\in\mathcal{I}_{t,b})$$

for all $j\in\{\pm1\}, r\in[m], i\in[n], (0,0)\leq(t,b)<[T^*,0]$. It follows that

$$\sum_{r=1}^{m}\left[\overline{\rho}_{y_i,r,i}^{(t,b+1)} - \overline{\rho}_{y_k,r,k}^{(t,b+1)}\right]$$

$$= \sum_{r=1}^{m} [\overline{\rho}_{y_i,r,i}^{(t,b)} - \overline{\rho}_{y_k,r,k}^{(t,b)}] - \frac{\eta(P-1)^2}{Bm} \cdot \left( |\widetilde{S}_i^{(t,b)}| \ell_i'^{(t,b)} \cdot \|\boldsymbol{\xi}_i\|_2^2 \, \mathbb{1}(i \in \mathcal{I}_{t,b}) \right.$$

$$\left. - |\widetilde{S}_k^{(t,b)}| \ell_k'^{(t,b)} \cdot \|\boldsymbol{\xi}_k\|_2^2 \, \mathbb{1}(k \in \mathcal{I}_{t,b}) \right),$$

for all $i, k \in [n]$ and $0 \le t \le T^*$, $b < H$.

If $\widetilde{b} \in \{1, 2, \cdots, H-1\}$, then the first statement for $(t', b') = (\widetilde{t}, \widetilde{b})$ and for the last $(t', b') < (\widetilde{t}, \widetilde{b})$ are the same. Otherwise, if $\widetilde{b} = 0$, we consider two separate cases: $\sum_{r=1}^{m} [\overline{\rho}_{y_i,r,i}^{(\widetilde{t}-1,0)} - \overline{\rho}_{y_k,r,k}^{(\widetilde{t}-1,0)}] \le 0.9\kappa$ and $\sum_{r=1}^{m} [\overline{\rho}_{y_i,r,i}^{(\widetilde{t}-1,0)} - \overline{\rho}_{y_k,r,k}^{(\widetilde{t}-1,0)}] > 0.9\kappa$.

When $\sum_{r=1}^{m} [\overline{\rho}_{y_i,r,i}^{(\widetilde{t}-1,0)} - \overline{\rho}_{y_k,r,k}^{(\widetilde{t}-1,0)}] \le 0.9\kappa$, we have

$$\sum_{r=1}^{m} [\overline{\rho}_{y_i,r,i}^{(\widetilde{t},0)} - \overline{\rho}_{y_k,r,k}^{(\widetilde{t},0)}]$$

$$= \sum_{r=1}^{m} [\overline{\rho}_{y_i,r,i}^{(\widetilde{t}-1,0)} - \overline{\rho}_{y_k,r,k}^{(\widetilde{t}-1,0)}] - \frac{\eta(P-1)^2}{Bm} \cdot \left( |\widetilde{S}_i^{(\widetilde{t}-1,b_i^{(\widetilde{t}-1)})}| \ell_i'^{(\widetilde{t}-1,b_i^{(\widetilde{t}-1)})} \cdot \|\boldsymbol{\xi}_i\|_2^2 \right.$$

$$\left. - |\widetilde{S}_k^{(\widetilde{t}-1,b_k^{(\widetilde{t}-1)})}| \ell_k'^{(\widetilde{t}-1,b_k^{(\widetilde{t}-1)})} \cdot \|\boldsymbol{\xi}_k\|_2^2 \right)$$

$$\le \sum_{r=1}^{m} [\overline{\rho}_{y_i,r,i}^{(\widetilde{t}-1,0)} - \overline{\rho}_{y_k,r,k}^{(\widetilde{t}-1,0)}] - \frac{\eta(P-1)^2}{Bm} \cdot |\widetilde{S}_i^{(\widetilde{t}-1,b_i^{(\widetilde{t}-1)})}| \ell_i'^{(\widetilde{t}-1,b_i^{(\widetilde{t}-1)})} \cdot \|\boldsymbol{\xi}_i\|_2^2$$

$$\le \sum_{r=1}^{m} [\overline{\rho}_{y_i,r,i}^{(\widetilde{t}-1,0)} - \overline{\rho}_{y_k,r,k}^{(\widetilde{t}-1,0)}] + \frac{\eta(P-1)^2}{B} \cdot \|\boldsymbol{\xi}_i\|_2^2$$

$$\le 0.9\kappa + 0.1\kappa$$

$$= \kappa,$$

where the first inequality is due to $\ell_i'^{(\widetilde{t}-1,b_i^{(\widetilde{t}-1)})} < 0$; the second inequality is due to $\left| S_i^{(\widetilde{t}-1,b_i^{(\widetilde{t}-1)})} \right| \le m$ and $-\ell_i'^{(\widetilde{t}-1,b_i^{(\widetilde{t}-1)})} < 1$; the third inequality is due to Condition 3.1.

On the other hand, for when $\sum_{r=1}^{m} [\overline{\rho}_{y_i,r,i}^{(\widetilde{t}-1,0)} - \overline{\rho}_{y_k,r,k}^{(\widetilde{t}-1,0)}] > 0.9\kappa$, we have from the (38) that

$$y_i \cdot f(\mathbf{W}^{(\widetilde{t}-1,b_i^{(\widetilde{t}-1)})}, \mathbf{x}_i) - y_k \cdot f(\mathbf{W}^{(\widetilde{t}-1,b_k^{(\widetilde{t}-1)})}, \mathbf{x}_k)$$

$$\ge \frac{1}{m} \sum_{r=1}^{m} [\overline{\rho}_{y_i,r,i}^{(\widetilde{t}-1,b_i^{(\widetilde{t}-1)})} - \overline{\rho}_{y_k,r,k}^{(\widetilde{t}-1,b_k^{(\widetilde{t}-1)})}] - 1.75$$

$$\ge \frac{1}{m} \sum_{r=1}^{m} [\overline{\rho}_{y_i,r,i}^{(\widetilde{t}-1,0)} - \overline{\rho}_{y_k,r,k}^{(\widetilde{t}-1,0)}] - 0.1\kappa - 1.75$$

$$\ge 0.9\kappa - 0.1\kappa - 0.54\kappa$$

$$= 0.26\kappa, \tag{39}$$

where the second inequality is due to $\kappa = 10$. Thus, according to Lemma C.6, we have

$$\frac{\ell_i'^{(\widetilde{t}-1,b_i^{(\widetilde{t}-1)})}}{\ell_k'^{(\widetilde{t}-1,b_k^{(\widetilde{t}-1)})}} \le \exp\left( y_k \cdot f(\mathbf{W}^{(\widetilde{t}-1,b_k^{(\widetilde{t}-1)})}, \mathbf{x}_k) - y_i \cdot f(\mathbf{W}^{(\widetilde{t}-1,b_i^{(\widetilde{t}-1)})}, \mathbf{x}_i) \right) \le \exp(-0.26\kappa).$$

Since $S_i^{(\widetilde{t}-1,0)} \subseteq \widetilde{S}_i^{(\widetilde{t}-1,b_i^{(\widetilde{t}-1)})}$, we have $\left| \widetilde{S}_k^{(\widetilde{t}-1,b_k^{(\widetilde{t}-1)})} \right| \ge 0.8\Phi(-1)m$ according to the fourth condition. Also we have that $|S_i^{(\widetilde{t}-1,b_i^{(\widetilde{t}-1)})}| \le m$. It follows that

$$\frac{|S_i^{(\widetilde{t}-1,b_i^{(\widetilde{t}-1)})}| \ell_i'^{(\widetilde{t}-1,b_i^{(\widetilde{t}-1)})}}{|S_k^{(\widetilde{t}-1,b_k^{(\widetilde{t}-1)})}| \ell_k'^{(\widetilde{t}-1,b_k^{(\widetilde{t}-1)})}} \le \frac{\exp(-0.26\kappa)}{0.8\Phi(-1)} < 0.8.$$

According to Lemma B.1, under event $\mathcal{E}_{\text{prelim}}$, we have

$$\left| \|\boldsymbol{\xi}_i\|_2^2 - d \cdot \sigma_p^2 \right| = O\big(\sigma_p^2 \cdot \sqrt{d \log(6n/\delta)}\big), \ \forall i \in [n].$$

Note that $d = \Omega(\log(6n/\delta))$ from Condition 3.1, it follows that

$$|S_i^{(\widetilde{t},b_i^{(\widetilde{t}-1)})}|(-\ell_i'^{(\widetilde{t},b_i^{(\widetilde{t}-1)})}) \cdot \|\boldsymbol{\xi}_i\|_2^2 < |S_k^{(\widetilde{t},b_k^{(\widetilde{t}-1)})}|(-\ell_k'^{(\widetilde{t},b_k^{(\widetilde{t}-1)})}) \cdot \|\boldsymbol{\xi}_k\|_2^2.$$

Then we have

$$\sum_{r=1}^{m} \left[\overline{\rho}_{y_i,r,i}^{(\widetilde{t},0)} - \overline{\rho}_{y_k,r,k}^{(\widetilde{t},0)}\right] \le \sum_{r=1}^{m} \left[\overline{\rho}_{y_i,r,i}^{(\widetilde{t}-1,0)} - \overline{\rho}_{y_k,r,k}^{(\widetilde{t}-1,0)}\right] \le \kappa,$$

which completes the proof of the first hypothesis at iteration $(t', b') = (\widetilde{t}, \widetilde{b})$. Next, by applying the approximation in (38), we are ready to verify the second hypothesis at iteration $(\widetilde{t}, \widetilde{b})$. In fact, for any $(t', b_1'), (t', b_2') \le (\widetilde{t}, \widetilde{b})$, we have

$$y_i \cdot f(\mathbf{W}^{(t',b_1')}, \mathbf{x}_i) - y_k \cdot f(\mathbf{W}^{(t',b_2')}, \mathbf{x}_k) \le \frac{1}{m} \sum_{r=1}^{m} \left[\overline{\rho}_{y_i,r,i}^{(t',b_1')} - \overline{\rho}_{y_k,r,k}^{(t',b_2')}\right] + 1.75$$

$$\le \frac{1}{m} \sum_{r=1}^{m} \left[\overline{\rho}_{y_i,r,i}^{(t',0)} - \overline{\rho}_{y_k,r,k}^{(t',0)}\right] + 0.1\kappa + 1.75$$

$$\le C_1,$$

where the first inequality is by (38); the last inequality is by induction hypothesis and taking $\kappa$ as 10 and $C_1$ as 5.

And the third hypothesis directly follows by noting that, for any $(t', b_1'), (t', b_2') \le (\widetilde{t}, \widetilde{b})$,

$$\frac{\ell_i'^{(t',b_1')}}{\ell_k'^{(t',b_2')}} \le \exp\big(y_k \cdot f(\mathbf{W}^{(t',b_1')}, \mathbf{x}_k) - y_i \cdot f(\mathbf{W}^{(t',b_2')}, \mathbf{x}_i)\big) \le \exp(C_1) = C_2.$$

For the fourth hypothesis, If $\widetilde{b} \in \{1, 2, \cdots, H-1\}$, then the first statement for $(t', b') = (\widetilde{t}, \widetilde{b})$ and for the last $(t', b') < (\widetilde{t}, \widetilde{b})$ are the same. Otherwise, if $\widetilde{b} = 0$, according to the gradient descent rule, we have

$$\langle \mathbf{w}_{y_i,r}^{(\widetilde{t},0)}, \boldsymbol{\xi}_i \rangle = \langle \mathbf{w}_{y_i,r}^{(\widetilde{t}-1,0)}, \boldsymbol{\xi}_i \rangle - \frac{\eta}{Bm} \cdot \sum_{b'=0}^{H-1} \sum_{i' \in \mathcal{I}_{\widetilde{t}-1,b'}} \ell_{i'}^{(\widetilde{t}-1,\widetilde{b})} \cdot \sigma'(\langle \mathbf{w}_{y_i,r}^{(\widetilde{t}-1,b')}, y_{i'}\boldsymbol{\mu} \rangle) \cdot \langle y_{i'}\boldsymbol{\mu}, \boldsymbol{\xi}_i \rangle y_{i'}$$

$$- \frac{\eta(P-1)}{Bm} \cdot \sum_{b'=0}^{H-1} \sum_{i' \in \mathcal{I}_{\widetilde{t}-1,b'}} \ell_{i'}^{(\widetilde{t}-1,b')} \cdot \sigma'(\langle \mathbf{w}_{y_i,r}^{(\widetilde{t}-1,\widetilde{b})}, \boldsymbol{\xi}_{i'} \rangle) \cdot \langle \boldsymbol{\xi}_{i'}, \boldsymbol{\xi}_i \rangle$$

$$= \langle \mathbf{w}_{y_i,r}^{(\widetilde{t}-1,0)}, \boldsymbol{\xi}_i \rangle - \frac{\eta}{Bm} \cdot \sum_{b'=0}^{H-1} \sum_{i' \in \mathcal{I}_{\widetilde{t}-1,b'}} \ell_{i'}^{(\widetilde{t}-1,b')} \cdot \sigma'(\langle \mathbf{w}_{y_i,r}^{(\widetilde{t}-1,b')}, \widehat{y}_{i'}\boldsymbol{\mu} \rangle) \cdot \langle y_{i'}\boldsymbol{\mu}, \boldsymbol{\xi}_i \rangle y_{i'}$$

$$- \frac{\eta(P-1)}{Bm} \cdot \ell_i^{(\widetilde{t}-1,b_i^{(\widetilde{t}-1)})} \cdot \sigma'(\langle \mathbf{w}_{y_i,r}^{(\widetilde{t}-1,\widetilde{b})}, \boldsymbol{\xi}_i \rangle) \cdot \|\boldsymbol{\xi}_i\|_2^2$$

$$- \frac{\eta(P-1)}{Bm} \cdot \sum_{b'=0}^{H-1} \sum_{i' \in \mathcal{I}_{\widetilde{t}-1,b'}} \ell_{i'}^{(\widetilde{t},b')} \cdot \sigma'(\langle \mathbf{w}_{y_i,r}^{(\widetilde{t},b')}, \boldsymbol{\xi}_{i'} \rangle) \cdot \langle \boldsymbol{\xi}_{i'}, \boldsymbol{\xi}_i \rangle \mathbb{1}(i' \ne i)$$

$$= \langle \mathbf{w}_{y_i,r}^{(\widetilde{t},0)}, \boldsymbol{\xi}_i \rangle - \frac{\eta(P-1)}{Bm} \cdot \underbrace{\ell_i^{(\widetilde{t}-1,b_i^{(\widetilde{t}-1)})} \cdot \|\boldsymbol{\xi}_i\|_2^2}_{I_4}$$

$$- \frac{\eta(P-1)}{Bm} \cdot \underbrace{\sum_{b'=0}^{H-1} \sum_{i' \in \mathcal{I}_{\widetilde{t}-1,b'}} \ell_{i'}^{(\widetilde{t}-1,b')} \cdot \sigma'(\langle \mathbf{w}_{y_i,r}^{(\widetilde{t}-1,b')}, \boldsymbol{\xi}_{i'} \rangle) \cdot \langle \boldsymbol{\xi}_{i'}, \boldsymbol{\xi}_i \rangle \mathbb{1}(i' \ne i)}_{I_5}$$

$$-\frac{\eta}{Bm}\cdot\sum_{\widetilde{b}=0}^{H-1}\sum_{i'\in\mathcal{I}_{\widetilde{t}-1,b'}}\underbrace{\ell_{i'}^{(\widetilde{t}-1,b')}\cdot\sigma'(\langle\mathbf{w}_{y_i,r}^{(\widetilde{t}-1,b')},y_{i'}\boldsymbol{\mu}\rangle)\cdot\langle y_{i'}\boldsymbol{\mu},\boldsymbol{\xi}_i\rangle y_{i'}}_{I_6},$$

for any $r\in S_i^{(\widetilde{t}-1,0)}$, where the last equality is by $\langle\mathbf{w}_{y_i,r}^{(\widetilde{t}-1,b_i^{(\widetilde{t}-1)})},\boldsymbol{\xi}_i\rangle>0$. Then we respectively estimate $I_4, I_5, I_6$. For $I_4$, according to Lemma B.1, we have

$$-I_4\geq|\ell_i^{(\widetilde{t}-1,b_i^{(\widetilde{t}-1)})}|\cdot\sigma_p^2 d/2.$$

For $I_5$, we have following upper bound

$$|I_5|\leq\sum_{b'=0}^{H-1}\sum_{i'\in\mathcal{I}_{\widetilde{t}-1,b'}}|\ell_{i'}^{(\widetilde{t}-1,b')}|\cdot\sigma'(\langle\mathbf{w}_{y_i,r}^{(\widetilde{t}-1,b')},\boldsymbol{\xi}_{i'}\rangle)\cdot|\langle\boldsymbol{\xi}_{i'},\boldsymbol{\xi}_i\rangle|\,\mathbb{1}(i'\neq i)$$

$$\leq\sum_{b'=0}^{H-1}\sum_{i'\in\mathcal{I}_{\widetilde{t}-1,b'}}|\ell_{i'}^{(\widetilde{t}-1,b')}|\cdot|\langle\boldsymbol{\xi}_{i'},\boldsymbol{\xi}_i\rangle|\,\mathbb{1}(i'\neq i)$$

$$\leq\sum_{b'=0}^{H-1}\sum_{i'\in\mathcal{I}_{\widetilde{t}-1,b'}}|\ell_{i'}^{(\widetilde{t}-1,b')}|\cdot 2\sigma_p^2\cdot\sqrt{d\log(6n^2/\delta)}$$

$$\leq nC_2|\ell_i^{(\widetilde{t}-1,b_i^{(\widetilde{t}-1)})}|\cdot 2\sigma_p^2\cdot\sqrt{d\log(6n^2/\delta)},$$

where the first inequality is due to triangle inequality, the second inequality is due to $\sigma'(z)\in\{0,1\}$, the third inequality is due to Lemma B.1, the forth inequality is due to the third hypothesis at epoch $\widetilde{t}-1$.

For $I_6$, we have following upper bound

$$|I_6|\leq\sum_{b'=0}^{H-1}\sum_{i'\in\mathcal{I}_{\widetilde{t}-1,b'}}|\ell_{i'}^{(\widetilde{t}-1,b')}|\cdot\sigma'(\langle\mathbf{w}_{y_i,r}^{(\widetilde{t}-1,b')},y_{i'}\boldsymbol{\mu}\rangle)\cdot|\langle y_{i'}\boldsymbol{\mu},\boldsymbol{\xi}_i\rangle|$$

$$\leq\sum_{b'=0}^{H-1}\sum_{i'\in\mathcal{I}_{\widetilde{t}-1,b'}}|\ell_{i'}^{(\widetilde{t}-1,b')}||\langle y_{i'}\boldsymbol{\mu},\boldsymbol{\xi}_i\rangle|$$

$$\leq nC_2|\ell_i^{(\widetilde{t}-1,b_i^{(\widetilde{t}-1)})}|\cdot\|\boldsymbol{\mu}\|_2\sigma_p\sqrt{2\log(6n/\delta)},$$

where the first inequality is by triangle inequality; the second inequality is due to $\sigma'(z)\in\{0,1\}$; the third inequality is by Lemma B.1; the last inequality is due to the third hypothesis at epoch $\widetilde{t}-1$.

Since $d\geq\max\{32C_2^2 n^2\cdot\log(6n^2/\delta),4C_2n\|\boldsymbol{\mu}\|\sigma_p^{-1}\sqrt{2\log(6n/\delta)}\}$, we have $-(P-1)I_4\geq\max\{(P-1)|I_5|/2,|I_6|/2\}$ and hence $-(P-1)I_4\geq(P-1)|I_5|+|I_6|$. It follows that

$$\langle\mathbf{w}_{y_i,r}^{(\widetilde{t},0)},\boldsymbol{\xi}_i\rangle\geq\langle\mathbf{w}_{y_i,r}^{(\widetilde{t}-1,0)},\boldsymbol{\xi}_i\rangle>\sigma_0\sigma_p\sqrt{d}/\sqrt{2},$$

for any $r\in S_i^{(\widetilde{t}-1,0)}$. Therefore, $S_i^{(0,0)}\subseteq S_i^{(\widetilde{t}-1,0)}\subseteq S_i^{(\widetilde{t},0)}$. And it directly follows by Lemma B.3 that $|S_i^{(\widetilde{t},0)}|\geq 0.8m\Phi(-1),\forall i\in[n]$.

For the fifth hypothesis, similar to the proof of the fourth hypothesis, we also have

$$\langle\mathbf{w}_{y_i,r}^{(\widetilde{t},0)},\boldsymbol{\xi}_i\rangle=\langle\mathbf{w}_{y_i,r}^{(\widetilde{t}-1,0)},\boldsymbol{\xi}_i\rangle-\frac{\eta(P-1)}{Bm}\cdot\underbrace{\ell_i^{(\widetilde{t}-1,b_i^{(t-1)})}\cdot\|\boldsymbol{\xi}_i\|_2^2}_{I_4}$$

$$-\frac{\eta(P-1)}{Bm}\cdot\underbrace{\sum_{b'=0}^{H-1}\sum_{i'\in\mathcal{I}_{\widetilde{t},b'}}\ell_{i'}^{(\widetilde{t}-1,b')}\cdot\sigma'(\langle\mathbf{w}_{y_i,r}^{(\widetilde{t}-1,b')},\boldsymbol{\xi}_{i'}\rangle)\cdot\langle\boldsymbol{\xi}_{i'},\boldsymbol{\xi}_i\rangle\,\mathbb{1}(i'\neq i)}_{I_5}$$

$$-\frac{\eta}{Bm}\cdot\sum_{b'=0}^{H-1}\sum_{i'\in\mathcal{I}_{\widetilde{t}-1,b'}}\ell_{i'}^{(\widetilde{t}-1,b')}\cdot\sigma'(\langle\mathbf{w}_{y_{i'},r}^{(\widetilde{t}-1,b')},y_{i'}\boldsymbol{\mu}\rangle)\cdot\langle y_{i'}\boldsymbol{\mu},\boldsymbol{\xi}_i\rangle y_{i'},$$

$$\underbrace{\phantom{-\frac{\eta}{Bm}\cdot\sum_{b'=0}^{H-1}\sum_{i'\in\mathcal{I}_{\widetilde{t}-1,b'}}\ell_{i'}^{(\widetilde{t}-1,b')}\cdot\sigma'(\langle\mathbf{w}_{y_{i'},r}^{(\widetilde{t}-1,b')},y_{i'}\boldsymbol{\mu}\rangle)\cdot\langle y_{i'}\boldsymbol{\mu},\boldsymbol{\xi}_i\rangle y_{i'}}}_{I_6}$$

for any $i\in S_{j,r}^{(\widetilde{t}-1,0)}$, where the equality holds due to $\langle\mathbf{w}_{j,r}^{(\widetilde{t}-1,b_i^{(\widetilde{t}-1)})},\boldsymbol{\xi}_i\rangle>0$ and $y_i=j$. By applying the same technique used in the proof of the fourth hypothesis, it follows that

$$\langle\mathbf{w}_{j,r}^{(\widetilde{t},0)},\boldsymbol{\xi}_i\rangle\geq\langle\mathbf{w}_{j,r}^{(\widetilde{t}-1,0)},\boldsymbol{\xi}_i\rangle>0,$$

for any $i\in S_{j,r}^{(\widetilde{t}-1,0)}$. Thus, we have $S_{j,r}^{(0,0)}\subseteq S_{j,r}^{(\widetilde{t}-1,0)}\subseteq S_{j,r}^{(\widetilde{t},0)}$. And it directly follows by Lemma B.4 that $|S_{j,r}^{(\widetilde{t},0)}|\geq n\Phi(-1)/4$.

$$\square$$

*Proof of Proposition C.2.* Our proof is based on induction. The results are obvious at iteration $(0,0)$ as all the coefficients are zero. Suppose that the results in Proposition C.2 hold for all iterations $(0,0)\leq(t,b)<(\widetilde{t},\widetilde{b})$. We aim to prove that they also hold for iteration $(\widetilde{t},\widetilde{b})$.

Firstly, We prove that (28) exists at iteration $(\widetilde{t},\widetilde{b})$, i.e., $\underline{\rho}_{j,r,i}^{(\widetilde{t},\widetilde{b})}\geq-\beta-10\sqrt{\log(6n^2/\delta)/d}\cdot n\alpha$ for any $r\in[m]$, $j\in\{\pm1\}$ and $i\in[n]$. Notice that $\underline{\rho}_{-j,r,i}^{(\widetilde{t},\widetilde{b})}=0$ for $j=y_i$, therefore we only need to consider the case that $j\neq y_i$. We also only need to consider the case of $\widetilde{b}=b_i^{(\widetilde{t})}+1$ since $\underline{\rho}_{-j,r,i}^{(\widetilde{t},\widetilde{b})}$ doesn't change in other cases according to (21).

When $\underline{\rho}_{j,r,t}^{(\widetilde{t},b_i^{(\widetilde{t})})}<-0.5\beta-5\sqrt{\log(6n^2/\delta)/d}\cdot n\alpha$, by (32) in Lemma C.3 we have that

$$(P-1)\langle\mathbf{w}_{j,r}^{(\widetilde{t},b_i^{(\widetilde{t})})},\boldsymbol{\xi}_i\rangle\leq\underline{\rho}_{-j,r,i}^{(\widetilde{t},b_i^{(\widetilde{t})})}+(P-1)\langle\mathbf{w}_{j,r}^{(0,0)},\boldsymbol{\xi}_i\rangle+5\sqrt{\frac{\log(6n^2/\delta)}{d}}n\alpha<0,$$

and thus

$$\underline{\rho}_{-j,r,i}^{(\widetilde{t},\widetilde{b})}=\underline{\rho}_{-j,r,i}^{(\widetilde{t},b_i^{(\widetilde{t})})}+\frac{\eta(P-1)^2}{Bm}\cdot\ell_i'^{(\widetilde{t},b_i^{(\widetilde{t})})}\cdot\sigma'(\langle\mathbf{w}_{j,r}^{(\widetilde{t},b_i^{(\widetilde{t})})},\boldsymbol{\xi}_i\rangle)\cdot\|\boldsymbol{\xi}_i\|_2^2.$$

$$=\underline{\rho}_{-j,r,i}^{(\widetilde{t},b_i^{(\widetilde{t})})}\geq-\beta-10\sqrt{\frac{\log(6n^2/\delta)}{d}}n\alpha,$$

where the last inequality is by induction hypothesis.

When $\underline{\rho}_{j,r,t}^{(\widetilde{t},b_i^{(\widetilde{t})})}\geq-0.5\beta-5\sqrt{\log(6n^2/\delta)/d}\cdot n\alpha$, we have

$$\underline{\rho}_{-j,r,i}^{(\widetilde{t},\widetilde{b})}=\underline{\rho}_{-j,r,i}^{(t,b_i^{(\widetilde{t})})}+\frac{\eta(P-1)^2}{Bm}\cdot\ell_i'^{(t,b_i^{(\widetilde{t})})}\cdot\sigma'(\langle\mathbf{w}_{j,r}^{(t,b_i^{(\widetilde{t})})},\boldsymbol{\xi}_i\rangle)\cdot\|\boldsymbol{\xi}_i\|_2^2$$

$$\geq-0.5\beta-5\sqrt{\frac{\log(6n^2/\delta)}{d}}n\alpha-\frac{\eta(P-1)^2\cdot3\sigma_p^2 d}{2Bm}$$

$$\geq-0.5\beta-10\sqrt{\frac{\log(6n^2/\delta)}{d}}n\alpha$$

$$\geq-\beta-10\sqrt{\frac{\log(6n^2/\delta)}{d}}n\alpha,$$

where the first inequality is by $\ell_i'^{(t,b_i^{(\widetilde{t})})}\in(-1,0)$ and $\|\boldsymbol{\xi}_i\|_2^2\leq(3/2)\sigma_p^2 d$ by Lemma B.1; the second inequality is due to $5\sqrt{\log(6n^2/\delta)/d}\cdot n\alpha\geq3\eta\sigma_p^2 d/(2Bm)$ by Condition 3.1.

Next we prove (27) holds for $(\widetilde{t},\widetilde{b})$. We only need to consider the case of $j=y_i$. Consider

$$|\ell_i'^{(\widetilde{t},\widetilde{b})}|=\frac{1}{1+\exp\{y_i\cdot[F_{+1}(\mathbf{W}_{+1}^{(\widetilde{t},\widetilde{b})},\mathbf{x}_i)-F_{-1}(\mathbf{W}_{-1}^{(\widetilde{t},\widetilde{b})},\mathbf{x}_i)]\}}$$

$$\leq\exp(-y_i\cdot[F_{+1}(\mathbf{W}_{+1}^{(\widetilde{t},\widetilde{b})},\mathbf{x}_i)-F_{-1}(\mathbf{W}_{-1}^{(\widetilde{t},\widetilde{b})},\mathbf{x}_i)])$$

$$\leq\exp(-F_{y_i}(\mathbf{W}_{y_i}^{(\widetilde{t},\widetilde{b})},\mathbf{x}_i)+0.5),$$

(40)

where the last inequality is by $F_j(\mathbf{W}_j^{(\widetilde{t},\widetilde{b})}, \mathbf{x}_i) \leq 0.5$ for $j \neq y_i$ according to Lemma C.4. Now recall the iterative update rule of $\overline{\rho}_{j,r,i}^{(t,b)}$:

$$\overline{\rho}_{j,r,i}^{(t,b+1)} = \overline{\rho}_{j,r,i}^{(t,b)} - \frac{\eta(P-1)^2}{Bm} \cdot \ell_i'^{(t,b)} \cdot \sigma'(\langle \mathbf{w}_{j,r}^{(t,b)}, \boldsymbol{\xi}_i \rangle) \cdot \|\boldsymbol{\xi}_i\|_2^2 \cdot \mathbb{1}(i \in \mathcal{I}_{t,b}).$$

Let $(t_{j,r,i}, b_{j,r,i})$ be the last time before $(\widetilde{t}, \widetilde{b})$ that $\overline{\rho}_{j,r,i}^{(t_{j,r,i}, b_{j,r,i})} \leq 0.5\alpha$. Then by iterating the update rule from $(t_{j,r,i}, b_{j,r,i})$ to $(\widetilde{t}, \widetilde{b})$, we get

$$\overline{\rho}_{j,r,i}^{(\widetilde{t},\widetilde{b})}$$
$$= \overline{\rho}_{j,r,i}^{(t_{j,r,i}, b_{j,r,i})} \underbrace{- \frac{\eta(P-1)^2}{Bm} \cdot \ell_i'^{(t_{j,r,i}, b_{j,r,i})} \cdot \mathbb{1}(\langle \mathbf{w}_{j,r}^{(t_{j,r,i}, b_{j,r,i})}, \boldsymbol{\xi}_i \rangle \geq 0) \cdot \mathbb{1}(i \in \mathcal{I}_{t,b}) \|\boldsymbol{\xi}_i\|_2^2}_{I_7} \tag{41}$$
$$\underbrace{- \sum_{(t_{j,r,i}, b_{j,r,i}) < (t,b) < (\widetilde{t},\widetilde{b})} \frac{\eta(P-1)^2}{Bm} \cdot \ell_i'^{(t,b)} \cdot \mathbb{1}(\langle \mathbf{w}_{j,r}^{(t,b)}, \boldsymbol{\xi}_i \rangle \geq 0) \cdot \mathbb{1}(i \in \mathcal{I}_{t,b}) \|\boldsymbol{\xi}_i\|_2^2}_{I_8}.$$

We first bound $I_7$ as follows:

$$|I_7| \leq (\eta(P-1)^2/Bm) \cdot \|\boldsymbol{\xi}_i\|_2^2 \leq (\eta(P-1)^2/Bm) \cdot 3\sigma_p^2 d/2 \leq 1 \leq 0.25\alpha,$$

where the first inequality is by $\ell_i'^{(t_{j,r,i}, b_{j,r,i})} \in (-1, 0)$; the second inequality is by Lemma B.1; the third inequality is by Condition 3.1; the last inequality is by our choice of $\alpha = 4\log(T^*)$ and $T^* \geq e$.

Second, we bound $I_8$. For $(t_{j,r,i}, b_{j,r,i}) < (t,b) < (\widetilde{t}, \widetilde{b})$ and $y_i = j$, we can lower bound the inner product $\langle \mathbf{w}_{j,r}^{(t,b)}, \boldsymbol{\xi}_i \rangle$ as follows

$$\begin{aligned}
\langle \mathbf{w}_{j,r}^{(t,b)}, \boldsymbol{\xi}_i \rangle &\geq \langle \mathbf{w}_{j,r}^{(0,0)}, \boldsymbol{\xi}_i \rangle + \frac{1}{P-1} \overline{\rho}_{j,r,i}^{(t,b)} - \frac{5}{P-1} \sqrt{\frac{\log(6n^2/\delta)}{d}} n\alpha \\
&\geq -\frac{0.5}{P-1}\beta + \frac{0.5}{P-1}\alpha - \frac{5}{P-1} \sqrt{\frac{\log(6n^2/\delta)}{d}} n\alpha \\
&\geq \frac{0.25}{P-1}\alpha,
\end{aligned} \tag{42}$$

where the first inequality is by (31) in Lemma C.3; the second inequality is by $\overline{\rho}_{j,r,i}^{(t,b)} > 0.5\alpha$ and $\langle \mathbf{w}_{j,r}^{(0,0)}, \boldsymbol{\xi}_i \rangle \geq -0.5\beta/(P-1)$ due to the definition of $t_{j,r,i}$ and $\beta$; the last inequality is by $\beta \leq 1/8 \leq 0.1\alpha$ and $5\sqrt{\log(6n^2/\delta)/d} \cdot n\alpha \leq 0.2\alpha$ by Condition 3.1.

Thus, plugging the lower bounds of $\langle \mathbf{w}_{j,r}^{(t,b)}, \boldsymbol{\xi}_i \rangle$ into $I_8$ gives

$$\begin{aligned}
|I_8| &\leq \sum_{(t_{j,r,i}, b_{j,r,i}) < (t,b) < (\widetilde{t},\widetilde{b})} \frac{\eta(P-1)^2}{Bm} \cdot \exp\left(-\frac{1}{m}\sum_{r=1}^{m}(P-1)\sigma(\langle \mathbf{w}_{j,r}^{(t,b)}, \boldsymbol{\xi}_i \rangle) + 0.5\right) \\
&\qquad\qquad\qquad\qquad \cdot \mathbb{1}(\langle \mathbf{w}_{j,r}^{(t,b)}, \boldsymbol{\xi}_i \rangle \geq 0) \cdot \|\boldsymbol{\xi}_i\|_2^2 \\
&\leq \frac{2\eta T^* n(P-1)^2}{Bm} \cdot \exp(-0.25\alpha)\exp(0.5) \cdot \frac{3\sigma_p^2 d}{2} \\
&\leq \frac{2\eta T^* n(P-1)^2}{Bm} \cdot \exp(-\log(T^*))\exp(0.5) \cdot \frac{3\sigma_p^2 d}{2} \\
&= \frac{2\eta n(P-1)^2}{Bm} \cdot \frac{3\sigma_p^2 d}{2} \exp(0.5) \leq 1 \leq 0.25\alpha,
\end{aligned}$$

where the first inequality is by (40); the second inequality is by (42); the third inequality is by $\alpha = 4\log(T^*)$; the fourth inequality is by Condition 3.1; the last inequality is by $\log(T^*) \geq 1$ and $\alpha = 4\log(T^*)$. Plugging the bound of $I_7, I_8$ into (41) completes the proof for $\overline{\rho}$.

For the upper bound of (29), we prove a augmented hypothesis that there exists a $i^* \in [n]$ with $y_{i^*} = j$ such that for $1 \le t \le T^*$ we have that $\gamma_{j,r}^{(t,0)}/\overline{\rho}_{j,r,i^*} \le C'\widehat{\gamma}$. Recall the iterative update rule of $\gamma_{j,r}^{(t,b)}$ and $\overline{\rho}_{j,r,i}^{(t,b)}$, we have

$$\overline{\rho}_{j,r,i}^{(t,b+1)} = \overline{\rho}_{j,r,i}^{(t,b)} - \frac{\eta(P-1)^2}{Bm} \cdot \ell_i'^{(t,b)} \cdot \sigma'(\langle \mathbf{w}_{j,r}^{(t,b)}, \boldsymbol{\xi}_i \rangle) \cdot \|\boldsymbol{\xi}_i\|_2^2 \cdot \mathbb{1}(y_i = j)\,\mathbb{1}(i \in \mathcal{I}_{t,b}),$$

$$\gamma_{j,r}^{(t,b+1)} = \gamma_{j,r}^{(t,b)} - \frac{\eta}{Bm} \cdot \Bigg[ \sum_{i \in \mathcal{I}_{t,b} \cap S_+} \ell_i'^{(t,b)} \sigma'(\langle \mathbf{w}_{j,r}^{(t,b)}, y_i \cdot \boldsymbol{\mu} \rangle)$$

$$- \sum_{i \in \mathcal{I}_{t,b} \cap S_-} \ell_i'^{(t,b)} \sigma'(\langle \mathbf{w}_{j,r}^{(t,b)}, y_i \cdot \boldsymbol{\mu} \rangle) \Bigg] \cdot \|\boldsymbol{\mu}\|_2^2.$$

According to the fifth statement of Lemma C.8, for any $i^* \in S_{j,r}^{(0,0)}$ it holds that $j = y_{i^*}$ and $\langle \mathbf{w}_{j,r}^{(t,b)}, \boldsymbol{\xi}_{i^*} \rangle \ge 0$ for any $(t, b) \le (\widetilde{t}, \widetilde{b})$. Thus, we have

$$\overline{\rho}_{j,r,i^*}^{(\widetilde{t},0)} = \overline{\rho}_{j,r,i^*}^{(\widetilde{t}-1,0)} - \frac{\eta(P-1)^2}{Bm} \cdot \ell_{i^*}'^{(\widetilde{t}-1,b_{i^*}^{(\widetilde{t}-1)})} \cdot \|\boldsymbol{\xi}_{i^*}\|_2^2 \ge \overline{\rho}_{j,r,i^*}^{(\widetilde{t}-1,0)} - \frac{\eta(P-1)^2}{Bm} \cdot \ell_{i^*}'^{(\widetilde{t}-1,b_{i^*}^{(\widetilde{t}-1)})} \cdot \sigma_p^2 d/2.$$

For the update rule of $\gamma_{j,r}^{(t,b)}$, according to Lemma C.8, we have

$$\sum_{b<H} \Bigg| \sum_{i \in \mathcal{I}_{t,b} \cap S_+} \ell_i'^{(t,b)} \sigma'(\langle \mathbf{w}_{j,r}^{(t,b)}, y_i \cdot \boldsymbol{\mu} \rangle) - \sum_{i \in \mathcal{I}_{t,b} \cap S_-} \ell_i'^{(t,b)} \sigma'(\langle \mathbf{w}_{j,r}^{(t,b)}, y_i \cdot \boldsymbol{\mu} \rangle) \Bigg|$$

$$\le C_2 n \Big| \ell_{i^*}'^{(\widetilde{T}-1,b_{i^*}^{(\widetilde{T}-1)})} \Big|.$$

Then, we have

$$\frac{\gamma_{j,r}^{(\widetilde{t},0)}}{\overline{\rho}_{j,r,i^*}^{(\widetilde{t},0)}} \le \max\left\{ \frac{\gamma_{j,r}^{(\widetilde{t}-1,0)}}{\overline{\rho}_{j,r,i^*}^{(\widetilde{t}-1,0)}}, \frac{C_2 n \ell_{i^*}'^{(\widetilde{t}-1,b_{i^*}^{(\widetilde{t}-1)})} \|\boldsymbol{\mu}\|_2^2}{(P-1)^2 \cdot \ell_{i^*}'^{(\widetilde{t}-1,b_{i^*}^{(\widetilde{t}-1)})} \cdot \sigma_p^2 d/2} \right\}$$

$$= \max\left\{ \frac{\gamma_{j,r}^{(\widetilde{t}-1,0)}}{\overline{\rho}_{j,r,i^*}^{(\widetilde{t}-1,0)}}, \frac{2C_2 n \|\boldsymbol{\mu}\|_2^2}{(P-1)^2 \sigma_p^2 d} \right\} \tag{43}$$

$$\le \frac{2C_2 n \|\boldsymbol{\mu}\|_2^2}{(P-1)^2 \sigma_p^2 d},$$

where the last inequality is by $\gamma_{j,r}^{(\widetilde{t}-1,0)}/\overline{\rho}_{j,r,i^*}^{(\widetilde{t}-1,0)} \le 2C_2\widehat{\gamma} = 2C_2 n\|\boldsymbol{\mu}\|_2^2/(P-1)^2\sigma_p^2 d$. Therefore,

$$\frac{\gamma_{j,r}^{(\widetilde{t},0)}}{\overline{\rho}_{j,r,i^*}^{(\widetilde{t},0)}} \le 2C_2\widehat{\gamma}.$$

For iterations other than the starting of en epoch, we have the following upper bound:

$$\frac{\gamma_{j,r}^{(\widetilde{t},b)}}{\overline{\rho}_{j,r,i^*}^{(\widetilde{t},b)}} \le \frac{2\gamma_{j,r}^{(\widetilde{t},0)}}{\overline{\rho}_{j,r,i^*}^{(\widetilde{t},0)}} \le 4C_2\widehat{\gamma}.$$

Thus, by taking $C' = 4C_2$, we have $\gamma_{j,r}^{(\widetilde{t},b)}/\overline{\rho}_{j,r,i^*}^{(\widetilde{t},b)} \le C'\widehat{\gamma}$.

On the other hand, when $(t, b) < (\frac{\log(2T^*/\delta)}{2c_3^2}, 0)$, we have

$$\gamma_{j,r}^{(t,b)} \ge -\frac{\log(2T^*/\delta)}{2c_3^2} \cdot \frac{\eta}{Bm} \cdot n \cdot \|\boldsymbol{\mu}\|_2^2 \ge -\frac{1}{12},$$

where the first inequality is due to update rule of $\gamma_{j,r}^{t,b}$, and the second inequality is due to Condition 3.1.

When $(t,b) \geq (\frac{\log(2T^*/\delta)}{2c_3^2}, 0)$, According to Lemma B.6, we have

$$\gamma_{j,r}^{(t,b)} \geq \sum_{(t',b') < (t,b)} \frac{\eta}{Bm} \Big[ \min_{i,b'} \ell_i'^{(t',b')} \min\{|\mathcal{I}_{t',b'} \cap S_+ \cap S_{-1}|, |\mathcal{I}_{t',b'} \cap S_+ \cap S_1|\}$$

$$- \max_{i,b'} \ell_i'^{(t',b')} |\mathcal{I}_{t',b'} \cap S_-|\Big] \cdot \|\boldsymbol{\mu}\|_2^2$$

$$\geq \frac{\eta}{Bm} \Big( \sum_{t'=0}^{t-1} (c_3 c_4 H \frac{B}{4} \min_{i,b'} \ell_i'^{(t',b')} - nq \max_{i,b'} \ell_i'^{(t',b')}) - nq \max_{i,b'} \ell_i'^{(t,b')})\Big) \|\boldsymbol{\mu}\|_2^2$$

$$\geq 0,$$

where the first inequality is due to the update rule of $\gamma_{j,r}^{(t,b)}$, the second inequality is due to Lemma B.6, and the third inequality is due to Condition 3.1. $\qquad\square$

## C.2 Decoupling with a Two-stage Analysis

### C.2.1 First Stage

**Lemma C.9.** *There exist*

$$T_1 = C_3 \eta^{-1} Bm(P-1)^{-2} \sigma_p^{-2} d^{-1}, T_2 = C_4 \eta^{-1} Bm(P-1)^{-2} \sigma_p^{-2} d^{-1},$$

*where $C_3 = \Theta(1)$ is a large constant and $C_4 = \Theta(1)$ is a small constant, such that*

- $\overline{\rho}_{j,r^*,i}^{(T_1,0)} \geq 2$ *for any* $r^* \in S_i^{(0,0)} = \{r \in [m] : \langle \mathbf{w}_{y_i,r}^{(0)}, \boldsymbol{\xi}_i \rangle > 0\}$, $j \in \{\pm 1\}$ *and* $i \in [n]$ *with* $y_i = j$.

- $\max_{j,r} \gamma_{j,r}^{(t,b)} = O(\widehat{\gamma})$ *for all* $(t,b) \leq (T_1, 0)$.

- $\max_{j,r,i} |\underline{\rho}_{j,r,i}^{(t,b)}| = \max\{\beta, O(n\sqrt{\log(n/\delta)} \log(T^*)/\sqrt{d})\}$ *for all* $(t,b) \leq (T_1, 0)$.

- $\min_{j,r} \gamma_{j,r}^{(t,0)} = \Omega(\widehat{\gamma})$ *for all* $t \geq T_2$.

- $\max_{j,r} \overline{\rho}_{j,r,i}^{(T_1,0)} = O(1)$ *for all* $i \in [n]$.

*Proof of Lemma C.9.* By Proposition C.2, we have that $\underline{\rho}_{j,r,i}^{(t,b)} \geq -\beta - 10n\sqrt{\frac{\log(6n^2/\delta)}{d}}\alpha$ for all $j \in \{\pm 1\}$, $r \in [m]$, $i \in [n]$ and $(0,0) \leq (t,b) \leq (T^*, 0)$. According to Lemma B.2, for $\beta$ we have

$$\beta = 2 \max_{i,j,r}\{|\langle \mathbf{w}_{j,r}^{(0,0)}, \boldsymbol{\mu} \rangle|, (P-1)|\langle \mathbf{w}_{j,r}^{(0,0)}, \boldsymbol{\xi}_i \rangle|\}$$

$$\leq 2 \max\{\sqrt{2\log(12m/\delta)} \cdot \sigma_0 \|\boldsymbol{\mu}\|_2, 2\sqrt{\log(12mn/\delta)} \cdot \sigma_0 (P-1)\sigma_p\sqrt{d}\}$$

$$= O\big(\sqrt{\log(mn/\delta)} \cdot \sigma_0 (P-1)\sigma_p\sqrt{d}\big),$$

where the last equality is by the first condition of Condition 3.1. Since $\underline{\rho}_{j,r,i}^{(t,b)} \leq 0$, we have that

$$\max_{j,r,i} |\underline{\rho}_{j,r,i}^{(t,b)}| = \max_{j,r,i} -\underline{\rho}_{j,r,i}^{(t,b)}$$

$$\leq \beta + 10\sqrt{\frac{\log(6n^2/\delta)}{d}}n\alpha$$

$$= \max\left\{\beta, O\big(\sqrt{\log(n/\delta)} \log(T^*) \cdot n/\sqrt{d}\big)\right\}.$$

Next, for the growth of $\gamma_{j,r}^{(t)}$, we have following upper bound

$$\gamma_{j,r}^{(t,b+1)} = \gamma_{j,r}^{(t,b)} - \frac{\eta}{Bm} \cdot \sum_{i \in \mathcal{I}_{t,b}} \ell_i'^{(t,b)} \sigma'(\langle \mathbf{w}_{j,r}^{(t,b)}, y_i \cdot \boldsymbol{\mu} \rangle) \cdot \|\boldsymbol{\mu}\|_2^2$$

$$\leq \gamma_{j,r}^{(t,b)} + \frac{\eta}{m} \cdot \|\boldsymbol{\mu}\|_2^2,$$

where the inequality is by $|\ell'| \leq 1$. Note that $\gamma_{j,r}^{(0,0)} = 0$ and recursively use the inequality $tB + b$ times we have

$$\gamma_{j,r}^{(t,b)} \leq \frac{\eta(tH + b)}{m} \cdot \|\boldsymbol{\mu}\|_2^2. \tag{44}$$

Since $n \cdot \mathrm{SNR}^2 = n\|\boldsymbol{\mu}\|_2^2/((P-1)^2\sigma_p^2 d) = \widehat{\gamma}$, we have

$$T_1 = C_3\eta^{-1}Bm(P-1)^{-2}\sigma_p^{-2}d^{-1} = C_3\eta^{-1}m\|\boldsymbol{\mu}\|_2^{-2}\widehat{\gamma}B/n.$$

And it follows that

$$\gamma_{j,r}^{(t)} \leq \frac{\eta(tH + b)}{m} \cdot \|\boldsymbol{\mu}\|_2^2 \leq \frac{\eta nT_1}{mB} \cdot \|\boldsymbol{\mu}\|_2^2 \leq C_3\widehat{\gamma},$$

for all $(0,0) \leq (t,b) \leq (T_1, 0)$.

For $\overline{\rho}_{j,r,i}^{(t)}$, recall from (20) that

$$\overline{\rho}_{y_i,r,i}^{(t+1,0)} = \overline{\rho}_{y_i,r,i}^{(t,0)} - \frac{\eta(P-1)^2}{Bm} \cdot \ell_i'^{(t,b_i^{(t)})} \cdot \sigma'(\langle \mathbf{w}_{y_i,r}^{(t,b_i^{(t)})}, \boldsymbol{\xi}_i\rangle) \cdot \|\boldsymbol{\xi}_i\|_2^2.$$

According to Lemma C.8, for any $r^* \in S_i^{(0,0)} = \{r \in [m] : \langle \mathbf{w}_{y_i,r}^{(0)}, \boldsymbol{\xi}_i\rangle > \sigma_0\sigma_p\sqrt{d}/\sqrt{2}\}$, we have $\langle \mathbf{w}_{y_i,r^*}^{(t,b)}, \boldsymbol{\xi}_i\rangle > 0$ for all $(0,0) \leq (t,b) \leq (T^*, 0)$ and hence

$$\overline{\rho}_{j,r^*,i}^{(t+1,0)} = \overline{\rho}_{j,r^*,i}^{(t,0)} - \frac{\eta(P-1)^2}{Bm} \cdot \ell_i'^{(t,b_i^{(t)})}\|\boldsymbol{\xi}_i\|_2^2.$$

For each $i$, we denote by $T_1^{(i)}$ the last time in the period $[0, T_1]$ satisfying that $\overline{\rho}_{y_i,r^*,i}^{(t,0)} \leq 2$. Then for $(0,0) \leq (t,b) < (T_1^{(i)}, 0)$, $\max_{j,r}\{|\overline{\rho}_{j,r,i}^{(t,b)}|, |\underline{\rho}_{j,r,i}^{(t,b)}|\} = O(1)$ and $\max_{j,r}\gamma_{j,r}^{(t,b)} = O(1)$. Therefore, we know that $F_{-1}(\mathbf{W}^{(t,b)}, \mathbf{x}_i), F_{+1}(\mathbf{W}^{(t,b)}, \mathbf{x}_i) = O(1)$. Thus there exists a positive constant $C$ such that $-\ell_i'^{(t,b)} \geq C \geq C_2$ for $0 \leq t \leq T_1^{(i)}$.

Then we have

$$\overline{\rho}_{y_i,r^*,i}^{(t,0)} \geq \frac{C\eta(P-1)^2\sigma_p^2 dt}{2Bm}.$$

Therefore, $\overline{\rho}_{y_i,r^*,i}^{(t,0)}$ will reach 2 within

$$T_1 = C_3\eta^{-1}Bm(P-1)^2\sigma_p^{-2}d^{-1}$$

iterations for any $r^* \in S_i^{(0,0)}$, where $C_3$ can be taken as $4/C$.

Next, we will discuss the lower bound of the growth of $\gamma_{j,r}^{(t,b)}$. For $\overline{\rho}_{j,r,i}^{(t,b)}$, we have

$$\overline{\rho}_{j,r,i}^{(t,b+1)} = \overline{\rho}_{j,r,i}^{(t,b)} - \frac{\eta(P-1)^2}{Bm} \cdot \ell_i'^{(t,b)} \cdot \sigma'(\langle \mathbf{w}_{j,r}^{(t,b)}, \boldsymbol{\xi}_i\rangle) \cdot \|\boldsymbol{\xi}_i\|_2^2 \cdot \mathbb{1}(y_i = j)\,\mathbb{1}(i \in \mathcal{I}_{t,b})$$

$$\leq \overline{\rho}_{j,r,i}^{(t,b)} + \frac{3\eta(P-1)^2\sigma_p^2 d}{2Bm}.$$

According to (44) and $\overline{\rho}_{j,r,i}^{(0,0)} = 0$, it follows that

$$\overline{\rho}_{j,r,i}^{(t,b)} \leq \frac{3\eta(P-1)^2\sigma_p^2 d(tH + b)}{2Bm}, \gamma_{j,r}^{(t,b)} \leq \frac{\eta(tH + b)}{m} \cdot \|\boldsymbol{\mu}\|_2^2. \tag{45}$$

Therefore, $\max_{j,r,i}\overline{\rho}_{j,r,i}^{(t,b)}$ will be smaller than 1 and $\gamma_{j,r}^{(t,b)}$ smaller than $\Theta(n\|\boldsymbol{\mu}\|_2^2/(P-1)^2\sigma_p^2 d) = \Theta(n \cdot \mathrm{SNR}^2) = \Theta(\widehat{\gamma}) = O(1)$ within

$$T_2 = C_4\eta^{-1}Bm(P-1)^{-2}\sigma_p^{-2}d^{-1}$$

iterations, where $C_4$ can be taken as $2/3$. Therefore, we know that $F_{-1}(\mathbf{W}^{(t,b)}, \mathbf{x}_i), F_{+1}(\mathbf{W}^{(t,b)}, \mathbf{x}_i) = O(1)$ in $(0,0) \leq (t,b) \leq (T_2, 0)$. Thus, there exists a positive constant $C$ such that $-\ell_i'^{(t,b)} \geq C$ for $0 \leq t \leq T_2$.

Recall that we denote $\{i \in [n]|y_i = y\}$ as $S_y$, and we have the update rule

$$\gamma_{j,r}^{(t,b+1)} = \gamma_{j,r}^{(t,b)} - \frac{\eta}{Bm} \cdot \left[ \sum_{i \in \mathcal{I}_{t,b} \cap S_+} \ell_i'^{(t,b)} \sigma'(\langle \mathbf{w}_{j,r}^{(t,b)}, \widehat{y}_i \cdot \boldsymbol{\mu} \rangle) \right.$$
$$\left. - \sum_{i \in \mathcal{I}_{t,b} \cap S_-} \ell_i'^{(t,b)} \sigma'(\langle \mathbf{w}_{j,r}^{(t,b)}, \widehat{y}_i \cdot \boldsymbol{\mu} \rangle) \right] \cdot \|\boldsymbol{\mu}\|_2^2.$$

For the growth of $\gamma_{j,r}^{(t,b)}$, if $\langle \mathbf{w}_{j,r}^{(t,b)}, \boldsymbol{\mu} \rangle \geq 0$, we have

$$\gamma_{j,r}^{(t,b+1)} = \gamma_{j,r}^{(t,b)} - \frac{\eta}{Bm} \cdot \left[ \sum_{i \in \mathcal{I}_{t,b} \cap S_+ \cap S_1} \ell_i'^{(t)} - \sum_{i \in \mathcal{I}_{t,b} \cap S_- \cap S_1} \ell_i'^{(t)} \right] \|\boldsymbol{\mu}\|_2^2$$
$$\geq \gamma_{j,r}^{(t,b)} + \frac{\eta}{Bm} \cdot \left[ C|\mathcal{I}_{t,b} \cap S_+ \cap S_1| - |\mathcal{I}_{t,b} \cap S_- \cap S_{-1}| \right] \cdot \|\boldsymbol{\mu}\|_2^2.$$

Similarly, if $\langle \mathbf{w}_{j,r}^{(t,b)}, \boldsymbol{\mu} \rangle < 0$,

$$\gamma_{j,r}^{(t,b+1)} \geq \gamma_{j,r}^{(t,b)} + \frac{\eta}{Bm} \cdot \left[ C|\mathcal{I}_{t,b} \cap S_+ \cap S_{-1}| - |\mathcal{I}_{t,b} \cap S_- \cap S_1| \right] \cdot \|\boldsymbol{\mu}\|_2^2.$$

Therefore, for $t \in [T_2, T_1]$, we have

$$\gamma_{j,r}^{(t,0)} \geq \sum_{(t',b')<(t,0)} \frac{\eta}{Bm} \left[ C \min\{|\mathcal{I}_{t',b'} \cap S_+ \cap S_{-1}|, |\mathcal{I}_{t',b'} \cap S_+ \cap S_1|\} - |\mathcal{I}_{t,b} \cap S_-| \right] \cdot \|\boldsymbol{\mu}\|_2^2$$
$$\geq \frac{\eta}{Bm} (c_3 t c_4 H C \frac{B}{4} - T_1 n q) \|\boldsymbol{\mu}\|_2^2$$
$$= \frac{\eta}{Bm} (c_3 c_4 t C \frac{n}{4} - T_1 n q) \|\mu\|_2^2$$
$$\geq \frac{\eta c_3 c_4 C t n \|\mu\|_2^2}{8Bm} \tag{46}$$
$$\geq \frac{c_3 c_4 C C_4 n \|\mu\|_2^2}{(P-1)^2 \sigma_p^2 d} = \Theta(n \cdot \text{SNR}^2) = \Theta(\widehat{\gamma}),$$

where the second inequality is due to Lemma B.6, the third inequality is due to $q < \frac{C_4 C c_3 c_4}{8 C_3}$ in Condition 3.1.

And it follows directly from (45) that

$$\overline{\rho}_{j,r,i}^{(T_1,0)} \leq \frac{3\eta (P-1)^2 \sigma_p^2 d T_1 H}{2Bm} = \frac{3 C_3}{2}, \quad \overline{\rho}_{j,r,i}^{(T_1,0)} = O(1),$$

which completes the proof. $\qquad\square$

### C.2.2 Second Stage

By the signal-noise decomposition, at the end of the first stage, we have

$$\mathbf{w}_{j,r}^{(t,b)} = \mathbf{w}_{j,r}^{(0,0)} + j\gamma_{j,r}^{(t,b)} \|\boldsymbol{\mu}\|_2^{-2} \boldsymbol{\mu} + \frac{1}{P-1} \sum_{i=1}^n \overline{\rho}_{j,r,i}^{(t,b)} \|\boldsymbol{\xi}_i\|_2^{-2} \boldsymbol{\xi}_i + \frac{1}{P-1} \sum_{i=1}^n \underline{\rho}_{j,r,i}^{(t,b)} \|\boldsymbol{\xi}_i\|_2^{-2} \boldsymbol{\xi}_i.$$

for $j \in [\pm 1]$ and $r \in [m]$. By the results we get in the first stage, we know that at the beginning of this stage, we have the following property holds:

- $\overline{\rho}_{j,r^*,i}^{(T_1,0)} \geq 2$ for any $r^* \in S_i^{(0,0)} = \{r \in [m] : \langle \mathbf{w}_{y_i,r}^{(0,0)}, \boldsymbol{\xi}_i \rangle > \sigma_0 \sigma_p \sqrt{d}/\sqrt{2}\}, j \in \{\pm 1\}$ and $i \in [n]$ with $y_i = j$.

- $\max_{j,r,i} |\underline{\rho}_{j,r,i}^{(T_1,0)}| = \max\{\beta, O(n\sqrt{\log(n/\delta)}\log(T^*)/\sqrt{d})\}$.

- $\gamma_{j,r}^{(T_1,0)} = \Theta(\widehat{\gamma})$ for any $j \in \{\pm 1\}, r \in [m]$.

where $\widehat{\gamma} = n \cdot \mathrm{SNR}^2$. Now we choose $\mathbf{W}^*$ as follows

$$\mathbf{w}_{j,r}^* = \mathbf{w}_{j,r}^{(0,0)} + \frac{20\log(2/\epsilon)}{P-1}\Big[\sum_{i=1}^n \mathbb{1}(j=y_i)\cdot\frac{\boldsymbol{\xi}_i}{\|\boldsymbol{\xi}_i\|_2^2}\Big].$$

**Lemma C.10.** *Under the same conditions as Theorem 3.2, we have that* $\|\mathbf{W}^{(T_1,0)} - \mathbf{W}^*\|_F \leq \widetilde{O}(m^{1/2}n^{1/2}(P-1)^{-1}\sigma_p^{-1}d^{-1/2}(1+\max\{\beta, n\sqrt{\log(n/\delta)}\log(T^*)/\sqrt{d}\}))$.

*Proof.*

$$\|\mathbf{W}^{(T_1,0)} - \mathbf{W}^*\|_F$$
$$\leq \|\mathbf{W}^{(T_1,0)} - \mathbf{W}^{(0,0)}\|_F + \|\mathbf{W}^* - \mathbf{W}^{(0,0)}\|_F$$
$$\leq O(\sqrt{m})\max_{j,r}\gamma_{j,r}^{(T_1,0)}\|\boldsymbol{\mu}\|_2^{-1} + \frac{1}{P-1}O(\sqrt{m})\max_{j,r}\Big\|\sum_{i=1}^n \overline{\rho}_{j,r,i}^{(T_1,0)}\cdot\frac{\boldsymbol{\xi}_i}{\|\boldsymbol{\xi}_i\|_2^2} + \sum_{i=1}^n \underline{\rho}_{j,r,i}^{(T_1,0)}\cdot\frac{\boldsymbol{\xi}_i}{\|\boldsymbol{\xi}_i\|_2^2}\Big\|_2$$
$$\quad + O(m^{1/2}n^{1/2}\log(1/\epsilon)(P-1)^{-1}\sigma_p^{-1}d^{-1/2})$$
$$= O(m^{1/2}\widehat{\gamma}\|\boldsymbol{\mu}\|_2^{-1}) + \widetilde{O}(m^{1/2}n^{1/2}(P-1)^{-1}\sigma_p^{-1}d^{-1/2}(1+\max\{\beta, n\sqrt{\log(n/\delta)}\log(T^*)/\sqrt{d}\}))$$
$$\quad + O(m^{1/2}n^{1/2}\log(1/\epsilon)(P-1)^{-1}\sigma_p^{-1}d^{-1/2})$$
$$= O(m^{1/2}n\cdot\mathrm{SNR}\cdot(P-1)^{-1}\sigma_p^{-1}d^{-1/2}(1+\max\{\beta, n\sqrt{\log(n/\delta)}\log(T^*)/\sqrt{d}\}))$$
$$\quad + \widetilde{O}(m^{1/2}n^{1/2}\log(1/\epsilon)(P-1)^{-1}\sigma_p^{-1}d^{-1/2})$$
$$= \widetilde{O}(m^{1/2}n^{1/2}(P-1)^{-1}\sigma_p^{-1}d^{-1/2}(1+\max\{\beta, n\sqrt{\log(n/\delta)}\log(T^*)/\sqrt{d}\})),$$

where the first inequality is by triangle inequality, the second inequality and the first equality are by our decomposition of $\mathbf{W}^{(T_1,0)}$, $\mathbf{W}^*$ and Lemma B.1; the second equality is by $n\cdot\mathrm{SNR}^2 = \Theta(\widehat{\gamma})$ and $\mathrm{SNR} = \|\boldsymbol{\mu}\|/(P-1)\sigma_p d^{1/2}$; the third equality is by $n^{1/2}\cdot\mathrm{SNR} = O(1)$. $\qquad\square$

**Lemma C.11.** *Under the same conditions as Theorem 3.2, we have that*

$$y_i\langle\nabla f(\mathbf{W}^{(t,b)}, \mathbf{x}_i), \mathbf{W}^*\rangle \geq \log(2/\epsilon)$$

*for all* $(T_1, 0) \leq (t,b) \leq (T^*, 0)$.

*Proof of Lemma C.11.* Recall that $f(\mathbf{W}^{(t,b)}) = (1/m)\sum_{j,r} j\cdot[\sigma(\langle\mathbf{w}_{j,r}^{(t,b)}, \widehat{y}_i\boldsymbol{\mu}\rangle) + (P-1)\sigma(\langle\mathbf{w}_{j,r}^{(t,b)}, \boldsymbol{\xi}_i\rangle)]$, thus we have

$$y_i\langle\nabla f(\mathbf{W}^{(t,b)}, \mathbf{x}_i), \mathbf{W}^*\rangle$$
$$= \frac{1}{m}\sum_{j,r}\sigma'(\langle\mathbf{w}_{j,r}^{(t,b)}, \widehat{y}_i\boldsymbol{\mu}\rangle)\langle y_i\widehat{y}_i\boldsymbol{\mu}, j\mathbf{w}_{j,r}^*\rangle + \frac{P-1}{m}\sum_{j,r}\sigma'(\langle\mathbf{w}_{j,r}^{(t,b)}, \boldsymbol{\xi}_i\rangle)\langle y_i\boldsymbol{\xi}_i, j\mathbf{w}_{j,r}^*\rangle$$
$$= \frac{1}{m}\sum_{j,r}\sum_{i'=1}^n\sigma'(\langle\mathbf{w}_{j,r}^{(t,b)}, \boldsymbol{\xi}_i\rangle)20\log(2/\epsilon)\mathbb{1}(j=y_{i'})\cdot\frac{\langle\boldsymbol{\xi}_{i'}, \boldsymbol{\xi}_i\rangle}{\|\boldsymbol{\xi}_{i'}\|_2^2}$$
$$\quad + \frac{1}{m}\sum_{j,r}\sum_{i'=1}^n\sigma'(\langle\mathbf{w}_{j,r}^{(t,b)}, \widehat{y}_i\boldsymbol{\mu}\rangle)20\log(2/\epsilon)\mathbb{1}(j=y_{i'})\cdot\frac{\langle\widehat{y}_i\boldsymbol{\mu}, \boldsymbol{\xi}_{i'}\rangle}{\|\boldsymbol{\xi}_{i'}\|_2^2}$$
$$\quad + \frac{1}{m}\sum_{j,r}\sigma'(\langle\mathbf{w}_{j,r}^{(t,b)}, \widehat{y}_i\boldsymbol{\mu}\rangle)\langle y_i\widehat{y}_i\boldsymbol{\mu}, j\mathbf{w}_{j,r}^{(0,0)}\rangle + \frac{P-1}{m}\sum_{j,r}\sigma'(\langle\mathbf{w}_{j,r}^{(t,b)}, \boldsymbol{\xi}_i\rangle)\langle y_i\boldsymbol{\xi}_i, j\mathbf{w}_{j,r}^{(0,0)}\rangle$$
$$\geq \frac{1}{m}\sum_{j=y_i,r}\sigma'(\langle\mathbf{w}_{j,r}^{(t,b)}, \boldsymbol{\xi}_i\rangle)20\log(2/\epsilon) - \frac{1}{m}\sum_{j,r}\sum_{i'\neq i}\sigma'(\langle\mathbf{w}_{j,r}^{(t,b)}, \boldsymbol{\xi}_i\rangle)20\log(2/\epsilon)\cdot\frac{|\langle\boldsymbol{\xi}_{i'}, \boldsymbol{\xi}_i\rangle|}{\|\boldsymbol{\xi}_{i'}\|_2^2}$$
$$\quad - \frac{1}{m}\sum_{j,r}\sum_{i'=1}^n\sigma'(\langle\mathbf{w}_{j,r}^{(t,b)}, \widehat{y}_i\boldsymbol{\mu}\rangle)20\log(2/\epsilon)\cdot\frac{|\langle\widehat{y}_i\boldsymbol{\mu}, \boldsymbol{\xi}_{i'}\rangle|}{\|\boldsymbol{\xi}_{i'}\|_2^2} - \frac{1}{m}\sum_{j,r}\sigma'(\langle\mathbf{w}_{j,r}^{(t,b)}, \widehat{y}_i\boldsymbol{\mu}\rangle)\beta$$

$$\geq \frac{1}{m}\sum_{j=y_i,r}\sigma'(\langle\mathbf{w}_{j,r}^{(t,b)},\boldsymbol{\xi}_i\rangle)20\log(2/\epsilon)\underbrace{-\frac{1}{m}\sum_{j,r}\sigma'(\langle\mathbf{w}_{j,r}^{(t,b)},\boldsymbol{\xi}_i\rangle)20\log(2/\epsilon)O\big(n\sqrt{\log(n/\delta)}/\sqrt{d}\big)}$$

$$\underbrace{-\frac{1}{m}\sum_{j,r}\sigma'(\langle\mathbf{w}_{j,r}^{(t,b)},\widehat{y}_i\boldsymbol{\mu}\rangle)O\big(n\sqrt{\log(n/\delta)}\cdot\mathrm{SNR}\cdot d^{-1/2}\big)}_{\mathrm{I}_{11}}\underbrace{-\frac{1}{m}\sum_{j,r}\sigma'(\langle\mathbf{w}_{j,r}^{(t,b)},y_i\boldsymbol{\mu}\rangle)\beta}_{\mathrm{I}_{12}}, \quad (47)$$

where the first inequality is by Lemma B.2 and the last inequality is by Lemma B.1. Then, we will bound each term in (47) respectively.

For $\mathrm{I}_{10}, \mathrm{I}_{11}, \mathrm{I}_{12}, \mathrm{I}_{14}$, we have that

$$|\mathrm{I}_{10}| \leq O\big(n\sqrt{\log(n/\delta)}/\sqrt{d}\big), \ |\mathrm{I}_{11}| \leq O\big(n\sqrt{\log(n/\delta)}\cdot\mathrm{SNR}\cdot d^{-1/2}\big),$$
$$|\mathrm{I}_{12}| \leq O(\beta), \quad (48)$$

For $j = y_i$ and $r \in S_i^{(0)}$, according to Lemma C.3, we have

$$(P-1)\langle\mathbf{w}_{j,r}^{(t,b)},\boldsymbol{\xi}_i\rangle \geq (P-1)\langle\mathbf{w}_{j,r}^{(0,0)},\boldsymbol{\xi}_i\rangle + \overline{\rho}_{j,r,i}^{(t,b)} - 5n\sqrt{\frac{\log(4n^2/\delta)}{d}}\alpha$$

$$\geq 2 - \beta - 5n\sqrt{\frac{\log(4n^2/\delta)}{d}}\alpha$$

$$\geq 1.5 - \beta > 0,$$

where the first inequality is by Lemma C.3; the second inequality is by $5n\sqrt{\frac{\log(4n^2/\delta)}{d}} \leq 0.5$; and the last inequality is by $\beta < 1.5$. Therefore, for $\mathrm{I}_9$, according to the fourth statement of Proposition C.8, we have

$$\mathrm{I}_9 \geq \frac{1}{m}|\widetilde{S}_i^{(t,b)}|20\log(2/\epsilon) \geq 2\log(2/\epsilon). \quad (49)$$

By plugging (48) and (49) into (47) and according to triangle inequality we have

$$y_i\langle\nabla f(\mathbf{W}^{(t,b)},\mathbf{x}_i),\mathbf{W}^*\rangle \geq \mathrm{I}_9 - |\mathrm{I}_{10}| - |\mathrm{I}_{11}| - |\mathrm{I}_{12}| - |\mathrm{I}_{14}| \geq \log(2/\epsilon),$$

which completes the proof. □

**Lemma C.12.** *Under Assumption 3.1, for $(0,0) \leq (t,b) \leq (T^*,0)$, the following result holds.*

$$\|\nabla L_{\mathcal{I}_{t,b}}(\mathbf{W}^{(t,b)})\|_F^2 \leq O(\max\{\|\boldsymbol{\mu}\|_2^2, (P-1)^2\sigma_p^2 d\})L_{\mathcal{I}_{t,b}}(\mathbf{W}^{(t,b)}).$$

*Proof.* We first prove that

$$\|\nabla f(\mathbf{W}^{(t,b)},\mathbf{x}_i)\|_F = O(\max\{\|\boldsymbol{\mu}\|_2, (P-1)\sigma_p\sqrt{d}\}). \quad (50)$$

Without loss of generality, we suppose that $\widehat{y}_i = 1$. Then we have that

$$\|\nabla f(\mathbf{W}^{(t,b)},\mathbf{x}_i)\|_F \leq \frac{1}{m}\sum_{j,r}\left\|\left[\sigma'(\langle\mathbf{w}_{j,r}^{(t,b)},\boldsymbol{\mu}\rangle)\boldsymbol{\mu} + (P-1)\sigma'(\langle\mathbf{w}_{j,r}^{(t,b)},\boldsymbol{\xi}_i\rangle)\boldsymbol{\xi}_i\right]\right\|_2$$

$$\leq \frac{1}{m}\sum_{j,r}\sigma'(\langle\mathbf{w}_{j,r}^{(t,b)},\boldsymbol{\mu}\rangle)\|\boldsymbol{\mu}\|_2 + \frac{P-1}{m}\sum_{j,r}\sigma'(\langle\mathbf{w}_{j,r}^{(t,b)},\boldsymbol{\xi}_i\rangle)\|\boldsymbol{\xi}_i\|_2$$

$$\leq 4\max\{\|\boldsymbol{\mu}\|_2, 2(P-1)\sigma_p\sqrt{d}\},$$

where the first and second inequalities are by triangle inequality, the third inequality is by Lemma B.1.

Then we upper bound the gradient norm $\|\nabla L_S(\mathbf{W}^{(t,b)})\|_F$ as:

$$\|\nabla L_{\mathcal{I}_{t,b}}(\mathbf{W}^{(t,b)})\|_F^2 \leq \left[\frac{1}{B}\sum_{i\in\mathcal{I}_{t,b}}\ell'\big(y_i f(\mathbf{W}^{(t,b)},\mathbf{x}_i)\big)\|\nabla f(\mathbf{W}^{(t,b)},\mathbf{x}_i)\|_F\right]^2$$

$$\leq \left[ \frac{1}{B} \sum_{i \in \mathcal{I}_{t,b}} O(\max\{\|\boldsymbol{\mu}\|_2, (P-1)\sigma_p\sqrt{d}\}) \big( -\ell'\big(y_i f(\mathbf{W}^{(t,b)}, \mathbf{x}_i)\big)\big) \right]^2$$

$$\leq O(\max\{\|\boldsymbol{\mu}\|_2^2, (P-1)^2\sigma_p^2 d\}) \cdot \frac{1}{B} \sum_{i \in \mathcal{I}_{t,b}} -\ell'\big(y_i f(\mathbf{W}^{(t,b)}, \mathbf{x}_i)\big)$$

$$\leq O(\max\{\|\boldsymbol{\mu}\|_2^2, (P-1)\sigma_p^2 d\}) L_{\mathcal{I}_{t,b}}(\mathbf{W}^{(t,b)}),$$

where the first inequality is by triangle inequality, the second inequality is by (50), the third inequality is by Cauchy-Schwartz inequality and the last inequality is due to the property of the cross entropy loss $-\ell' \leq \ell$. $\qquad\square$

**Lemma C.13.** *Under the same conditions as Theorem 3.2, we have that*

$$\|\mathbf{W}^{(t,b)} - \mathbf{W}^*\|_F^2 - \|\mathbf{W}^{(t+1,b)} - \mathbf{W}^*\|_F^2 \geq \eta L_{\mathcal{I}_{t,b}}(\mathbf{W}^{(t,b)}) - \eta\epsilon$$

*for all* $(T_1, 0) \leq (t, b) \leq (T^*, 0)$.

*Proof of Lemma C.13.* We have

$$\|\mathbf{W}^{(t,b)} - \mathbf{W}^*\|_F^2 - \|\mathbf{W}^{(t+1,b)} - \mathbf{W}^*\|_F^2$$
$$= 2\eta \langle \nabla L_{\mathcal{I}_{t,b}}(\mathbf{W}^{(t,b)}), \mathbf{W}^{(t,b)} - \mathbf{W}^* \rangle - \eta^2 \|\nabla L_S(\mathbf{W}^{(t,b)})\|_F^2$$
$$= \frac{2\eta}{B} \sum_{i \in \mathcal{I}_{t,b}} \ell_i'^{(t,b)}[y_i f(\mathbf{W}^{(t,b)}, \mathbf{x}_i) - \langle \nabla f(\mathbf{W}^{(t,b)}, \mathbf{x}_i), \mathbf{W}^* \rangle] - \eta^2 \|\nabla L_{\mathcal{I}_{t,b}}(\mathbf{W}^{(t,b)})\|_F^2$$
$$\geq \frac{2\eta}{B} \sum_{i \in \mathcal{I}_{t,b}} \ell_i'^{(t,b)}[y_i f(\mathbf{W}^{(t,b)}, \mathbf{x}_i) - \log(2/\epsilon)] - \eta^2 \|\nabla L_S(\mathbf{W}^{(t,b)})\|_F^2$$
$$\geq \frac{2\eta}{B} \sum_{i \in \mathcal{I}_{t,b}} [\ell\big(y_i f(\mathbf{W}^{(t,b)}, \mathbf{x}_i)\big) - \epsilon/2] - \eta^2 \|\nabla L_{\mathcal{I}_{t,b}}(\mathbf{W}^{(t,b)})\|_F^2$$
$$\geq \eta L_{\mathcal{I}_{t,b}}(\mathbf{W}^{(t,b)}) - \eta\epsilon,$$

where the first inequality is by Lemma C.11; the second inequality is due to the convexity of the cross-entropy function; the last inequality is due to Lemma C.12. $\qquad\square$

**Lemma C.14.** *Under the same conditions as Theorem 3.2, with probability* $1 - \delta$, *we have that*

$$\left| L_{\mathcal{I}^{(t,b)}}(\mathbf{W}^{(t,b)}) - L_{\mathcal{I}^{(t,b)}}(\mathbf{W}^{(t,0)}) \right| \leq \epsilon,$$

*for all* $(T_1, 0) \leq (t, b) \leq (T^*, 0)$.

*Proof.*

$$\left| L_{\mathcal{I}^{(t,b)}}(\mathbf{W}^{(t,b)}) - L_{\mathcal{I}^{(t,b)}}(\mathbf{W}^{(t,0)}) \right|$$
$$\leq \frac{1}{B} \sum_{i \in \mathcal{I}_{t,b}} \left| \ell(y_i f(\mathbf{W}^{(t,b)}, x_i)) - \ell(y_i f(\mathbf{W}^{(t,0)}, x_i)) \right|$$
$$\leq \frac{1}{B} \sum_{i \in \mathcal{I}_{t,b}} \left| y_i f(\mathbf{W}^{(t,b)}, x_i) - y_i f(\mathbf{W}^{(t,0)}, x_i) \right|$$
$$\leq \frac{1}{B} \sum_{i \in \mathcal{I}_{t,b}} \frac{1}{m} \sum_{j,r} \left( \left| \langle \mathbf{w}_{j,r}^{(t,b)} - \mathbf{w}_{j,r}^{(t,0)}, \boldsymbol{\mu} \rangle \right| + (P-1) \left| \langle \mathbf{w}_{j,r}^{(t,b)} - \mathbf{w}_{j,r}^{(t,0)}, \boldsymbol{\xi}_i \rangle \right| \right)$$
$$\leq \frac{H\eta(P-1)}{Bm} \|\mu\|_2 \sigma_p \sqrt{2\log(6n/\delta)} + \frac{H\eta(P-1)^2}{Bm} 2\sigma_p^2 \sqrt{d\log(6n^2/\delta)}$$
$$\leq \epsilon,$$

where the first inequality is due to triangle inequality, the second inequality is due to $|\ell_i'| \leq 1$, the third inequality is due to triangle inequality and the definition of neural networks, the forth inequality is due to parameter update rule (17) and Lemma B.1, and the fifth inequality is due to Condition 3.1. $\qquad\square$

**Lemma C.15.** *Under the same conditions as Theorem 3.2, for all $T_1 \leq t \leq T^*$, we have* $\max_{j,r,i} |\varrho_{-j,r,i}^{(t,b)}| = \max \left\{ O\left( \sqrt{\log(mn/\delta)} \cdot \sigma_0(P-1)\sigma_p\sqrt{d} \right), O\left( n\sqrt{\log(n/\delta)} \log(T^*)/\sqrt{d} \right) \right\}$. *Besides,*

$$\frac{1}{(s-T_1)H} \sum_{(T_1,0)\leq(t,b)<(s,0)} L_{\mathcal{I}_{t,b}}(\mathbf{W}^{(t,b)}) \leq \frac{\|\mathbf{W}^{(T_1,0)} - \mathbf{W}^*\|_F^2}{\eta(s-T_1)H} + \epsilon$$

*for all $T_1 \leq t \leq T^*$. Therefore, we can find an iterate with training loss smaller than $2\epsilon$ within $T = T_1 + \left\lceil \|\mathbf{W}^{(T_1)} - \mathbf{W}^*\|_F^2/(\eta\epsilon) \right\rceil = T_1 + \widetilde{O}(\eta^{-1}\epsilon^{-1}mnd^{-1}\sigma_p^{-2})$ iterations.*

*Proof of Lemma C.15.* Note that $\max_{j,r,i} |\varrho_{-j,r,i}^{(t)}| = \max \left\{ O\left( \sqrt{\log(mn/\delta)} \cdot \sigma_0(P-1)\sigma_p\sqrt{d} \right), O\left( n\sqrt{\log(n/\delta)} \log(T^*)/\sqrt{d} \right) \right\}$ can be proved in the same way as Lemma C.9.

For any $T_1 \leq s \leq T^*$, by taking a summation of the inequality in Lemma C.13 and dividing $(s-T_1)H$ on both sides, we obtain that

$$\frac{1}{(s-T_1)H} \sum_{(T_1,0)\leq(t,b)<(s,0)} L_{\mathcal{I}_{t,b}}(\mathbf{W}^{(t,b)}) \leq \frac{\|\mathbf{W}^{(T_1,0)} - \mathbf{W}^*\|_F^2}{\eta(s-T_1)H} + \epsilon.$$

According to the definition of $T$, we have

$$\frac{1}{(T-T_1)H} \sum_{(T_1,0)\leq(t,b)<(T,0)} L_{\mathcal{I}_{t,b}}(\mathbf{W}^{(t,b)}) \leq 2\epsilon.$$

Then there exists an epoch $T_1 \leq t \leq T^*$ such that

$$\frac{1}{H} \sum_{b=0}^{H-1} L_{\mathcal{I}_{t,b}}(\mathbf{W}^{(t,b)}) \leq 2\epsilon.$$

Thus, according to Lemma C.14, we have

$$L_S(\mathbf{W}^{(t,0)}) \leq 3\epsilon.$$

$\square$

**Lemma C.16.** *Under the same conditions as Theorem 3.2, we have*

$$\sum_{i=1}^{n} \overline{\rho}_{j,r,i}^{(t,b)}/\gamma_{j',r'}^{(t,b)} = \Theta(\mathrm{SNR}^{-2}) \tag{51}$$

*for all $j, j' \in \{\pm 1\}$, $r, r' \in [m]$ and $(T_2,0) \leq (t,b) \leq (T^*,0)$.*

*Proof.* Now suppose that there exists $(0,0) < (\widetilde{T},0) \leq (T^*,0)$ such that $\sum_{i=1}^{n} \overline{\rho}_{j,r,i}^{(t,0)}/\gamma_{j',r'}^{(t,b)} = \Theta(\mathrm{SNR}^{-2})$ for all $(0,0) < (t,0) < (\widetilde{T},0)$. Then for $\overline{\rho}_{j,r,i}^{(t,b)}$, according to Lemma C.1, we have

$$\gamma_{j,r}^{(t+1,0)} = \gamma_{j,r}^{(t,b)} - \frac{\eta}{Bm} \cdot \sum_{b<H} \left[ \sum_{i\in S_+\cap\mathcal{I}_{t,b}} \ell_i'^{(t,b_i^{(t)})} \sigma'(\langle \mathbf{w}_{j,r}^{(t,b_i^{(t)})}, \widehat{y}_i \cdot \boldsymbol{\mu} \rangle) \right.$$

$$\left. - \sum_{i\in S_-\cap\mathcal{I}_{t,b}} \ell_i'^{(t,b_i^{(t)})} \sigma'(\langle \mathbf{w}_{j,r}^{(t,b_i^{(t)})}, \widehat{y}_i \cdot \boldsymbol{\mu} \rangle) \right] \cdot \|\boldsymbol{\mu}\|_2^2,$$

$$\overline{\rho}_{j,r,i}^{(t+1,0)} = \overline{\rho}_{j,r,i}^{(t,0)} - \frac{\eta(P-1)^2}{Bm} \cdot \ell_i'^{(t,b_i^{(t)})} \cdot \sigma'(\langle \mathbf{w}_{j,r}^{(t,b_i^{(t)})}, \boldsymbol{\xi}_i \rangle) \cdot \|\boldsymbol{\xi}_i\|_2^2 \cdot \mathbb{1}(y_i = j),$$

It follows that

$$\sum_{i=1}^{n} \overline{\rho}_{j,r,i}^{(\widetilde{T},0)}$$

$$
\begin{aligned}
&= \sum_{i:y_i=j} \overline{\rho}_{j,r,i}^{(\widetilde{T},0)} \\
&= \sum_{i:y_i=j} \overline{\rho}_{j,r,i}^{(\widetilde{T}-1,0)} - \frac{\eta(P-1)^2}{Bm} \cdot \sum_{i:y_i=j} \ell_i'^{(\widetilde{T}-1,b_i^{(\widetilde{T}-1)})} \cdot \sigma'(\langle \mathbf{w}_{j,r}^{(\widetilde{T}-1,b_i^{(\widetilde{T}-1)})}, \boldsymbol{\xi}_i \rangle) \|\boldsymbol{\xi}_i\|_2^2 \\
&= \sum_{i=1}^{n} \overline{\rho}_{j,r,i}^{(\widetilde{T}-1,0)} - \frac{\eta(P-1)^2}{Bm} \cdot \sum_{i \in \widetilde{S}_{j,r}^{(\widetilde{T}-1,\widetilde{b}_i^{(\widetilde{T}-1)})}} \ell_i'^{(\widetilde{T}-1,b_i^{(\widetilde{T}-1)})} \|\boldsymbol{\xi}_i\|_2^2 \\
&\geq \sum_{i=1}^{n} \overline{\rho}_{j,r,i}^{(\widetilde{T}-1)} + \frac{\eta(P-1)^2 \sigma_p^2 d H \Phi(-1)}{8m} \cdot \min_{i \in \widetilde{S}_{j,r}^{(\widetilde{t},\widetilde{b}-1)} \cap \mathcal{I}_{\widetilde{t},\widetilde{b}-1}} |\ell_i'^{(\widetilde{T}-1,b_i^{(\widetilde{T}-1)})}|, \quad (52)
\end{aligned}
$$

where the last equality is by the definition of $S_{j,r}^{(\widetilde{T}-1)}$ as $\{i \in [n] : y_i = j, \langle \mathbf{w}_{j,r}^{(\widetilde{T}-1)}, \boldsymbol{\xi}_i \rangle > 0\}$; the last inequality is by Lemma B.1 and the fifth statement of Lemma C.8. And

$$
\begin{aligned}
\gamma_{j',r'}^{(\widetilde{T},0)} &\leq \gamma_{j',r'}^{(\widetilde{T}-1,0)} - \frac{\eta}{Bm} \cdot \sum_{i \in S_+} \ell_i'^{(\widetilde{T}-1,b_i^{(\widetilde{T}-1)})} \sigma'(\langle \mathbf{w}_{j',r'}^{(\widetilde{T}-1,b_i^{\widetilde{T}-1})}, \widehat{y}_i \cdot \boldsymbol{\mu} \rangle) \cdot \|\boldsymbol{\mu}\|_2^2 \\
&\leq \gamma_{j',r'}^{(\widetilde{T}-1,0)} + \frac{H\eta\|\boldsymbol{\mu}\|_2^2}{m} \cdot \max_{i \in S_+} |\ell_i'^{(\widetilde{T}-1)}|. \quad (53)
\end{aligned}
$$

According to the third statement of Lemma C.8, we have $\max_{i \in S_+} |\ell_i'^{(\widetilde{T}-1,b_i^{(\widetilde{T}-1)})}| \leq C_2 \min_{i \in S_{j,r}^{(\widetilde{T}-1,b_i^{(\widetilde{T}-1)})}} |\ell_i'^{(\widetilde{T}-1)}|$. Then by combining (52) and (53), we have

$$
\frac{\sum_{i=1}^{n} \overline{\rho}_{j,r,i}^{(\widetilde{T},0)}}{\gamma_{j',r'}^{(\widetilde{T},0)}} \geq \min \left\{ \frac{\sum_{i=1}^{n} \overline{\rho}_{j,r,i}^{(\widetilde{T}-1,0)}}{\gamma_{j',r'}^{(\widetilde{T}-1,0)}}, \frac{(P-1)^2 \sigma_p^2 d}{16 C_2 \|\boldsymbol{\mu}\|_2^2} \right\} = \Theta(\text{SNR}^{-2}). \quad (54)
$$

On the other hand, we will now show $\frac{\sum_{i=1}^{n} \overline{\rho}_{j,r,i}^{(t,0)}}{\gamma_{j',r'}^{(t,0)}} \leq \Theta(\text{SNR}^{-2})$ for $t \geq T_2$ by induction. By Lemma B.1 and (52), we have

$$
\begin{aligned}
\sum_{i=1}^{n} \overline{\rho}_{j,r,i}^{(T_2,0)} &\leq \sum_{i=1}^{n} \overline{\rho}_{j,r,i}^{(T_2-1,0)} + \frac{3\eta(P-1)^2 \sigma_p^2 dn}{2Bm} \\
&\leq \frac{3\eta(P-1)^2 \sigma_p^2 dn T_2}{2Bm}.
\end{aligned}
$$

And, by (46), we know that at $t = T_2$, we have

$$
\gamma_{j',r'}^{(T_2,0)} \geq \frac{\eta c_3 c_4 C T_2 n \|\mu\|_2^2}{8Bm}.
$$

Thus,

$$
\frac{\sum_{i=1}^{n} \overline{\rho}_{j,r,i}^{(T_2,0)}}{\gamma_{j',r'}^{(T_2,0)}} \leq \Theta(\text{SNR}^{-2}).
$$

Suppose $\frac{\sum_{i=1}^{n} \overline{\rho}_{j,r,i}^{(T,0)}}{\gamma_{j',r'}^{(T,0)}} \leq \Theta(\text{SNR}^{-2})$. According to the decomposition, we have:

$$
\begin{aligned}
\langle \mathbf{w}_{j,r}^{(T,b)}, \widehat{y}_i \boldsymbol{\mu} \rangle &= \langle \mathbf{w}_{j,r}^{(0,0)}, \widehat{y}_i \boldsymbol{\mu} \rangle + j \cdot \gamma_{j,r}^{(T,b)} \cdot \widehat{y}_i \\
&\quad + \frac{1}{P-1} \sum_{i=1}^{n} \overline{\rho}_{j,r,i}^{(T,b)} \cdot \|\boldsymbol{\xi}_i\|_2^{-2} \langle \boldsymbol{\xi}_i, \widehat{y}_i \boldsymbol{\mu} \rangle + \frac{1}{P-1} \sum_{i=1}^{n} \underline{\rho}_{j,r,i}^{(T,b)} \cdot \|\boldsymbol{\xi}_i\|_2^{-2} \langle \boldsymbol{\xi}_i, \widehat{y}_i \boldsymbol{\mu} \rangle. \quad (55)
\end{aligned}
$$

And we have that

$$|\langle \mathbf{w}_{j,r}^{(0,0)}, \widehat{y}_i\boldsymbol{\mu}\rangle + \frac{1}{P-1}\sum_{i=1}^{n}\overline{\rho}_{j,r,i}^{(T,b)}\cdot\|\boldsymbol{\xi}_i\|_2^{-2}\langle\boldsymbol{\xi}_i,\widehat{y}\boldsymbol{\mu}\rangle + \frac{1}{P-1}\sum_{i=1}^{n}\underline{\rho}_{j,r,i}^{(T,b)}\cdot\|\boldsymbol{\xi}_i\|_2^{-2}\langle\boldsymbol{\xi}_i,\widehat{y}\boldsymbol{\mu}\rangle|$$

$$\leq \beta/2 + |\sum_{i=1}^{n}\overline{\rho}_{j,r,i}^{(T,b)}|\frac{4\|\boldsymbol{\mu}\|_2\sqrt{2\log(6n/\delta)}}{\sigma_p d(P-1)}$$

$$\leq \beta/2 + \frac{\Theta(\mathrm{SNR}^{-1})\gamma_{j,r}^{(T,b)}}{\sqrt{d}}$$

$$\leq \gamma_{j,r}^{(T,0)},$$

where the first inequality is due to triangle inequality and Lemma B.1, the second inequality is due to induction hypothesis, and the last inequality is due to Condition 3.1.

Thus, the sign of $\langle\mathbf{w}_{j,r}^{(T,b)},\widehat{y}_i\boldsymbol{\mu}\rangle$ is persistent through out the epoch. Then, without loss of generality, we suppose $\langle\mathbf{w}_{j,r}^{(T,b)},\boldsymbol{\mu}\rangle > 0$. Thus, the update rule of $\gamma$ is:

$$\gamma_{j,r}^{(t,b+1)}$$

$$= \gamma_{j,r}^{(t,b)} - \frac{\eta}{Bm}\cdot\left[\sum_{i\in\mathcal{I}_{T,b}\cap S_+\cap S_1}\ell_i'^{(T,b)} - \sum_{i\in\mathcal{I}_{T,b}\cap S_-\cap S_1}\ell_i'^{(T,b)}\right]\|\boldsymbol{\mu}\|_2^2$$

$$\geq \gamma_{j,r}^{(T,b)} + \frac{\eta}{Bm}\cdot\left[\min_{i\in\mathcal{I}_{T,b}}\ell_i'^{(T,b)}|\mathcal{I}_{T,b}\cap S_+\cap S_1| - \max_{i\in\mathcal{I}_{T,b}}|\mathcal{I}_{T,b}\cap S_-\cap S_{-1}|\right]\cdot\|\boldsymbol{\mu}\|_2^2.$$

Therefore,

$$\gamma_{j,r}^{(T+1,0)} \geq \gamma_{j,r}^{(T,b)} + \frac{\eta}{Bm}\cdot\left[\min\ell_i^{(T,b_i^{(T)})}|S_+\cap S_1| - \max\ell_i^{(T,b_i^{(T)})}|S_-\cap S_{-1}|\right]\cdot\|\boldsymbol{\mu}\|_2^2. \tag{56}$$

And, by (52), we have

$$\sum_{i=1}^{n}\overline{\rho}_{j,r,i}^{(T+1,0)} \leq \sum_{i=1}^{n}\overline{\rho}_{j,r,i}^{(T,0)} + \frac{\eta(P-1)^2\sigma_p^2 dH\Phi(-1)}{8m}\cdot\max\left|\ell_i'^{(T,b_i^{(T)})}\right|. \tag{57}$$

Thus, combining (56) and (57), we have

$$\frac{\sum_{i=1}^{n}\overline{\rho}_{j,r,i}^{(T+1,0)}}{\gamma_{j,r}^{(T+1,0)}}$$

$$\leq \max\left\{\frac{\sum_{i=1}^{n}\overline{\rho}_{j,r,i}^{(T,0)}}{\gamma_{j,r}^{(T,0)}}, \frac{(P-1)^2\sigma_p^2 dn\Phi(-1)\cdot\max|\ell_i'^{(T,b_i^{(T)})}|}{8\left[\min\ell_i^{(T,b_i^{(T)})}|S_+\cap S_1| - \max\ell_i^{(T,b_i^{(T)})}|S_-\cap S_{-1}|\right]\cdot\|\boldsymbol{\mu}\|_2^2}\right\}$$

$$\leq \Theta(\mathrm{SNR}^{-2}), \tag{58}$$

where the last inequality is due to the induction hypothesis, the third statement of Lemma C.8, and Lemma B.5. Thus, by induction, we have for all $T_1 \leq t \leq T^*$ that

$$\frac{\sum_{i=1}^{n}\overline{\rho}_{j,r,i}^{(t,0)}}{\gamma_{j',r'}^{(t,0)}} \leq \Theta(\mathrm{SNR}^{-2}).$$

And for $(T_1,0) \leq (t,b) \leq (T^*,0)$, we can bound the ratio as follows:

$$\frac{\sum_{i=1}^{n}\overline{\rho}_{j,r,i}^{(t,b)}}{\gamma_{j',r'}^{(t,b)}} \leq \frac{4\sum_{i=1}^{n}\overline{\rho}_{j,r,i}^{(t,0)}}{\gamma_{j',r'}^{(t,0)}} \leq \Theta(\mathrm{SNR}^{-2}),$$

where the first inequality is due to the update rule of $\overline{\rho}_{j,r,i}^{(t,b)}$ and $\overline{\rho}_{j,r,i}^{(t,b)}$. Thus, we have completed the proof. $\qquad\square$

## C.3 Test Error Analysis

In this section, we present and prove the exact upper bound and lower bound of test error in Theorem 3.2. Since we have resolved the challenges brought by stochastic mini-batch parameter update, the remaining proof for test error is similar to the counterpart in Kou et al. (2023).

### C.3.1 Test Error Upper Bound

First, we prove the upper bound of test error in Theorem 3.2 when the training loss converges.

**Theorem C.17** (Second part of Theorem 3.2). *Under the same conditions as Theorem 3.2, then there exists a large constant $C_1$ such that when $n\|\boldsymbol{\mu}\|_2^2 \geq C_1(P-1)^4\sigma_p^4 d$, for time $t$ defined in Lemma C.15, we have the test error*

$$\mathbb{P}_{(\mathbf{x},y)\sim\mathcal{D}}\big(y \neq \mathrm{sign}(f(\mathbf{W}^{(t,0)},\mathbf{x}))\big) \leq p + \exp\left(-n\|\boldsymbol{\mu}\|_2^4/(C_2(P-1)^4\sigma_p^4 d)\right),$$

*where $C_2 = O(1)$.*

*Proof.* The proof is similar to the proof of Theorem E.1 in Kou et al. (2023). The only difference is substituting $\boldsymbol{\xi}$ in their proof with $(P-1)\boldsymbol{\xi}$. □

### C.3.2 Test Error Lower Bound

In this part, we prove the lower bound of the test error in Theorem 3.2. We give two key Lemmas.

**Lemma C.18.** *For $(T_1, 0) \leq (t, b) < (T^*, 0)$, denote $g(\boldsymbol{\xi}) = \sum_{j,r} j(P-1)\sigma(\langle \mathbf{w}_{j,r}^{(t,b)}, \boldsymbol{\xi}\rangle)$. There exists a fixed vector $\mathbf{v}$ with $\|\mathbf{v}\|_2 \leq 0.06\sigma_p$ such that*

$$\sum_{j'\in\{\pm 1\}} [g(j'\boldsymbol{\xi} + \mathbf{v}) - g(j'\boldsymbol{\xi})] \geq 4C_6 \max_{j\in\{\pm 1\}} \left\{\sum_r \gamma_{j,r}^{(t,b)}\right\}, \tag{59}$$

*for all $\boldsymbol{\xi} \in \mathbb{R}^d$.*

*Proof of Lemma C.18.* The proof is similar to the proof of Lemma 5.8 in Kou et al. (2023). The only difference is substituting $\boldsymbol{\xi}$ in their proof with $(P-1)\boldsymbol{\xi}$. □

**Lemma C.19** (Proposition 2.1 in Devroye et al. (2018)). *The TV distance between $\mathcal{N}(0, \sigma_p^2 \mathbf{I}_d)$ and $\mathcal{N}(\mathbf{v}, \sigma_p^2 \mathbf{I}_d)$ is smaller than $\|\mathbf{v}\|_2/2\sigma_p$.*

Then, we can prove the lower bound of the test error.

**Theorem C.20** (Third part of Theorem 3.2). *Suppose that $n\|\boldsymbol{\mu}\|_2^4 \leq C_3 d(P-1)^4\sigma_p^4$, then we have that $L_{\mathcal{D}}^{0-1}(\mathbf{W}^{(t,0)}) \geq p + 0.1$, where $C_3$ is an sufficiently large absolute constant.*

*Proof.* The proof is similar to the proof of Theorem 4.3 in Kou et al. (2023). The only difference is substituting $\boldsymbol{\xi}$ in their proof with $(P-1)\boldsymbol{\xi}$. □

## D Proofs for SAM

### D.1 Noise Memorization Prevention

The following lemma shows the update rule of the neural network

**Lemma D.1.** *We denote $\ell_i'^{(t,b)} = \ell'[y_i \cdot f(\mathbf{W}^{(t,b)}, \mathbf{x}_i)]$, then the adversarial point of $\mathbf{W}^{(t,b)}$ is $\mathbf{W}^{(t,b)} + \widehat{\boldsymbol{\epsilon}}^{(t,b)}$, where*

$$\widehat{\boldsymbol{\epsilon}}_{j,r}^{(t,b)} = \frac{\tau}{m} \frac{\sum_{i\in\mathcal{I}_{t,b}}\sum_{p\in[P]}\ell_i'^{(t,b)} j \cdot y_i\sigma'(\langle \mathbf{w}_{j,r}^{(t,b)}, \mathbf{x}_{i,p}\rangle)\mathbf{x}_{i,p}}{\|\nabla_{\mathbf{W}}L_{\mathcal{I}_{t,b}}(\mathbf{W}^{(t,b)})\|_F}.$$

*Then the training update rule of the parameter is*

$$\mathbf{w}_{j,r}^{(t+1,b)} = \mathbf{w}_{j,r}^{(t,b)} - \frac{\eta}{Bm}\sum_{i\in\mathcal{I}_{t,b}}\sum_{p\in[P]}\ell_i'^{(t,b)}\sigma'(\langle \mathbf{w}_{j,r}^{(t,b)} + \widehat{\boldsymbol{\epsilon}}_{j,r}^{(t,b)}, \mathbf{x}_{i,p}\rangle)j \cdot \mathbf{x}_{i,p}$$

$$= \mathbf{w}_{j,r}^{(t,b)} - \frac{\eta}{Bm} \sum_{i \in \mathcal{I}_{t,b}} \sum_{p \in [P]} \ell_i'^{(t,b)} \sigma'(\langle \mathbf{w}_{j,r}^{(t,b)}, \mathbf{x}_{i,p} \rangle + \langle \widehat{\boldsymbol{\epsilon}}_{t,j,r}, \mathbf{x}_{i,p} \rangle) j \cdot \mathbf{x}_{i,p}$$

$$= \mathbf{w}_{j,r}^{(t,b)} - \frac{\eta}{Bm} \sum_{i \in \mathcal{I}_{t,b}} \ell_i'^{(t,b)} \sigma'(\langle \mathbf{w}_{j,r}^{(t,b)}, y\boldsymbol{\mu} \rangle + \langle \widehat{\boldsymbol{\epsilon}}_{t,j,r}, y\boldsymbol{\mu} \rangle) j \boldsymbol{\mu}$$

$$\underbrace{- \frac{\eta(P-1)}{Bm} \sum_{i \in \mathcal{I}_{t,b}} \ell_i'^{(t,b)} \sigma'(\langle \mathbf{w}_{j,r}^{(t,b)}, \boldsymbol{\xi}_i \rangle + \langle \widehat{\boldsymbol{\epsilon}}_{j,r}^{(t,b)}, \boldsymbol{\xi}_i \rangle) j y_i \boldsymbol{\xi}_i}_{\mathrm{NoiseTerm}}.$$

We will show that the noise term will be small if we train with the SAM algorithm. We consider the first stage where $t \leq T_1$ where $T_1 = mB/(12n\eta\|\boldsymbol{\mu}\|_2^2)$. Then, the following property holds.

**Proposition D.2.** *Under Assumption 3.1, for $0 \leq t \leq T_1$, we have that*

$$\gamma_{j,r}^{(0,0)}, \overline{\rho}_{j,r,i}^{(0,0)}, \underline{\rho}_{j,r,i}^{(0,0)} = 0, \tag{60}$$

$$0 \leq \gamma_{j,r}^{(t,b)} \leq 1/12, \tag{61}$$

$$0 \leq \overline{\rho}_{j,r,i}^{(t,b)} \leq 1/12, \tag{62}$$

$$0 \geq \underline{\rho}_{j,r,i}^{(t,b)} \geq -\beta - 10\sqrt{\frac{\log(6n^2/\delta)}{d}} n. \tag{63}$$

*Besides, $\gamma_{j,r}^{(T_1,0)} = \Omega(1)$.*

**Lemma D.3.** *Under Assumption 3.1, suppose* (27), (28) *and* (29) *hold at iteration $t$. Then, for all $r \in [m]$, $j \in \{\pm 1\}$ and $i \in [n]$,*

$$\left| \langle \mathbf{w}_{j,r}^{(t,b)} - \mathbf{w}_{j,r}^{(0,0)}, \boldsymbol{\mu} \rangle - j \cdot \gamma_{j,r}^{(t,b)} \right| \leq \mathrm{SNR}\sqrt{\frac{32\log(6n/\delta)}{d}} n\alpha, \tag{64}$$

$$\left| \langle \mathbf{w}_{j,r}^{(t,b)} - \mathbf{w}_{j,r}^{(0,0)}, \boldsymbol{\xi}_i \rangle - \frac{1}{P-1}\underline{\rho}_{j,r,i}^{(t,b)} \right| \leq \frac{5}{P-1}\sqrt{\frac{\log(6n^2/\delta)}{d}} n\alpha, \ j \neq y_i, \tag{65}$$

$$\left| \langle \mathbf{w}_{j,r}^{(t,b)} - \mathbf{w}_{j,r}^{(0,0)}, \boldsymbol{\xi}_i \rangle - \frac{1}{P-1}\overline{\rho}_{j,r,i}^{(t,b)} \right| \leq \frac{5}{P-1}\sqrt{\frac{\log(6n^2/\delta)}{d}} n\alpha, \ j = y_i. \tag{66}$$

*Proof of Lemma D.3.* Notice that $1/12 < \alpha$, if the condition (61), (62), (63) holds, (27), (28) and (29) also holds. Therefore, by Lemma C.3, we know that Lemma D.3 also holds. $\square$

**Lemma D.4.** *Under Assumption 3.1, suppose* (61), (62), (63) *hold at iteration $t, b$. Then, for all $j \in \{\pm 1\}$ and $i \in [n]$, $F_j(\mathbf{W}_j^{(t,b)}, \mathbf{x}_i) \leq 0.5$. Therefore $-0.3 \geq \ell_i' \geq -0.7$.*

*Proof.* Notice that $1/12 < \alpha$, if the condition (61), (62), (63) holds, (27), (28) and (29) also holds. Therefore, by Lemma C.4, we know that for all $j \neq y_i$ and $i \in [n]$, $F_j(\mathbf{W}_j^{(t,b)}, \mathbf{x}_i) \leq 0.5$. Next we will show that for $j = y_i$, $F_j(\mathbf{W}_j^{(t,b)}, \mathbf{x}_i) \leq 0.5$ also holds.

According to Lemma D.3, we have

$$F_j(\mathbf{W}_j^{(t,b)}, \mathbf{x}_i) = \frac{1}{m}\sum_{r=1}^m [\sigma(\langle \mathbf{w}_{j,r}^{(t,b)}, y_i\boldsymbol{\mu} \rangle) + (P-1)\sigma(\langle \mathbf{w}_{j,r}^{(t)}, \boldsymbol{\xi}_i \rangle)]$$

$$\leq 2\max\{|\langle \mathbf{w}_{j,r}^{(t,b)}, y_i\boldsymbol{\mu} \rangle|, (P-1)|\langle \mathbf{w}_{j,r}^{(t)}, \boldsymbol{\xi}_i \rangle|\}$$

$$\leq 6\max\left\{ |\langle \mathbf{w}_{j,r}^{(0)}, \widehat{y}_i\boldsymbol{\mu} \rangle|, (P-1)|\langle \mathbf{w}_{j,r}^{(0)}, \boldsymbol{\xi}_i \rangle|, \mathrm{SNR}\sqrt{\frac{32\log(6n/\delta)}{d}} n\alpha, \right.$$

$$\left. 5\sqrt{\frac{\log(6n^2/\delta)}{d}} n\alpha, |\gamma_{j,r}^{(t,b)}|, |\underline{\rho}_{j,r,i}^{(t,b)}| \right\}$$

$$\leq 6\max\left\{ \beta, \mathrm{SNR}\sqrt{\frac{32\log(6n/\delta)}{d}} n\alpha, 5\sqrt{\frac{\log(6n^2/\delta)}{d}} n\alpha, |\gamma_{j,r}^{(t,b)}|, |\overline{\rho}_{j,r,i}^{(t,b)}| \right\}$$

$$\leq 0.5,$$

where the second inequality is by (64), (65) and (66); the third inequality is due to the definition of $\beta$; the last inequality is by (25), (61), (62).

Since $F_j(\mathbf{W}_j^{(t,b)}, \mathbf{x}_i) \in [0, 0.5]$ we know that

$$-0.3 \geq -\frac{1}{1 + \exp(0.5)} \geq \ell_i' \geq -\frac{1}{1 + \exp(-0.5)} \geq -0.7.$$

$\square$

Based on the previous foundation lemmas, we can provide the key lemma of SAM which is different from the dynamic of SGD.

**Lemma D.5.** *Under Assumption 3.1, suppose* (61)*,* (62) *and* (63) *hold at iteration $t, b$. We have that if $\langle \mathbf{w}_{j,r}^{(t,b)}, \boldsymbol{\xi}_k \rangle \geq 0$, $k \in \mathcal{I}_{t,b}$ and $j = y_k$, then $\langle \mathbf{w}_{j,r}^{(t,b)} + \widehat{\boldsymbol{\epsilon}}_{j,r}^{(t,b)}, \boldsymbol{\xi}_k \rangle < 0$.*

*Proof.* We first prove that there for $t \leq T_1$, there exists a constant $C_2$ such that

$$\|\nabla_{\mathbf{W}} L_{\mathcal{I}_{t,b}}(\mathbf{W}^{(t,b)})\|_F \leq C_2 P \sigma_p \sqrt{d/B}.$$

Recall that

$$L_{\mathcal{I}_{t,b}}(\mathbf{W}^{(t,b)}) = \frac{1}{B} \sum_{i \in \mathcal{I}_{t,b}} \ell(y_i f(\mathbf{W}^{(t,b)}, x_i)),$$

we have

$$\nabla_{\mathbf{w}_{j,r}} L_{\mathcal{I}_{t,b}}(\mathbf{W}^{(t,b)}) = \frac{1}{B} \sum_{i \in \mathcal{I}_{t,b}} \nabla_{\mathbf{w}_{j,r}} \ell(y_i f(\mathbf{W}^{(t,b)}, \mathbf{x}_i))$$

$$= \frac{1}{B} \sum_{i \in \mathcal{I}_{t,b}} y_i \ell'(y_i f(\mathbf{W}^{(t,b)}, \mathbf{x}_i)) \nabla_{\mathbf{w}_{j,r}} f(\mathbf{W}^{(t,b)}, \mathbf{x}_i)$$

$$= \frac{1}{Bm} \sum_{i \in \mathcal{I}_{t,b}} y_i \ell_i'^{(t,b)} [\sigma'(\langle \mathbf{w}_{j,r}^{(t,b)}, \boldsymbol{\mu} \rangle) \cdot \boldsymbol{\mu} + \sigma'(\langle \mathbf{w}_{j,r}^{(t,b)}, \boldsymbol{\xi}_i \rangle) \cdot (P-1)\boldsymbol{\xi}_i].$$

We have

$$\|\nabla_{\mathbf{w}_{j,r}} L_{\mathcal{I}_{t,b}}(\mathbf{W}^{(t,b)})\|_2$$

$$\leq \frac{1}{Bm} \left\| \sum_{i \in \mathcal{I}_{t,b}} |\ell_i'^{(t,b)}| \cdot \sigma'(\langle \mathbf{w}_{j,r}^{(t)}, \boldsymbol{\mu} \rangle) \cdot \boldsymbol{\mu} \right\|_2 + \frac{1}{Bm} \left\| \sum_{i \in \mathcal{I}_{t,b}} |\ell_i'^{(t,b)}| \cdot \sigma'(\langle \mathbf{w}_{j,r}^{(t)}, \boldsymbol{\xi}_i \rangle) \cdot (P-1)\boldsymbol{\xi}_i \right\|_2$$

$$\leq 0.7 m^{-1} \|\boldsymbol{\mu}\|_2 + 1.4(P-1)m^{-1}\sigma_p\sqrt{d/B}$$

$$\leq 2Pm^{-1}\sigma_p\sqrt{d/B},$$

and

$$\|\nabla_{\mathbf{W}} L_{\mathcal{I}_{t,b}}(\mathbf{W}^{(t,b)})\|_F^2 = \sum_{j,r} \|\nabla_{\mathbf{w}_{j,r}} L_{\mathcal{I}_{t,b}}(\mathbf{W}^{(t,b)})\|_2^2 \leq 2m(2Pm^{-1}\sigma_p\sqrt{d/B})^2,$$

leading to

$$\|\nabla_{\mathbf{W}} L_{\mathcal{I}_{t,b}}(\mathbf{W}^{(t,b)})\|_F \leq 2\sqrt{2}P\sigma_p\sqrt{d/Bm}.$$

From Lemma D.1, we have

$$\langle \widehat{\boldsymbol{\epsilon}}_{j,r}^{(t,b)}, \boldsymbol{\xi}_k \rangle = \frac{\tau}{mB} \|\nabla_{\mathbf{W}} L_{\mathcal{I}_{t,b}}(\mathbf{W}^{(t,b)})\|_F^{-1} \sum_{i \in \mathcal{I}_{t,b}} \sum_{p \in [P]} \ell_i'^{(t)} j \cdot y_i \sigma'(\langle \mathbf{w}_{j,r}^{(t)}, \mathbf{x}_{i,p} \rangle) \langle \mathbf{x}_{i,p}, \boldsymbol{\xi}_k \rangle$$

$$= \frac{\tau}{mB} \|\nabla_{\mathbf{W}} L_{\mathcal{I}_{t,b}}(\mathbf{W}^{(t,b)})\|_F^{-1} \cdot \left( \sum_{i \in \mathcal{I}_{t,b}, i \neq k} \ell_i'^{(t,b)} j y_i \cdot (P-1)\sigma'(\langle \mathbf{w}_{j,r}^{(t,b)}, \boldsymbol{\xi}_i \rangle) \langle \boldsymbol{\xi}_i, \boldsymbol{\xi}_k \rangle \right.$$

$$\left. + \ell_k'^{(t)} j y_k \cdot (P-1)\sigma'(\langle \mathbf{w}_{j,r}^{(t)}, \boldsymbol{\xi}_k \rangle) \langle \boldsymbol{\xi}_k, \boldsymbol{\xi}_k \rangle \right.$$

$$+ \sum_{i \in \mathcal{I}_{t,b}} \ell_i'^{(t,b)} j \cdot \sigma'(\langle \mathbf{w}_{j,r}^{(t,b)}, y_i \boldsymbol{\mu} \rangle \langle \boldsymbol{\mu}, \boldsymbol{\xi}_k \rangle)$$

$$\leq \frac{\tau}{mC_2 P \sigma_p \sqrt{Bd}} \left[ 0.8B(P-1)\sigma_P^2 \sqrt{d \log(6n^2/\delta)} + 0.4B\sigma_P \|\boldsymbol{\mu}\|_2 \sqrt{2\log(6n^2/\delta)} \right.$$

$$\left. - 0.15(P-1)\sigma_p^2 d \right]$$

$$< -C\frac{\tau \sigma_p \sqrt{d}}{m\sqrt{B}}$$

$$= -\frac{1}{4(P-1)}, \tag{67}$$

where we the last equality is by choosing $\tau = \frac{m\sqrt{B}}{C_3 P \sigma_p \sqrt{d}}$. Now we give an upper bound of $\langle \mathbf{w}_{j,r}^{(t)}, \boldsymbol{\xi}_k \rangle$, by (66) we have that

$$\langle \mathbf{w}_{j,r}^{(t)}, \boldsymbol{\xi}_k \rangle \leq 3 \max \left\{ |\langle \mathbf{w}_{j,r}^{(0)}, \boldsymbol{\xi}_i \rangle|, 5\sqrt{\frac{\log(6n^2/\delta)}{d}} n\alpha, |\overline{\rho}_{j,r,i}^{(t,b)}| \right\} \leq 1/(4(P-1)). \tag{68}$$

Combining (67) and (68) completes the proof. $\square$

**Lemma D.6.** *Under Assumption 3.1, suppose* (61), (62), (63) *hold at iteration* $t, b$. *Then* (62) *also holds for* $t, b+1$.

*Proof.* Now consider the SAM algorithm. Recall that

$$\overline{\rho}_{j,r,i}^{(t,b+1)} = \overline{\rho}_{j,r,i}^{(t,b)} - \frac{\eta(P-1)^2}{Bm} \cdot \ell_i'^{(t,b)} \cdot \sigma'(\langle \mathbf{w}_{j,r}^{(t)} + \widehat{\boldsymbol{\epsilon}}_{j,r}^{(t)}, \boldsymbol{\xi}_i \rangle) \cdot \|\boldsymbol{\xi}_i\|_2^2 \cdot \mathbb{1}(y_i = j) \mathbb{1}(i \in \mathcal{I}_{t,b}).$$

**Case 1:** $i \notin \mathcal{I}_{t,b}$. In this case, clearly we have that $\overline{\rho}_{j,r,i}^{(t,b+1)} = \overline{\rho}_{j,r,i}^{(t,b)} \leq 1/12$.

**Case 2:** $i \in \mathcal{I}_{t,b}$ and $\langle \mathbf{w}_{j,r}^{(t,b)}, \boldsymbol{\xi}_i \rangle \geq 0$, then by Lemma D.5, we have that $\langle \mathbf{w}_{j,r}^{(t)} + \widehat{\boldsymbol{\epsilon}}_{j,r}^{(t)}, \boldsymbol{\xi}_k \rangle < 0$, therefore we have that $\overline{\rho}_{j,r,i}^{(t,b+1)} = \overline{\rho}_{j,r,i}^{(t,b)} \leq 1/12$.

**Case 3:** $i \in \mathcal{I}_{t,b}$ and $\langle \mathbf{w}_{j,r}^{(t,b)}, \boldsymbol{\xi}_i \rangle \leq 0$ then by (66) and triangle inequality, we can conclude that $\overline{\rho}_{j,r,i}^{(t,b)}$ can not reach a constant order,

$$\overline{\rho}_{j,r,i}^{(t,b)} \leq (P-1) |\langle \mathbf{w}_{j,r}^{(t,b)} - \mathbf{w}_{j,r}^{(0,0)}, \boldsymbol{\xi}_i \rangle| + 5\sqrt{\frac{\log(6n^2/\delta)}{d}} n\alpha.$$

Then we can give an upper bound for $\overline{\rho}_{j,r,i}^{(t+1,b)}$ since we only take one small step further,

$$\overline{\rho}_{j,r,i}^{(t,b+1)} \leq (P-1) |\langle \mathbf{w}_{j,r}^{(t,b)} - \mathbf{w}_{j,r}^{(0,0)}, \boldsymbol{\xi}_i \rangle| + 5\sqrt{\frac{\log(6n^2/\delta)}{d}} n\alpha + \frac{\eta(P-1)^2}{Bm} \cdot 2d\sigma_p^2 \leq 1/12.$$

$\square$

*Proof of Proposition D.2.* We will use induction to give the proof. The results are obvious hold at $t = 0$ as all the coefficients are zero. Suppose that there exists $\widetilde{T} \leq T_1$ such that the results in Proposition D.2 hold for all time $(0,0) \leq (t,b) \leq (\widetilde{T}-1, \widetilde{b}-1)$. We aim to prove that (61), (62), (63) also hold for iteration $(\widetilde{T}-1, \widetilde{b})$.

First, we prove that (61) holds for hold for iteration $(\widetilde{T}-1, \widetilde{b})$. Notice that

$$\gamma_{j,r}^{(t,b+1)} = \gamma_{j,r}^{(t,b)} - \frac{\eta}{Bm} \cdot \sum_{i \in \mathcal{I}_{t,b}} \ell_i'^{(t,b)} \sigma'(\langle \mathbf{w}_{j,r}^{(t,b)}, y_i \cdot \boldsymbol{\mu} \rangle) \cdot \|\boldsymbol{\mu}\|_2^2 \leq \gamma_{j,r}^{(t,b)} + \frac{\eta}{m} \|\boldsymbol{\mu}\|_2^2,$$

where the last inequality is by the fact that $|\ell_i'^{(t,b+1)}| \leq 1$ and $\sigma' \leq 1$. Notice that $\widetilde{T} - 1 \leq T_1$, we can conclude that,

$$\gamma_{j,r}^{(\widetilde{T}, \widetilde{b})} \leq T_1 \cdot (n/B) \cdot \frac{\eta}{m} \|\boldsymbol{\mu}\|_2^2 \leq 1/12.$$

Second, by Lemma D.6, we know that (62) holds for $(\widetilde{T} - 1, \widetilde{b})$.

Last, we need to prove that (63) holds $(\widetilde{T} - 1, \widetilde{b})$. The proof is similar to previous proof without SAM.

When $\underline{\rho}_{j,r,k}^{(\widetilde{T}-1,\widetilde{b}-1)} < -0.5(P-1)\beta - 6\sqrt{\frac{\log(6n^2/\delta)}{d}}n\alpha$, by (31), we have

$$\langle \mathbf{w}_{j,r}^{(\widetilde{T}-1,\widetilde{b}-1)}, \boldsymbol{\xi}_k \rangle < \langle \mathbf{w}_{j,r}^{(0,0)}, \boldsymbol{\xi}_k \rangle + \frac{1}{P-1}\underline{\rho}_{j,r,k}^{(\widetilde{T}-1,\widetilde{b}-1)} + \frac{5}{P-1}\sqrt{\frac{\log(6n^2/\delta)}{d}}n\alpha$$

$$\leq -\frac{1}{P-1}\sqrt{\frac{\log(6n^2/\delta)}{d}}n\alpha,$$

and we have

$$\langle \widehat{\boldsymbol{\epsilon}}_{j,r}^{(\widetilde{T}-1,\widetilde{b}-1)}, \boldsymbol{\xi}_i \rangle = \frac{\tau}{mB}\|\nabla_{\mathbf{W}}L_{\mathcal{I}_{\widetilde{T}-1,\widetilde{b}-1}}(\mathbf{W}^{(\widetilde{T}-1,\widetilde{b}-1)})\|_F^{-1}\sum_{i\in\mathcal{I}_{\widetilde{T}-1,\widetilde{b}-1}}\sum_{p\in[P]}\ell_i^{\prime(\widetilde{T}-1,\widetilde{b}-1)}j\cdot y_i$$

$$\sigma^{\prime}(\langle \mathbf{w}_{j,r}^{(\widetilde{T}-1,\widetilde{b}-1)}, \mathbf{x}_{i,p}\rangle)\langle \mathbf{x}_{i,p}, \boldsymbol{\xi}_k\rangle$$

$$= \frac{\tau}{mB}\|\nabla_{\mathbf{W}}L_{\mathcal{I}_{\widetilde{T}-1,\widetilde{b}-1}}(\mathbf{W}^{(\widetilde{T}-1,\widetilde{b}-1)})\|_F^{-1}\cdot\Bigg(\sum_{i\in\mathcal{I}_{\widetilde{T}-1,\widetilde{b}-1},i\neq k}\ell_i^{\prime(\widetilde{T}-1,\widetilde{b}-1)}j\cdot y_i$$

$$(P-1)\sigma^{\prime}(\langle \mathbf{w}_{j,r}^{(\widetilde{T}-1,\widetilde{b}-1)}, \boldsymbol{\xi}_i\rangle)\langle \boldsymbol{\xi}_i, \boldsymbol{\xi}_k\rangle + \ell_k^{\prime(t)}jy_k\cdot(P-1)\sigma^{\prime}(\langle \mathbf{w}_{j,r}^{(t)}, \boldsymbol{\xi}_k\rangle)\langle \boldsymbol{\xi}_k, \boldsymbol{\xi}_k\rangle$$

$$+ \sum_{i\in\mathcal{I}_{\widetilde{T}-1,\widetilde{b}-1}}\ell_i^{\prime(\widetilde{T}-1,\widetilde{b}-1)}j\cdot\sigma^{\prime}(\langle \mathbf{w}_{j,r}^{(\widetilde{T}-1,\widetilde{b}-1)}, y_i\boldsymbol{\mu}\rangle\langle \boldsymbol{\mu}, \boldsymbol{\xi}_k\rangle)\Bigg)$$

$$\leq \frac{\tau}{mC_2P\sigma_p\sqrt{Bd}}\Bigg[0.8B(P-1)\sigma_P^2\sqrt{d\log(6n^2/\delta)} + 0.4B\sigma_P\|\boldsymbol{\mu}\|_2\sqrt{2\log(6n^2/\delta)}\Bigg]$$

$$\leq C_4\frac{\tau\sqrt{B}\sigma_p\sqrt{\log(6n^2/\delta)}}{m}$$

$$= C_4\frac{B\sqrt{\log(6n^2/\delta)}}{C_3P\sqrt{d}}$$

$$\leq \frac{1}{P}\sqrt{\frac{\log(6n^2/\delta)}{d}}n\alpha,$$

and thus $\langle \mathbf{w}_{j,r}^{(\widetilde{T}-1,\widetilde{b}-1)} + \widehat{\boldsymbol{\epsilon}}_{j,r}^{(\widetilde{T}-1,\widetilde{b}-1)}, \boldsymbol{\xi}_i \rangle < 0$ which leads to

$$\underline{\rho}_{j,r,i}^{(\widetilde{T}-1,\widetilde{b})} = \underline{\rho}_{j,r,i}^{(\widetilde{T}-1,\widetilde{b}-1)} + \frac{\eta(P-1)^2}{Bm}\cdot\ell_i^{\prime(\widetilde{T}-1,\widetilde{b}-1)}\cdot\sigma^{\prime}(\langle \mathbf{w}_{j,r}^{(\widetilde{T}-1,\widetilde{b}-1)}, \boldsymbol{\xi}_i\rangle)\cdot\|\boldsymbol{\xi}_i\|_2^2\cdot$$

$$\mathbb{1}(y_i = -j)\mathbb{1}(i\in\mathcal{I}_{\widetilde{T}-1,\widetilde{b}-1})$$

$$= \underline{\rho}_{j,r,i}^{(\widetilde{T}-1,\widetilde{b}-1)}.$$

Therefore, we have

$$\underline{\rho}_{j,r,i}^{(\widetilde{T}-1,\widetilde{b})} = \underline{\rho}_{j,r,i}^{(\widetilde{T}-1,\widetilde{b}-1)} \geq -(P-1)\beta - 5P\sqrt{\frac{\log(6n^2/\delta)}{d}}n\alpha.$$

When $\underline{\rho}_{j,r,i}^{(\widetilde{T}-1,\widetilde{b}-1)} \geq -0.5(P-1)\beta - 5\sqrt{\frac{\log(6n^2/\delta)}{d}}n\alpha$, we have that

$$\underline{\rho}_{j,r,i}^{(\widetilde{T}-1,\widetilde{b})} \geq \underline{\rho}_{j,r,i}^{(\widetilde{T}-1,\widetilde{b}-1)} + \frac{\eta(P-1)^2}{Bm}\cdot\ell_i^{\prime(\widetilde{T}-1,\widetilde{b}-1)}\cdot\|\boldsymbol{\xi}_i\|_2^2$$

$$\geq \underline{\rho}_{j,r,i}^{(\widetilde{T}-1,\widetilde{b}-1)} - \frac{0.4\eta(P-1)^2}{Bm}\cdot2d\sigma_p^2$$

$$\geq -(P-1)\beta - 5P\sqrt{\frac{\log(6n^2/\delta)}{d}}n\alpha.$$

Therefore, the induction is completed, and thus Proposition D.2 holds.

Next, we will prove that $\gamma_{j,r}^{(t)}$ can achieve $\Omega(1)$ after $T_1 = mB/(12n\eta\|\boldsymbol{\mu}\|_2^2)$ iterations. By Lemma B.6, we know that there exists $c_3 \cdot T_1$ epochs such that at least $c_4 \cdot H$ batches in these epochs, satisfy

$$|S_+ \cap S_y \cap \mathcal{I}_{t,b}| \in \left[\frac{B}{4}, \frac{3B}{4}\right]$$

for both $y = +1$ and $y = -1$. For SAM, we have the following update rule for $\gamma_{j,r}^{(t,b)}$:

$$\gamma_{j,r}^{(t,b+1)} = \gamma_{j,r}^{(t,b)} - \frac{\eta}{Bm} \sum_{i\in\mathcal{I}_{t,b}\cap S_+} \ell_i'^{(t,b)}\sigma'(\langle \mathbf{w}_{j,r}^{(t,b)} + \widehat{\boldsymbol{\epsilon}}_{j,r}^{(t,b)}, y_i \cdot \boldsymbol{\mu}\rangle) \cdot \|\boldsymbol{\mu}\|_2^2$$
$$+ \frac{\eta}{Bm} \sum_{i\in\mathcal{I}_{t,b}\cap S_-} \ell_i'^{(t,b)}\sigma'(\langle \mathbf{w}_{j,r}^{(t,b)} + \widehat{\boldsymbol{\epsilon}}_{j,r}^{(t,b)}, y_i \cdot \boldsymbol{\mu}\rangle) \cdot \|\boldsymbol{\mu}\|_2^2.$$

If $\langle \mathbf{w}_{j,r}^{(t,b)} + \widehat{\boldsymbol{\epsilon}}_{j,r}^{(t,b)}, \boldsymbol{\mu}\rangle \geq 0$, we have

$$\gamma_{j,r}^{(t,b+1)} = \gamma_{j,r}^{(t,b)} - \frac{\eta}{Bm} \cdot \left[\sum_{i\in\mathcal{I}_{t,b}\cap S_+\cap S_1} \ell_i'^{(t)} - \sum_{i\in\mathcal{I}_{t,b}\cap S_+\cap S_{-1}} \ell_i'^{(t)}\right]\|\boldsymbol{\mu}\|_2^2$$
$$\geq \gamma_{j,r}^{(t,b)} + \frac{\eta}{Bm} \cdot \left(0.3|\mathcal{I}_{t,b}\cap S_+\cap S_1| - 0.7|\mathcal{I}_{t,b}\cap S_+\cap S_{-1}|\right) \cdot \|\boldsymbol{\mu}\|_2^2.$$

If $\langle \mathbf{w}_{j,r}^{(t,b)} + \widehat{\boldsymbol{\epsilon}}_{j,r}^{(t,b)}, \boldsymbol{\mu}\rangle < 0$, we have

$$\gamma_{j,r}^{(t,b+1)} = \gamma_{j,r}^{(t,b)} + \frac{\eta}{Bm} \cdot \left[\sum_{i\in\mathcal{I}_{t,b}\cap S_+\cap S_{-1}} \ell_i'^{(t)} - \sum_{i\in\mathcal{I}_{t,b}\cap S_+\cap S_1} \ell_i'^{(t)}\right]\|\boldsymbol{\mu}\|_2^2$$
$$\geq \gamma_{j,r}^{(t,b)} + \frac{\eta}{Bm} \cdot \left(0.3|\mathcal{I}_{t,b}\cap S_+\cap S_{-1}| - 0.7|\mathcal{I}_{t,b}\cap S_+\cap S_1|\right) \cdot \|\boldsymbol{\mu}\|_2^2.$$

Therefore, we have

$$\gamma_{j,r}^{(T_1,0)} \geq \frac{\eta}{Bm}(0.3 \cdot c_3 T_1 \cdot c_4 H \cdot 0.25 B - 0.7 T_1 nq)\|\boldsymbol{\mu}\|_2^2$$
$$= \frac{\eta}{Bm}(0.075 c_3 c_4 T_1 n - 0.7 T_1 nq)\|\boldsymbol{\mu}\|_2^2$$
$$\geq \frac{\eta}{16Bm} c_3 c_4 T_1 n \|\boldsymbol{\mu}\|_2^2$$
$$= \frac{c_3 c_4}{192} = \Omega(1).$$

$\square$

**Lemma D.7.** *Suppose Condition 3.1 holds. Then we have that* $\left\|\mathbf{w}_{j,r}^{(T_1,0)}\right\|_2 = \Theta(\sigma_0\sqrt{d})$ *and*

$$\langle \mathbf{w}_{j,r}^{(T_1,0)}, j\boldsymbol{\mu}\rangle = \Omega(1),$$
$$\langle \mathbf{w}_{-j,r}^{(T_1,0)}, j\boldsymbol{\mu}\rangle = -\Omega(1),$$
$$\widehat{\beta} := 2\max_{i,j,r}\{|\langle \mathbf{w}_{j,r}^{(T_1,0)}, \boldsymbol{\mu}\rangle|, (P-1)|\langle \mathbf{w}_{j,r}^{(T_1,0)}, \boldsymbol{\xi}_i\rangle|\} = O(1).$$

*Besides, for* $S_i^{(t,b)}$ *and* $S_{j,r}^{(t,b)}$ *defined in Lemma B.3 and B.4, we have that*

$$|S_i^{(T_1,0)}| = \Omega(m), \forall i \in [n]$$
$$|S_{j,r}^{(T_1)}| = \Omega(n), \forall j \in \{\pm 1\}, r \in [m].$$

## D.2 Test Error Analysis

*Proof of Theorem 4.1.* Recall that

$$\mathbf{w}_{j,r}^{(t,b)} = \mathbf{w}_{j,r}^{(0,0)} + j \cdot \gamma_{j,r}^{(t,b)} \cdot \|\boldsymbol{\mu}\|_2^{-2} \cdot \boldsymbol{\mu} + \frac{1}{P-1} \sum_{i=1}^{n} \rho_{j,r,i}^{(t,b)} \cdot \|\boldsymbol{\xi}_i\|_2^{-2} \cdot \boldsymbol{\xi}_i,$$

by triangle inequality we have

$$\left| \left\|\mathbf{w}_{j,r}^{(T_1,0)}\right\|_2 - \left\|\mathbf{w}_{j,r}^{(0,0)}\right\|_2 \right| \leq |\gamma_{j,r}^{(t,b)}| \cdot \|\boldsymbol{\mu}\|_2^{-1} + \frac{1}{P-1} \left\| \sum_{i=1}^{n} |\rho_{j,r,i}^{(t,b)}| \cdot \|\boldsymbol{\xi}_i\|_2^{-2} \cdot \boldsymbol{\xi}_i \right\|_2$$

$$\leq \frac{1}{12} \|\boldsymbol{\mu}\|_2^{-1} + \frac{\sqrt{n}}{12(P-1)} (\sigma_p^2 d/2)^{-1/2}$$

$$\leq \frac{1}{6} \|\boldsymbol{\mu}\|_2^{-1}.$$

By the condition on $\sigma_0$ and Lemma B.2, we have

$$\left\|\mathbf{w}_{j,r}^{(T_1,0)}\right\|_2 = \Theta(\left\|\mathbf{w}_{j,r}^{(0,0)}\right\|_2) = \Theta(\sigma_0 \sqrt{d}).$$

By taking the inner product with $\boldsymbol{\mu}$ and $\boldsymbol{\xi}_i$, we can get

$$\langle \mathbf{w}_{j,r}^{(t,b)}, \boldsymbol{\mu} \rangle = \langle \mathbf{w}_{j,r}^{(0,0)}, \boldsymbol{\mu} \rangle + j \cdot \gamma_{j,r}^{(t,b)} + \frac{1}{P-1} \sum_{i=1}^{n} \rho_{j,r,i}^{(t,b)} \cdot \|\boldsymbol{\xi}_i\|_2^{-2} \cdot \langle \boldsymbol{\xi}_i, \boldsymbol{\mu} \rangle,$$

and

$$\langle \mathbf{w}_{j,r}^{(t,b)}, \boldsymbol{\xi}_i \rangle = \langle \mathbf{w}_{j,r}^{(0,0)}, \boldsymbol{\xi}_i \rangle + j \cdot \gamma_{j,r}^{(t,b)} \cdot \|\boldsymbol{\mu}\|_2^{-2} \cdot \langle \boldsymbol{\mu}, \boldsymbol{\xi}_i \rangle + \frac{1}{P-1} \sum_{i'=1}^{n} \rho_{j,r,i'}^{(t,b)} \cdot \|\boldsymbol{\xi}_{i'}\|_2^{-2} \cdot \langle \boldsymbol{\xi}_{i'}, \boldsymbol{\xi}_i \rangle$$

$$= \langle \mathbf{w}_{j,r}^{(0,0)}, \boldsymbol{\xi}_i \rangle + j \cdot \gamma_{j,r}^{(t,b)} \cdot \|\boldsymbol{\mu}\|_2^{-2} \cdot \langle \boldsymbol{\mu}, \boldsymbol{\xi}_i \rangle + \frac{1}{P-1} \rho_{j,r,i}^{(t,b)}$$

$$+ \frac{1}{P-1} \sum_{i \neq i'} \rho_{j,r,i'}^{(t,b)} \cdot \|\boldsymbol{\xi}_{i'}\|_2^{-2} \cdot \langle \boldsymbol{\xi}_{i'}, \boldsymbol{\xi}_i \rangle.$$

Then, we have

$$\langle \mathbf{w}_{j,r}^{(T_1,0)}, j\boldsymbol{\mu} \rangle = \langle \mathbf{w}_{j,r}^{(0,0)}, j\boldsymbol{\mu} \rangle + \gamma_{j,r}^{(T_1,0)} + \frac{1}{P-1} \sum_{i=1}^{n} \rho_{j,r,i}^{(T_1,0)} \cdot \|\boldsymbol{\xi}_i\|_2^{-2} \cdot \langle \boldsymbol{\xi}_i, j\boldsymbol{\mu} \rangle$$

$$\geq \gamma_{j,r}^{(T_1,0)} - |\langle \mathbf{w}_{j,r}^{(0,0)}, \boldsymbol{\mu} \rangle| - \frac{1}{P-1} \sum_{i=1}^{n} |\rho_{j,r,i}^{(T_1,0)}| \cdot \|\boldsymbol{\xi}_i\|_2^{-2} \cdot |\langle \boldsymbol{\xi}_i, \boldsymbol{\mu} \rangle|$$

$$\geq \gamma_{j,r}^{(T_1,0)} - \sqrt{2\log(12m/\delta)} \cdot \sigma_0 \|\boldsymbol{\mu}\|_2 - \frac{n}{12(P-1)} (\sigma_0^2 d/2)^{-1} \|\boldsymbol{\mu}\|_2 \sigma_p \cdot \sqrt{2\log(6n/\delta)}$$

$$\geq \frac{1}{2} \gamma_{j,r}^{(T_1,0)},$$

and

$$\langle \mathbf{w}_{-j,r}^{(T_1,0)}, j\boldsymbol{\mu} \rangle = \langle \mathbf{w}_{-j,r}^{(0,0)}, j\boldsymbol{\mu} \rangle - \gamma_{-j,r}^{(T_1,0)} - \frac{1}{P-1} \sum_{i=1}^{n} \rho_{-j,r,i}^{(T_1,0)} \cdot \|\boldsymbol{\xi}_i\|_2^{-2} \cdot \langle \boldsymbol{\xi}_i, j\boldsymbol{\mu} \rangle$$

$$\leq -\gamma_{-j,r}^{(T_1,0)} + |\langle \mathbf{w}_{-j,r}^{(0,0)}, \boldsymbol{\mu} \rangle| + \frac{1}{P-1} \sum_{i=1}^{n} |\rho_{-j,r,i}^{(T_1,0)}| \cdot \|\boldsymbol{\xi}_i\|_2^{-2} \cdot |\langle \boldsymbol{\xi}_i, \boldsymbol{\mu} \rangle|$$

$$\leq -\gamma_{-j,r}^{(T_1,0)} + \sqrt{2\log(12m/\delta)} \cdot \sigma_0 \|\boldsymbol{\mu}\|_2 + \frac{n}{12(P-1)} (\sigma_0^2 d/2)^{-1} \|\boldsymbol{\mu}\|_2 \sigma_p \cdot \sqrt{2\log(6n/\delta)}$$

$$\leq -\frac{1}{2} \gamma_{j,r}^{(T_1,0)},$$

where the last inequality is by the condition on $\sigma_0$ and $\gamma_{j,r}^{(T_1,0)} = \Omega(1)$. Thus, it follows that

$$\langle \mathbf{w}_{j,r}^{(T_1,0)}, j\boldsymbol{\mu} \rangle = \Omega(1), \ \langle \mathbf{w}_{-j,r}^{(T_1,0)}, j\boldsymbol{\mu} \rangle = -\Omega(1).$$

By triangle inequality, we have

$$|\langle \mathbf{w}_{j,r}^{(T_1,0)}, \boldsymbol{\mu} \rangle| \leq |\langle \mathbf{w}_{j,r}^{(0,0)}, \boldsymbol{\mu} \rangle| + |\gamma_{j,r}^{(T_1,0)}| + \frac{1}{P-1} \sum_{i=1}^{n} |\rho_{j,r,i}^{(t,b)}| \cdot \|\boldsymbol{\xi}_i\|_2^{-2} \cdot |\langle \boldsymbol{\xi}_i, \boldsymbol{\mu} \rangle|$$

$$\leq \frac{1}{2}\beta + \frac{1}{12} + \frac{n}{P-1} \cdot \frac{1}{12} (\sigma_p^2 d/2)^{-1} \cdot \|\boldsymbol{\mu}\|_2 \sigma_p \cdot \sqrt{2\log(6n/\delta)}$$

$$= \frac{1}{2}\beta + \frac{1}{12} + \frac{n}{6(P-1)} \|\boldsymbol{\mu}\|_2 \sqrt{2\log(6n/\delta)}/(\sigma_p d)$$

$$\leq \frac{1}{6},$$

and

$$|\langle \mathbf{w}_{j,r}^{(T_1,0)}, \boldsymbol{\xi}_i \rangle| \leq |\langle \mathbf{w}_{j,r}^{(0,0)}, \boldsymbol{\xi}_i \rangle| + |\gamma_{j,r}^{(T_1,0)}| \cdot \|\boldsymbol{\mu}\|_2^{-2} \cdot |\langle \boldsymbol{\mu}, \boldsymbol{\xi}_i \rangle| + \frac{1}{P-1} |\rho_{j,r,i}^{(T_1,0)}|$$

$$+ \frac{1}{P-1} \sum_{i \neq i'} |\rho_{j,r,i'}^{(T_1,0)}| \cdot \|\boldsymbol{\xi}_{i'}\|_2^{-2} \cdot |\langle \boldsymbol{\xi}_{i'}, \boldsymbol{\xi}_i \rangle|$$

$$\leq \frac{1}{2}\beta + \frac{1}{12} \|\boldsymbol{\mu}\|_2^{-1} \sigma_p \cdot \sqrt{2\log(6n/\delta)} + \frac{1}{12(P-1)}$$

$$+ \frac{n}{12(P-1)} (\sigma_p^2 d/2)^{-1} 2\sigma_p^2 \cdot \sqrt{d\log(6n^2/\delta)}$$

$$\leq \frac{1}{2}\beta + \frac{1}{12(P-1)} + \frac{1}{6} \|\boldsymbol{\mu}\|_2^{-1} \sigma_p \cdot \sqrt{\log(6n/\delta)}$$

$$\leq \frac{1}{6}.$$

This leads to

$$\widehat{\beta} := 2 \max_{i,j,r} \{ |\langle \mathbf{w}_{j,r}^{(T_1,0)}, \boldsymbol{\mu} \rangle|, (P-1)|\langle \mathbf{w}_{j,r}^{(T_1,0)}, \boldsymbol{\xi}_i \rangle| \} = O(1).$$

In addition, we also have for $t \leq T_1$ and $j = y_i$ that

$$\langle \mathbf{w}_{j,r}^{(t,b)}, \boldsymbol{\xi}_i \rangle - \langle \mathbf{w}_{j,r}^{(0,0)}, \boldsymbol{\xi}_i \rangle$$

$$\geq \frac{1}{P-1} \rho_{j,r,i}^{(t,b)} - \gamma_{j,r}^{(t,b)} \cdot \|\boldsymbol{\mu}\|_2^{-2} \cdot |\langle \boldsymbol{\mu}, \boldsymbol{\xi}_i \rangle| - \frac{1}{P-1} \sum_{i \neq i'} |\rho_{j,r,i'}^{(t,b)}| \cdot \|\boldsymbol{\xi}_{i'}\|_2^{-2} \cdot |\langle \boldsymbol{\xi}_{i'}, \boldsymbol{\xi}_i \rangle|$$

$$\geq -\gamma_{j,r}^{(t,b)} \cdot \|\boldsymbol{\mu}\|_2^{-2} \cdot |\langle \boldsymbol{\mu}, \boldsymbol{\xi}_i \rangle| - \frac{1}{P-1} \sum_{i \neq i'} |\rho_{j,r,i'}^{(t,b)}| \cdot \|\boldsymbol{\xi}_{i'}\|_2^{-2} \cdot |\langle \boldsymbol{\xi}_{i'}, \boldsymbol{\xi}_i \rangle|$$

$$\geq -\frac{1}{12} \|\boldsymbol{\mu}\|_2^{-2} \cdot |\langle \boldsymbol{\mu}, \boldsymbol{\xi}_i \rangle| - \frac{n}{12(P-1)} \|\boldsymbol{\xi}_{i'}\|_2^{-2} \cdot |\langle \boldsymbol{\xi}_{i'}, \boldsymbol{\xi}_i \rangle|$$

$$\geq -\frac{1}{12} \|\boldsymbol{\mu}\|_2^{-1} \sigma_p \cdot \sqrt{2\log(6n/\delta)} - \frac{n}{12(P-1)} (\sigma_p^2 d/2)^{-1} 2\sigma_p^2 \cdot \sqrt{d\log(6n^2/\delta)}$$

$$= -\frac{1}{12} \|\boldsymbol{\mu}\|_2^{-1} \sigma_p \cdot \sqrt{2\log(6n/\delta)} - \frac{n}{3(P-1)} \sqrt{\log(6n^2/\delta)/d}$$

$$\geq -\frac{1}{6} \|\boldsymbol{\mu}\|_2^{-1} \sigma_p \cdot \sqrt{\log(6n/\delta)}.$$

Now let $\bar{S}_i^{(0,0)}$ denote $\{r : \langle \mathbf{w}_{y_i,r}^{(0,0)}, \boldsymbol{\xi}_i \rangle > \sigma_0 \sigma_p \sqrt{d} \}$ and let $\bar{S}_{j,r}^{(0,0)}$ denote $\{i \in [n] : y_i = j, \langle \mathbf{w}_{y_i,r}^{(t,b)}, \boldsymbol{\xi}_i \rangle > \sigma_0 \sigma_p \sqrt{d} \}$. By the condition on $\sigma_0$, we have for $t \leq T_1$ that

$$\langle \mathbf{w}_{j,r}^{(t,b)}, \boldsymbol{\xi}_i \rangle \geq \frac{1}{\sqrt{2}} \langle \mathbf{w}_{j,r}^{(0,0)}, \boldsymbol{\xi}_i \rangle,$$

for any $r \in \bar{S}_i^{(0,0)}$ or $i \in \bar{S}_{j,r}^{(0,0)}$. Therefore, we have $\bar{S}_i^{(0,0)} \subseteq S_i^{(T_1,0)}$ and $\bar{S}_{j,r}^{(0,0)} \subseteq S_{j,r}^{(T_1,0)}$ and hence

$$0.8\Phi(-\sqrt{2})m \le |\bar{S}_i^{(0,0)}| \le |S_i^{(T_1,0)}| = \Omega(m),$$
$$0.25\Phi(-\sqrt{2})n \le |\bar{S}_{j,r}^{(0,0)}| \le |S_{j,r}^{(T_1,0)}| = \Omega(n),$$

where $\Phi(\cdot)$ is the CDF of the standard normal distribution. $\qquad\square$

Now we can give proof of Theorem 4.

*Proof of Theorem 4.* After the training process of SAM after $T_1$, we get $\mathbf{W}^{(T_1,0)}$. To differentiate the SAM process and SGD process. We use $\widetilde{\mathbf{W}}$ to denote the trajectory obtained by SAM in the proof, i.e., $\widetilde{\mathbf{W}}^{(T_1,0)}$. By Proposition D.2, we have that

$$\widetilde{\mathbf{w}}_{j,r}^{(T_1,0)} = \widetilde{\mathbf{w}}_{j,r}^{(0,0)} + j \cdot \widetilde{\gamma}_{j,r}^{(T_1,0)} \cdot \frac{\boldsymbol{\mu}}{\|\boldsymbol{\mu}\|_2^2} + \frac{1}{P-1}\sum_{i=1}^n \widetilde{\overline{\rho}}_{j,r,i}^{(T_1,0)} \cdot \frac{\boldsymbol{\xi}_i}{\|\boldsymbol{\xi}_i\|_2^2} + \frac{1}{P-1}\sum_{i=1}^n \widetilde{\underline{\rho}}_{j,r,i}^{(T_1,0)} \cdot \frac{\boldsymbol{\xi}_i}{\|\boldsymbol{\xi}_i\|_2^2}, \tag{69}$$

where $\widetilde{\gamma}_{j,r}^{(T_1,0)} = \Theta(1)$, $\widetilde{\overline{\rho}}_{j,r,i}^{(T_1,0)} \in [0,1/12]$, $\widetilde{\underline{\rho}}_{j,r,i}^{(T_1,0)} \in [-\beta - 10\sqrt{\log(6n^2/\delta)/d}n, 0]$. Then the SGD start at $\mathbf{W}^{(0,0)} := \widetilde{\mathbf{W}}^{(T_1,0)}$. Notice that by Lemma D.7, we know that the initial weights of SGD (i.e., the end weight of SAM) $\mathbf{W}^{(0,0)}$ still satisfies the conditions for Subsection C.1 and C.2. Therefore, following the same analysis in Subsection C.1 and C.2, we have that there exist $t = \widetilde{O}(\eta^{-1}\epsilon^{-1}mnd^{-1}P^{-2}\sigma_p^{-2})$ such that $L_S(\mathbf{W}^{(t,0)}) \le \epsilon$. Besides,

$$\mathbf{w}_{j,r}^{(t,0)} = \mathbf{w}_{j,r}^{(0,0)} + j \cdot \gamma_{j,r}^{(t,0)} \cdot \frac{\boldsymbol{\mu}}{\|\boldsymbol{\mu}\|_2^2} + \frac{1}{P-1}\sum_{i=1}^n \overline{\rho}_{j,r,i}^{(t,0)} \cdot \frac{\boldsymbol{\xi}_i}{\|\boldsymbol{\xi}_i\|_2^2} + \frac{1}{P-1}\sum_{i=1}^n \underline{\rho}_{j,r,i}^{(t,0)} \cdot \frac{\boldsymbol{\xi}_i}{\|\boldsymbol{\xi}_i\|_2^2}$$

for $j \in [\pm 1]$ and $r \in [m]$ where

$$\gamma_{j,r}^{(t,0)} = \Theta(\text{SNR}^2)\sum_{i\in[n]} \overline{\rho}_{j,r,i}^{(t,0)}, \quad \overline{\rho}_{j,r,i}^{(t,0)} \in [0,\alpha], \quad \underline{\rho}_{j,r,i}^{(t,0)} \in [-\alpha,0]. \tag{70}$$

Next, we will evaluate the test error for $\mathbf{W}^{(t,0)}$. Notice that we use $(t)$ as the shorthand notation of $(t,0)$. For the sake of convenience, we use $(\mathbf{x}, \widehat{y}, y) \sim \mathcal{D}$ to denote the following: data point $(\mathbf{x}, y)$ follows distribution $\mathcal{D}$ defined in Definition 2.1, and $\widehat{y}$ is its true label. We can write out the test error as

$$\begin{aligned}
&\mathbb{P}_{(\mathbf{x},y)\sim\mathcal{D}}\big(y \ne \text{sign}(f(\mathbf{W}^{(t)},\mathbf{x}))\big) \\
&= \mathbb{P}_{(\mathbf{x},y)\sim\mathcal{D}}\big(yf(\mathbf{W}^{(t)},\mathbf{x}) \le 0\big) \\
&= \mathbb{P}_{(\mathbf{x},y)\sim\mathcal{D}}\big(yf(\mathbf{W}^{(t)},\mathbf{x}) \le 0, y \ne \widehat{y}\big) + \mathbb{P}_{(\mathbf{x},\widehat{y},y)\sim\mathcal{D}}\big(yf(\mathbf{W}^{(t)},\mathbf{x}) \le 0, y = \widehat{y}\big) \quad (71) \\
&= p \cdot \mathbb{P}_{(\mathbf{x},\widehat{y},y)\sim\mathcal{D}}\big(\widehat{y}f(\mathbf{W}^{(t)},\mathbf{x}) \ge 0\big) + (1-p) \cdot \mathbb{P}_{(\mathbf{x},\widehat{y},y)\sim\mathcal{D}}\big(\widehat{y}f(\mathbf{W}^{(t)},\mathbf{x}) \le 0\big) \\
&\le p + \mathbb{P}_{(\mathbf{x},\widehat{y},y)\sim\mathcal{D}}\big(\widehat{y}f(\mathbf{W}^{(t)},\mathbf{x}) \le 0\big),
\end{aligned}$$

where in the second equation we used the definition of $\mathcal{D}$ in Definition 2.1. It therefore suffices to provide an upper bound for $\mathbb{P}_{(\mathbf{x},\widehat{y})\sim\mathcal{D}}\big(\widehat{y}f(\mathbf{W}^{(t)},\mathbf{x}) \le 0\big)$. To achieve this, we write $\mathbf{x} = (\widehat{y}\boldsymbol{\mu}, \boldsymbol{\xi})$, and get

$$\begin{aligned}
\widehat{y}f(\mathbf{W}^{(t)},\mathbf{x}) &= \frac{1}{m}\sum_{j,r}\widehat{y}j[\sigma(\langle\mathbf{w}_{j,r}^{(t)},\widehat{y}\boldsymbol{\mu}\rangle) + \sigma(\langle\mathbf{w}_{j,r}^{(t)},\boldsymbol{\xi}\rangle)] \\
&= \frac{1}{m}\sum_r[\sigma(\langle\mathbf{w}_{\widehat{y},r}^{(t)},\widehat{y}\boldsymbol{\mu}\rangle) + (P-1)\sigma(\langle\mathbf{w}_{\widehat{y},r}^{(t)},\boldsymbol{\xi}\rangle)] \\
&\quad - \frac{1}{m}\sum_r[\sigma(\langle\mathbf{w}_{-\widehat{y},r}^{(t)},\widehat{y}\boldsymbol{\mu}\rangle) + (P-1)\sigma(\langle\mathbf{w}_{-\widehat{y},r}^{(t)},\boldsymbol{\xi}\rangle)]. \quad (72)
\end{aligned}$$

The inner product with $j = \widehat{y}$ can be bounded as

$$\langle \mathbf{w}_{\widehat{y},r}^{(t)}, \widehat{y}\boldsymbol{\mu} \rangle = \langle \mathbf{w}_{\widehat{y},r}^{(0)}, \widehat{y}\boldsymbol{\mu} \rangle + \gamma_{\widehat{y},r}^{(t)} + \frac{1}{(P-1)} \sum_{i=1}^{n} \overline{\rho}_{\widehat{y},r,i}^{(t)} \cdot \|\boldsymbol{\xi}_i\|_2^{-2} \cdot \langle \boldsymbol{\xi}_i, \widehat{y}\boldsymbol{\mu} \rangle$$

$$+ \frac{1}{(P-1)} \sum_{i=1}^{n} \underline{\rho}_{\widehat{y},r,i}^{(t)} \cdot \|\boldsymbol{\xi}_i\|_2^{-2} \cdot \langle \boldsymbol{\xi}_i, \widehat{y}\boldsymbol{\mu} \rangle$$

$$\geq \langle \mathbf{w}_{\widehat{y},r}^{(0)}, \widehat{y}\boldsymbol{\mu} \rangle + \gamma_{\widehat{y},r}^{(t)} - \frac{\sqrt{2\log(6n/\delta)}}{P-1} \cdot \sigma_p \|\boldsymbol{\mu}\|_2 \cdot (\sigma_p^2 d/2)^{-1} \left[ \sum_{i=1}^{n} \overline{\rho}_{\widehat{y},r,i}^{(t)} + \sum_{i=1}^{n} |\underline{\rho}_{\widehat{y},r,i}^{(t)}| \right]$$

$$= \langle \mathbf{w}_{\widehat{y},r}^{(0)}, \widehat{y}\boldsymbol{\mu} \rangle + \gamma_{\widehat{y},r}^{(t)} - \Theta\big(\sqrt{\log(n/\delta)} \cdot (P\sigma_p d)^{-1} \|\boldsymbol{\mu}\|_2\big) \cdot \Theta(\mathrm{SNR}^{-2}) \cdot \gamma_{\widehat{y},r}^{(t)}$$

$$= \langle \mathbf{w}_{\widehat{y},r}^{(0)}, \widehat{y}\boldsymbol{\mu} \rangle + \big[1 - \Theta\big(\sqrt{\log(n/\delta)} \cdot P\sigma_p/\|\boldsymbol{\mu}\|_2\big)\big] \gamma_{\widehat{y},r}^{(t)}$$

$$= \langle \mathbf{w}_{\widehat{y},r}^{(0)}, \widehat{y}\boldsymbol{\mu} \rangle + \Theta(\gamma_{\widehat{y},r}^{(t)})$$

$$= \Omega(1),$$

$$(73)$$

where the inequality is by Lemma B.1; the second equality is obtained by plugging in the coefficient orders we summarized at (70); the third equality is by the condition $\mathrm{SNR} = \|\boldsymbol{\mu}\|_2/P\sigma_p\sqrt{d}$; the fourth equality is due to $\|\boldsymbol{\mu}\|_2^2 \geq C \cdot P^2 \sigma_p^2 \log(n/\delta)$ in Condition 3.1, so for sufficiently large constant $C$ the equality holds; the last equality is by Lemma D.7. Moreover, we can deduce in a similar manner that

$$\langle \mathbf{w}_{-\widehat{y},r}^{(t)}, \widehat{y}\boldsymbol{\mu} \rangle = \langle \mathbf{w}_{-\widehat{y},r}^{(0)}, \widehat{y}\boldsymbol{\mu} \rangle - \gamma_{-\widehat{y},r}^{(t)} + \sum_{i=1}^{n} \overline{\rho}_{-\widehat{y},r,i}^{(t)} \cdot \|\boldsymbol{\xi}_i\|_2^{-2} \cdot \langle \boldsymbol{\xi}_i, -\widehat{y}\boldsymbol{\mu} \rangle$$

$$+ \sum_{i=1}^{n} \underline{\rho}_{-\widehat{y},r,i}^{(t)} \cdot \|\boldsymbol{\xi}_i\|_2^{-2} \cdot \langle \boldsymbol{\xi}_i, \widehat{y}\boldsymbol{\mu} \rangle$$

$$\leq \langle \mathbf{w}_{-\widehat{y},r}^{(0)}, \widehat{y}\boldsymbol{\mu} \rangle - \gamma_{-\widehat{y},r}^{(t)} \qquad (74)$$

$$+ \sqrt{2\log(6n/\delta)} \cdot \sigma_p \|\boldsymbol{\mu}\|_2 \cdot (\sigma_p^2 d/2)^{-1} \left[ \sum_{i=1}^{n} \overline{\rho}_{-\widehat{y},r,i}^{(t)} + \sum_{i=1}^{n} |\underline{\rho}_{-\widehat{y},r,i}^{(t)}| \right]$$

$$= \langle \mathbf{w}_{-\widehat{y},r}^{(0)}, \widehat{y}\boldsymbol{\mu} \rangle - \Theta(\gamma_{-\widehat{y},r}^{(t)})$$

$$= -\Omega(1) < 0,$$

where the second equality holds based on similar analyses as in (73).

Denote $g(\boldsymbol{\xi})$ as $\sum_r \sigma(\langle \mathbf{w}_{-\widehat{y},r}^{(t)}, \boldsymbol{\xi} \rangle)$. According to Theorem 5.2.2 in Vershynin (2018), we know that for any $x \geq 0$ it holds that

$$\mathbb{P}(g(\boldsymbol{\xi}) - \mathbb{E}g(\boldsymbol{\xi}) \geq x) \leq \exp\Big(-\frac{cx^2}{\sigma_p^2 \|g\|_{\mathrm{Lip}}^2}\Big), \qquad (75)$$

where $c$ is a constant. To calculate the Lipschitz norm, we have

$$|g(\boldsymbol{\xi}) - g(\boldsymbol{\xi}')| = \left| \sum_{r=1}^{m} \sigma(\langle \mathbf{w}_{-\widehat{y},r}^{(t)}, \boldsymbol{\xi} \rangle) - \sum_{r=1}^{m} \sigma(\langle \mathbf{w}_{-\widehat{y},r}^{(t)}, \boldsymbol{\xi}' \rangle) \right|$$

$$\leq \sum_{r=1}^{m} \left| \sigma(\langle \mathbf{w}_{-\widehat{y},r}^{(t)}, \boldsymbol{\xi} \rangle) - \sigma(\langle \mathbf{w}_{-\widehat{y},r}^{(t)}, \boldsymbol{\xi}' \rangle) \right|$$

$$\leq \sum_{r=1}^{m} |\langle \mathbf{w}_{-\widehat{y},r}^{(t)}, \boldsymbol{\xi} - \boldsymbol{\xi}' \rangle|$$

$$\leq \sum_{r=1}^{m} \left\| \mathbf{w}_{-\widehat{y},r}^{(t)} \right\|_2 \cdot \|\boldsymbol{\xi} - \boldsymbol{\xi}'\|_2,$$

where the first inequality is by triangle inequality, the second inequality is by the property of ReLU; and the last inequality is by Cauchy-Schwartz inequality. Therefore, we have

$$\|g\|_{\text{Lip}} \le \sum_{r=1}^{m} \|\mathbf{w}_{-\widehat{y},r}^{(t)}\|_2, \tag{76}$$

and since $\langle \mathbf{w}_{-\widehat{y},r}^{(t)}, \boldsymbol{\xi} \rangle \sim \mathcal{N}\big(0, \|\mathbf{w}_{-\widehat{y},r}^{(t)}\|_2^2 \sigma_p^2\big)$, we can get

$$\mathbb{E}g(\boldsymbol{\xi}) = \sum_{r=1}^{m} \mathbb{E}\sigma(\langle \mathbf{w}_{-\widehat{y},r}^{(t)}, \boldsymbol{\xi} \rangle) = \sum_{r=1}^{m} \frac{\|\mathbf{w}_{-\widehat{y},r}^{(t)}\|_2 \sigma_p}{\sqrt{2\pi}} = \frac{\sigma_p}{\sqrt{2\pi}} \sum_{r=1}^{m} \|\mathbf{w}_{-\widehat{y},r}^{(t)}\|_2.$$

Next we seek to upper bound the 2-norm of $\mathbf{w}_{j,r}^{(t)}$. First, we tackle the noise section in the decomposition, namely:

$$\left\| \sum_{i=1}^{n} \rho_{j,r,i}^{(t)} \cdot \|\boldsymbol{\xi}_i\|_2^{-2} \cdot \boldsymbol{\xi}_i \right\|_2^2$$

$$= \sum_{i=1}^{n} \rho_{j,r,i}^{(t)}{}^2 \cdot \|\boldsymbol{\xi}_i\|_2^{-2} + 2 \sum_{1 \le i_1 < i_2 \le n} \rho_{j,r,i_1}^{(t)} \rho_{j,r,i_2}^{(t)} \cdot \|\boldsymbol{\xi}_{i_1}\|_2^{-2} \cdot \|\boldsymbol{\xi}_{i_2}\|_2^{-2} \cdot \langle \boldsymbol{\xi}_{i_1}, \boldsymbol{\xi}_{i_2} \rangle$$

$$\le 4\sigma_p^{-2}d^{-1} \sum_{i=1}^{n} \rho_{j,r,i}^{(t)}{}^2 + 2 \sum_{1 \le i_1 < i_2 \le n} \left| \rho_{j,r,i_1}^{(t)} \rho_{j,r,i_2}^{(t)} \right| \cdot (16\sigma_p^{-4}d^{-2}) \cdot (2\sigma_p^2 \sqrt{d\log(6n^2/\delta)})$$

$$= 4\sigma_p^{-2}d^{-1} \sum_{i=1}^{n} \rho_{j,r,i}^{(t)}{}^2 + 32\sigma_p^{-2}d^{-3/2}\sqrt{\log(6n^2/\delta)}\left[ \left( \sum_{i=1}^{n} |\rho_{j,r,i}^{(t)}| \right)^2 - \sum_{i=1}^{n} \rho_{j,r,i}^{(t)}{}^2 \right]$$

$$= \Theta(\sigma_p^{-2}d^{-1}) \sum_{i=1}^{n} \rho_{j,r,i}^{(t)}{}^2 + \widetilde{\Theta}(\sigma_p^{-2}d^{-3/2}) \left( \sum_{i=1}^{n} |\rho_{j,r,i}^{(t)}| \right)^2$$

$$\le \left[ \Theta(\sigma_p^{-2}d^{-1}n^{-1}) + \widetilde{\Theta}(\sigma_p^{-2}d^{-3/2}) \right] \left( \sum_{i=1}^{n} |\overline{\rho}_{j,r,i}^{(t)}| + \sum_{i=1}^{n} |\underline{\rho}_{j,r,i}^{(t)}| \right)^2$$

$$\le \Theta(\sigma_p^{-2}d^{-1}n^{-1}) \left( \sum_{i=1}^{n} \overline{\rho}_{j,r,i}^{(t)} \right)^2,$$

where for the first inequality, we used Lemma B.1; for the second inequality, we used the definition of $\overline{\rho}, \rho$; for the second to last equation, we plugged in coefficient orders. We can thus upper bound the 2-norm of $\mathbf{w}_{j,r}^{(t)}$ as:

$$\|\mathbf{w}_{j,r}^{(t)}\|_2 \le \|\mathbf{w}_{j,r}^{(0)}\|_2 + \gamma_{j,r}^{(t)} \cdot \|\boldsymbol{\mu}\|_2^{-1} + \frac{1}{P-1} \left\| \sum_{i=1}^{n} \rho_{j,r,i}^{(t)} \cdot \|\boldsymbol{\xi}_i\|_2^{-2} \cdot \boldsymbol{\xi}_i \right\|_2$$

$$\le \|\mathbf{w}_{j,r}^{(0)}\|_2 + \gamma_{j,r}^{(t)} \cdot \|\boldsymbol{\mu}\|_2^{-1} + \Theta(P^{-1}\sigma_p^{-1}d^{-1/2}n^{-1/2}) \cdot \sum_{i=1}^{n} \overline{\rho}_{j,r,i}^{(t)}$$

$$= \Theta(\sigma_0\sqrt{d}) + \Theta(P^{-1}\sigma_p^{-1}d^{-1/2}n^{-1/2}) \cdot \sum_{i=1}^{n} \overline{\rho}_{j,r,i}^{(t)}, \tag{77}$$

where the first inequality is due to the triangle inequality, and the equality is due to the following comparisons:

$$\frac{\gamma_{j,r}^{(t)} \cdot \|\boldsymbol{\mu}\|_2^{-1}}{\Theta(P^{-1}\sigma_p^{-1}d^{-1/2}n^{-1/2}) \cdot \sum_{i=1}^{n} \overline{\rho}_{j,r,i}^{(t)}} = \Theta(P^{-1}\sigma_p d^{1/2}n^{1/2}\|\boldsymbol{\mu}\|_2^{-1}\text{SNR}^2)$$

$$= \Theta(P^{-1}\sigma_p^{-1}d^{-1/2}n^{1/2}\|\boldsymbol{\mu}\|_2)$$

$$= O(1)$$

based on the coefficient order $\sum_{i=1}^{n} \overline{\rho}_{j,r,i}^{(t)}/\gamma_{j,r}^{(t)} = \Theta(\mathrm{SNR}^{-2})$, the definition $\mathrm{SNR} = \|\boldsymbol{\mu}\|_2/(\sigma_p\sqrt{d})$, and the condition for $d$ in Condition 3.1; and also $\|\mathbf{w}_{j,r}^{(0)}\|_2 = \Theta(\sigma_0\sqrt{d})$ based on Lemma D.7. With this and (73), we analyze the key component in (80):

$$
\begin{aligned}
\frac{\sum_r \sigma(\langle \mathbf{w}_{\widehat{y},r}^{(t)}, \widehat{y}\boldsymbol{\mu}\rangle)}{(P-1)\sigma_p \sum_{r=1}^{m} \big\|\mathbf{w}_{-\widehat{y},r}^{(t)}\big\|_2} &\geq \frac{\Theta(1)}{\Theta(\sigma_0\sqrt{d}) + \Theta(P^{-1}\sigma_p^{-1}d^{-1/2}n^{-1/2})\cdot \sum_{i=1}^{n}\overline{\rho}_{j,r,i}^{(t)}} \\
&\geq \frac{\Theta(1)}{\Theta(\sigma_0\sqrt{d}) + O(P^{-1}\sigma_p^{-1}d^{-1/2}n^{1/2}\alpha)} \\
&\geq \min\{\Omega(\sigma_0^{-1}d^{-1/2}), \Omega(P\sigma_p d^{1/2}n^{-1/2}\alpha^{-1})\} \\
&\geq 1.
\end{aligned}
\tag{78}
$$

It directly follows that

$$
\sum_r \sigma(\langle \mathbf{w}_{\widehat{y},r}^{(t)}, \widehat{y}\boldsymbol{\mu}\rangle) - \frac{(P-1)\sigma_p}{\sqrt{2\pi}} \sum_{r=1}^{m} \big\|\mathbf{w}_{-\widehat{y},r}^{(t)}\big\|_2 > 0.
\tag{79}
$$

Now using the method in (75) with the results above, we plug (74) into (72) and then (71), to obtain

$$
\begin{aligned}
\mathbb{P}_{(\mathbf{x},\widehat{y},y)\sim\mathcal{D}}&\big(\widehat{y}f(\boldsymbol{W}^{(t)}, \mathbf{x}) \leq 0\big) \\
&\leq \mathbb{P}_{(\mathbf{x},\widehat{y},y)\sim\mathcal{D}}\bigg( \sum_r \sigma(\langle \mathbf{w}_{-\widehat{y},r}^{(t)}, \boldsymbol{\xi}\rangle) \geq (1/(P-1)) \sum_r \sigma(\langle \mathbf{w}_{\widehat{y},r}^{(t)}, \widehat{y}\boldsymbol{\mu}\rangle)\bigg) \\
&= \mathbb{P}_{(\mathbf{x},\widehat{y},y)\sim\mathcal{D}}\bigg( g(\boldsymbol{\xi}) - \mathbb{E}g(\boldsymbol{\xi}) \geq (1/(P-1)) \sum_r \sigma(\langle \mathbf{w}_{\widehat{y},r}^{(t)}, \widehat{y}\boldsymbol{\mu}\rangle) - \frac{\sigma_p}{\sqrt{2\pi}} \sum_{r=1}^{m} \big\|\mathbf{w}_{-\widehat{y},r}^{(t)}\big\|_2 \bigg) \\
&\leq \exp\bigg[ -\frac{c\Big((1/(P-1))\sum_r \sigma(\langle \mathbf{w}_{\widehat{y},r}^{(t)}, \widehat{y}\boldsymbol{\mu}\rangle) - (\sigma_p/\sqrt{2\pi})\sum_{r=1}^{m}\big\|\mathbf{w}_{-\widehat{y},r}^{(t)}\big\|_2\Big)^2}{\sigma_p^2\Big(\sum_{r=1}^{m}\big\|\mathbf{w}_{-\widehat{y},r}^{(t)}\big\|_2\Big)^2} \bigg] \\
&= \exp\bigg[ -c\bigg( \frac{\sum_r \sigma(\langle \mathbf{w}_{\widehat{y},r}^{(t)}, \widehat{y}\boldsymbol{\mu}\rangle)}{(P-1)\sigma_p \sum_{r=1}^{m}\big\|\mathbf{w}_{-\widehat{y},r}^{(t)}\big\|_2} - 1/\sqrt{2\pi} \bigg)^2 \bigg] \\
&\leq \exp(c/2\pi)\exp\bigg( -0.5c\Big( \frac{\sum_r \sigma(\langle \mathbf{w}_{\widehat{y},r}^{(t)}, \widehat{y}\boldsymbol{\mu}\rangle)}{(P-1)\sigma_p \sum_{r=1}^{m}\big\|\mathbf{w}_{-\widehat{y},r}^{(t)}\big\|_2} \Big)^2 \bigg),
\end{aligned}
\tag{80}
$$

where the second inequality is by (79) and plugging (76) into (75), the third inequality is due to the fact that $(s-t)^2 \geq s^2/2 - t^2, \forall s, t \geq 0$.

And we can get from (78) and (80) that

$$
\begin{aligned}
\mathbb{P}_{(\mathbf{x},\widehat{y},y)\sim\mathcal{D}}\big(\widehat{y}f(\boldsymbol{W}^{(t)}, \mathbf{x}) \leq 0\big) &\leq \exp(c/2\pi)\exp\bigg( -0.5c\Big( \frac{\sum_r \sigma(\langle \mathbf{w}_{\widehat{y},r}^{(t)}, \widehat{y}\boldsymbol{\mu}\rangle)}{(P-1)\sigma_p \sum_{r=1}^{m}\big\|\mathbf{w}_{-\widehat{y},r}^{(t)}\big\|_2} \Big)^2 \bigg) \\
&\leq \exp\Big( \frac{c}{2\pi} - C\min\{\sigma_0^{-2}d^{-1}, P\sigma_p^2 dn^{-1}\alpha^{-2}\} \Big) \\
&\leq \exp\Big( -0.5C\min\{\sigma_0^{-2}d^{-1}, P\sigma_p^2 dn^{-1}\alpha^{-2}\} \Big) \\
&\leq \epsilon,
\end{aligned}
$$

where $C = O(1)$, the last inequality holds since $\sigma_0^2 \leq 0.5Cd^{-1}\log(1/\epsilon)$ and $d \geq 2C^{-1}P^{-1}\sigma_p^{-2}n\alpha^2\log(1/\epsilon)$.

$\square$

