# OpenReview forum: "Why Does Sharpness-Aware Minimization Generalize Better Than SGD?"
_NeurIPS.cc/2023/Conference — NeurIPS 2023 poster_

### Official Review · Reviewer_EhwP · 2023-06-16

**Soundness:** 3 good
**Presentation:** 3 good
**Contribution:** 4 excellent
**Rating:** 7
**Confidence:** 4

**Summary:**

This paper presents an in-depth theoretical examination of Sharpness-Aware Minimization (SAM) in the context of feature learning. The authors identify the challenge of overfitting in the training of these networks, a problem that becomes more prominent as the model size increases. The traditional gradient-based methods like Gradient Descent (GD) and Stochastic Gradient Descent (SGD) are identified to suffer from unstable training and harmful overfitting. The authors highlight SAM as a promising alternative that has demonstrated improved generalization even in situations where label noise is present. However, they point out the current lack of understanding on why SAM outperforms SGD, especially in nonlinear neural networks and classification tasks.

The core contribution of this paper is filling this knowledge gap by providing theoretical reasoning on why SAM has better generalization than SGD, particularly in the context of two-layer convolutional ReLU networks. The authors characterize the conditions under which benign overfitting can occur when using SGD, and further demonstrate a phase transition phenomenon unique to SGD. They formally prove that under these same conditions, SAM can achieve benign overfitting, and therefore has a distinct advantage in terms of generalization error. Specifically, the paper highlights the ability of SAM to mitigate noise learning, which prevents it from succumbing to harmful overfitting early on in the training process. This aspect of SAM allows it to facilitate more effective learning of weak features. The theoretical findings are supported by experiments on both synthetic and real data, bolstering the credibility of the presented theory.

**Strengths:**

- **Addressing a Significant Issue**: The authors tackle the important issue of overfitting in the training of large neural networks. They delve into a key challenge facing the field of deep learning, thus making their work relevant and timely.

- **Theoretical Contributions**: This paper provides a strong theoretical analysis of why Sharpness-Aware Minimization (SAM) outperforms Stochastic Gradient Descent (SGD), particularly for two-layer convolutional ReLU networks. This contributes to the current understanding of how to improve the generalization of large neural networks.

- **Comprehensive Study**: The authors carry out an in-depth comparison of SAM and SGD, using both synthetic and real data. This robust approach adds to the validity and comprehensiveness of their findings.

- **Novelty**: The authors claim that this is the first benign overfitting result for a neural network trained with mini-batch SGD. This claim, if validated, could signify a novel contribution to the field.

- **Clarity and Organization**: The paper appears to be well-structured and clearly written, with a good overview of the background literature, a clear statement of the problem, and a detailed account of the authors' contributions.


**Weaknesses:**

- **Limited Scope**: The paper's focus on two-layer convolutional ReLU networks with fixed second layer might limit the generalizability of its findings to other types of networks. Further research could be needed to determine if SAM has similar benefits for different architectures.

- **Presentation Could Be Enhanced**: The current version of the paper could benefit from further polishing in terms of presentation. For instance, the y-axis in Figure 2 lacks clarity — it's not immediately evident whether it represents a normal scale or a logarithmic scale. Also, the 'dimension' it not clear. The term 'clear samples' could be more accurately represented as 'clean samples.' Additionally, there's inconsistency in the usage of 'P-1' and 'P', which should ideally be standardized throughout the paper for improved readability and comprehension.

- **Partial Theoretical Results**: The authors provide a thorough analysis of both benign and harmful overfitting scenarios for SGD. However, the corresponding harmful overfitting regime for SAM appears to be absent from the theoretical results, which could make the comparison somewhat incomplete.

**Questions:**

Could you please share your thoughts on the potential behavior of SAM within the gray region depicted in Figure 1?


**Limitations:**

I believe the authors could provide a more extensive evaluation of potential limitations inherent in their research. Engaging in a comprehensive discussion regarding these constraints would strengthen the overall rigor and transparency of the study.

---

> ### Author Rebuttal · Authors · 2023-08-10
>
> Thank you for your strong support. Below, we provide answers to your comments and questions, and we will ensure that the revisions are made in the final draft.
>
> **Q1**. Further research could be needed to determine if SAM has similar benefits for different architectures.
>
> **A1**. Thank you for your constructive feedback. Since this is the first paper towards formally understanding why SAM outperforms SGD for ReLU networks, we choose to avoid introducing unnecessary complexities in the data/CNN models and the analysis to make the result easy to follow. This ensures clarity and ease of understanding. Exploring more complex data/CNN models certainly presents an exciting avenue for future research, and we plan to investigate it in subsequent studies.
>
> ---
>
> **Q2**. The y-axis in Figure 2 lacks clarity. There's inconsistency in the usage of 'P-1' and 'P'.
>
> **A2**. Thank you for your valuable feedback, the y-axis represents a normal scale with a range of 1000-21000. We will address the issues mentioned above in our final version. As for the usage of ‘P-1’ and ‘P’, the data has P patches, where P-1 patches among them are noise. Therefore, ‘P-1’ and ‘P’ both exist in our paper. However, we understand this may cause confusion and will find a way for clear expression in our final version.
>
> ---
>
> **Q3**. The corresponding harmful overfitting regime for SAM appears to be absent from the theoretical results, which could make the comparison somewhat incomplete.
>
> **A3**. The main focus of our paper is not to provide a complete characterization of benign/harmful regions of SAM but to show that SAM has a larger benign overfitting region than SGD (Figure 1). This suffices to explain why SAM can generalize better than SGD. We leave the investigation of harmful overfitting regime for SAM as a future work.
>
> ---
>
> **Q4**. Could you please share your thoughts on the potential behavior of SAM within the gray region depicted in Figure 1?
>
> **A4**. We conjecture that SAM generalizes badly on the gray region due to our empirical results in Figure 2 where the blue color represents high test error. It is an interesting future direction to prove the phase transition between harmful overfitting and benign overfitting for the SAM algorithm.

---

> > ### Comment · Reviewer_EhwP · 2023-08-14
> >
> > Thank you for your rebuttal to my comments and questions. I appreciate the time and effort you have put into addressing each of my concerns.
> >
> > As a final step in my review, I would like to request access to the code of the experiments you used during the rebuttal. This will allow me to further verify the results and ensure the integrity of the conclusions drawn.
> >
> > Once again, thank you for your detailed responses. I am satisfied with the reponses you have proposed, and I believe they will enhance the quality and clarity of the paper.

---

> > > ### Author Response · Authors · 2023-08-14
> > >
> > > Thank you very much for your positive feedback! In compliance with the rebuttal and discussion guideline, we have made the code available at an anonymous link and sent it to the Area Chair. We anticipate that you will get access to the code very soon. Once again, we really appreciate your valuable comments and suggestions, which help us improve our work!

---

### Official Review · Reviewer_CEVG · 2023-06-29

**Soundness:** 4 excellent
**Presentation:** 4 excellent
**Contribution:** 4 excellent
**Rating:** 8
**Confidence:** 4

**Summary:**

This paper studies the question of why SAM generalizes better than SGD on a specific binary classification task. The task looks like a special case of the sparse coding model where the relevant parameter is a signal-to-noise ratio (SNR). The authors present theoretical results on the performance of SAM and SGD for this specific task. More specifically, their main results say that SGD requires higher SNR in the data distribution to generalize, while SAM can generalize even with smaller SNR.

**Strengths:**

- The paper has a very comprehensive end-to-end analysis for neural network learning, which is usually very technically complicated.
- The paper is overall well-written and it was easy to follow.
- Figure 1 and Figure 2 have very nice resemblance.

**Weaknesses:**

- Discussion of the results seems not quite comprehensive; see the comments below.

**Questions:**

- The binary classification considered in this work seems to be an instantiation of the well known sparse coding problem. It would be helpful to motivate the model by presenting connection to the well known sparse coding problem.

- Connection to large learning rates GD/SGD. Recently there have been several works trying to understand why large learning rate GD/SGD generalize better than small learning rate ones. It would be clarifying to see if that's also the case for the model considered in this work. It seems that there are recent works that show that large learning rates help with generalization for a sparse coding problem (e.g. [1]). Given this, it would be a nice clarification to test at least empirically if one can achieve SAM's performance with larger learning rate GD/SGD.

- Related to the comment about the large learning rate, Q. Is the result for SGD also true for Gradient Flow (continuous dynamics of SGD)?

- The theoretical results show that SAM updates help prevent memorizing the spurious feature (or harmful overfitting) during the early stage of the training. For the model in this paper, does that mean you can switch from SAM to SGD after a few epochs and basically achieve the same performance as SAM?

- The required perturbation radius ($\tau$) in theory scales inversely with $\sqrt{d}$, which seems much smaller than what people choose in practice. Does this requirement consistent with the experiments in the paper? I think it's an important point since a sufficiently large $\tau$ distinguishes SAM from SGD.

- The original motivation for SAM was to encourage the landscape around the solution to be more flat. Does that intuition also hold true for the sparse coding problem considered in this work? Does the prevention of harmful memorization lead to a flatter landscape when you consider the flatness around the iterates of SAM?

- Does the oscillation of SAM (dynamics characterized in the previous works mentioned in this paper) also occur in the training dynamics of SAM for the model in this paper? If so, does the oscillation another explanation as to why SAM does not learn spurious features?

- In the experiments, it seems that only full-batch GD/SAM are tested for the synthetic model setting. Could you comment on the behavior for SGD and Stochastic SAM?

- Also, many recent works have studied un-normalized variant of SAM (usually referred to as USAM), where the ascent step normalization is removed. Are the main results for SAM also true for USAM or is the normalization necessary for the main results?

- In the related work section, it would be nice to see discussion on other techniques (large learning rates, label noise etc) as there have been many works on their effects on generalization (e.g. [1] [2] [3]).



- [1] Learning threshold neurons via the "edge of stability" (https://arxiv.org/abs/2212.07469)
- [2] SGD with Large Step Sizes Learns Sparse Features (https://arxiv.org/abs/2210.05337)
- [3] What Happens after SGD Reaches Zero Loss? --A Mathematical Framework (https://arxiv.org/abs/2110.06914)
- [4] Label Noise SGD Provably Prefers Flat Global Minimizers (https://arxiv.org/abs/2106.06530)

**Limitations:**

Please see the comments in the Question section.

---

> ### Author Rebuttal · Authors · 2023-08-10
>
> Thank you for your positive feedback! Due to space limits, we answer your major comments and questions as follows.
>
> **Q1**. It would be helpful to motivate the model by presenting a connection to the well known sparse coding problem.
>
> **A1**. Thank you for bringing our attention to the sparse coding problem. This is definitely a good way to motivate our study. We will emphasize the connection of our data model to the sparse coding problem in the revision and add the related work.
>
> ---
>
> **Q2**. Connection to large learning rates GD/SGD … It would be a nice clarification to test at least empirically if one can achieve SAM's performance with larger learning rate GD/SGD.
>
> **A2**. Thank you for pointing out the relevant studies and suggesting to test larger learning rates for GD/SGD. We will cite these works and study the effects of learning rate variations in our revision.
> Firstly, we apologize for the oversight in Section 5, where we mistakenly mentioned a learning rate of 0.1, which should be 0.01. This will be corrected in the revision.
>
> Additionally, we've conducted a detailed ablation study on the learning rate in the uploaded PDF. Specifically, we tested the implications of larger learning rates  0.1 and 1 with the same condition in Section 5. Figure 2 in the uploaded pdf indicates that slightly larger learning rates indeed boosts the generalization performance of SGD. The benign overfitting region is enlarged for learning rates of 0.1 and 1 when contrasted with 0.01. This trend resonates with the findings of the studies you've recommended. Importantly, even with this expansion, the benign overfitting region remains smaller than what is empirically observed with SAM. Our conclusion is that while SGD with a larger learning rate exhibits improved generalization, it still falls short of matching SAM's performance.
> We'll integrate these observations and provide a more comprehensive discussion on learning rate in the revision.
>
> ---
>
> **Q3**. Is the result for SGD also true for Gradient Flow (continuous dynamics of SGD)?
>
> **A3**. We require only an upper bound for the learning rate $\eta$. Therefore, our theory is applicable for any sufficiently small learning rate and should potentially be extended to gradient flow.
>
> ---
>
> **Q4**. Does that mean you can switch from SAM to SGD after a few epochs and basically achieve the same performance as SAM?
>
> **A4**. Yes, under the model studied in this paper, transitioning from SAM to SGD after certain epochs yields performance comparable to that of SAM. This observation aligns with practical findings, such as Figure 9 in Andriushchenko and Flammarion (2022) .
>
> ---
>
> **Q5**. The required perturbation radius in theory scales inversely with $\sqrt{d}$, which seems much smaller than what people choose in practice.
>
> **A5**. The required perturbation radius in Theorem 4.1 is $\tau = \Theta(m\sqrt{B}/P\sigma_p\sqrt{d})$, which depends on many factors and also hides the constant. Take the synthetic data as an example, we have $(m \sqrt{B} / P \sigma_{p} \sqrt{d} = 10  * \sqrt{20} / (2 * 1 * \sqrt{10000}) \approx 0.22 $ which is not small at all. We choose $\tau = 0.03$ due to the constant omitted in the bound.
>
> ---
>
> **Q6**. Does the prevention of harmful memorization lead to a flatter landscape when you consider the flatness around the iterates of SAM?
>
> **A6**. Previous papers focus on the flatness (sharpness) to explain the success of SAM. Our paper provides a different perspective for why SAM generalizes better than SGD. In our current analysis, we were not able to show that the prevention of harmful memorization can make SAM converge to flat minima. Yet this is a very interesting and important question to study in the future.
>
> ---
>
> **Q7**. Does the oscillation of SAM also occur in the training dynamics of SAM for the model in this paper?
>
> **A7**. Thank you for this insightful question. Our current theory does not imply the oscillation of SAM dynamics. In our experiments, we did not visualize the dynamics of SAM iterates due to the very complex landscape of the ReLU network (2-layer ReLU on synthetic data and ResNet50/WRN-16-8 on CIFAR10). It is also not easy to visualize the neural network weights in a 2D/3D space. We plan to investigate this question in our future work.
>
> ---
>
> **Q8**. Could you comment on the behavior for SGD and Stochastic SAM?
>
> **A8**. The comparison keeps the same for SGD and SAM with smaller batch sizes. We have provided additional experiments for them. Please see Figure 1 in the uploaded pdf.
>
> ---
>
> **Q9**. Are the main results for SAM also true for USAM or is the normalization necessary for the main results?
>
> **A9**. The main results for our results relied on the normalization and we carefully characterize it in our proof. Our results do not directly apply to the unnormalized SAM.
>
> ---
>
> **Q10**. In the related work section, it would be nice to see discussion on other techniques.
>
> **A10**. Thank you for your suggestion. We will discuss these techniques and related works in our revision.
>
> ---
>
> [1] Ahn et al., "Learning threshold neurons via the" edge of stability"." arXiv preprint, 2022.
>
> [2] Andriushchenko et al. "Sgd with large step sizes learns sparse features." ICML, 2023.
>
> [3] Li et al., "What Happens after SGD Reaches Zero Loss?--A Mathematical Framework." arXiv preprint, 2021.
>
> [4] Damian et al., "Label noise sgd provably prefers flat global minimizers." NeurIPS, 2021.

---

> > ### Comment · Reviewer_CEVG · 2023-08-13
> > **Thank you for your response**
> >
> > I read the author response, and it's excellent.
> >
> > Assuming that the authors will reflect all of them in the final version, I raise my score to 8. Great work and congrats!
> >
> > Also, regarding Q9 "Are the main results for SAM also true for USAM or is the normalization necessary for the main results?," if the authors have some intelligent things to say regarding that question, please add the results/discussions in the final version. It would be very helpful for the readers in light of the following recent work: Dai et al. 2023 (See the reference below.)
> >
> > In a nutshell, this recent work seems to show that for SAM's practicality, the normalization part is **necessary**. Since the theoretical results in this work show that the normalization step is necessary for the generalization of the resulting solution, this would give a lot of intuitions for practitioners and valuable insights for the ML community!
> >
> > Dai et al. "The Crucial Role of Normalization in Sharpness-Aware Minimization" (https://arxiv.org/abs/2305.15287)

---

> > > ### Author Response · Authors · 2023-08-13
> > > **Thank you!**
> > >
> > > Thank you for raising the score and for your positive feedback! We will make sure to incorporate all the promised changes into our final version. Thank you for pointing out the additional related work on the necessity of normalization in SAM. We will discuss this work as well and articulate our insights in the final version. Indeed, understanding the role of normalization is an important problem that may gain some insights from our theoretical analysis.

---

### Official Review · Reviewer_jpeN · 2023-07-05

**Soundness:** 4 excellent
**Presentation:** 3 good
**Contribution:** 3 good
**Rating:** 6
**Confidence:** 4

**Summary:**

The paper aims to provide a theoretical basis for the superiority of SAM over SGD. Different from former explanation based on Hessian information, the authors firstly discuss the loss landscape of non-smooth neural networks like two-layer convolutional ReLU networks. Notably, the paper proves that under conditions of harmful overfitting using SGD, SAM can deliver benign overfitting and outperform SGD in terms of generalization error. The experiments are conducted to demonstrate that SAM can outperforms SGD in terms of generalization error by mitigating noise learning and enhancing the efficiency of weak feature learning.

**Strengths:**

- The motivation is clear and novel. Previous researches focused on shallow model with implicit smooth loss. As a more challenging setting, the paper studies two-layer convolutional networks with ReLU units.

- The theoretical analysis is convincing. In terms of benign overfitting, the authors discuss the phenomenon and condition in SGD firstly. In harmful overfitting, the authors theoretically prove that SAM can outperform SGD in the early stage of learning.

**Weaknesses:**

- The experiment results may not be reliable. In Section 5, the author sets the learning rate to 0.1. However, according to recent research [1] and my practical experience, I suspect the learning rate is too high and may lead the model to overfit easily. The model may not easily overfit if we use SGD with suitable learning rate. Hence, it might be better to add an experiment and discuss the influence of learning rate.

- Potential overclaiming based on the analysis and experiment. At the beginning of the paper, the authors claim that SAM can help model learn weak features effectively. However, I do not see obvious related content that can support the authors claim. I recommend the authors modify their claims or try to discuss the relationship between SAM and weak features more clearly.

[1] Andriushchenko et al. “A Modern Look at the Relationship between Sharpness and Generalization”, ICML, 2023.

-------

After reading the responses from the authors, my concerns has been well addressed. I will keep my score as weak accept.

**Questions:**

See weakness.

---

> ### Author Rebuttal · Authors · 2023-08-10
>
> Thank you for your positive feedback. We provide our responses below and will make the corresponding adjustments in the final version.
>
> **Q1**. I suspect the learning rate 0.1 is too high … It might be better to add an experiment and discuss the influence of the learning rate.
>
> **A1**. Thank you for drawing attention to the recent work [1]. We'll cite and discuss it in the related work section. Additionally, we appreciate your keen observation regarding the learning rate. We indeed used a learning rate of 0.01, not 0.1, in Section 5. We will fix this typo in the revision.
>
> Furthermore, we've conducted an ablation study on the learning rate, which is presented in the uploaded pdf. Specifically, we experimented with learning rates of 0.001, 0.01, 0.1, and 1 under the conditions described in Section 5. Figure 2 in the uploaded PDF shows that for learning rates of 0.01 and 0.001, the patterns of harmful and benign overfitting are quite similar and consistent with our Theorem 3.2. Since SGD with a small learning rate needs a longer time to converge, we didn't finish the experiments for extremely smaller learning rates due to the time limit. But given that Theorem 3.2 holds when the learning rate $\eta$ is sufficiently small, we believe that the phase transition for a smaller learning rate would also align with the results from learning rates of 0.01 and 0.001 if training longer.
>
> Interestingly, we observed that larger learning rates like 0.1 and 1 enhance SGD's generalization performance, but are still worse than SAM. The benign overfitting region expands for learning rates of 0.1 and 1 when compared with 0.01 and 0.001. This observation might be tied to the phenomenon highlighted by Reviewer CEVG. We'll incorporate these findings and further discuss the impact of learning rate variations in our revision.
>
> [1] Andriushchenko et al. “A Modern Look at the Relationship between Sharpness and Generalization”, ICML, 2023.
>
> ---
>
> **Q2**. I recommend the authors modify their claims or try to discuss the relationship between SAM and weak features more clearly.
>
> **A2**. Thank you for your feedback. We believe there may be some confusion caused by the term "weak feature". In our paper, 'weak features' refer to features when the signal-to-noise ratio is low. We will clarify this point by removing “weak” in our revision. Our claim was based on the fact that, for benign overfitting, SGD requires the norm of features to be at least in the order of $d^{1/4}\sigma_p$, while SAM only necessitates the norm of features to be almost a constant. This weaker requirement of SAM allows it to learn features more effectively compared to SGD. Figure 1 in our manuscript aims to provide an intuitive visualization of this result.

---

> > ### Comment · Reviewer_jpeN · 2023-08-21
> >
> > Thank you for the detailed response. My concerns have been well addressed, so I will keep my score as weak accept.

---

### Official Review · Reviewer_TY4G · 2023-07-05

**Soundness:** 3 good
**Presentation:** 1 poor
**Contribution:** 4 excellent
**Rating:** 5
**Confidence:** 3

**Summary:**

This paper presents two theoretical contributions regarding benign/malign overfitting of two-layer convolutional ReLU neural networks. For an idealized data distribution, it i) gives the conditions (with respect to the dimension of the data and to the signal strength) under which benign/malign overfitting occurs when training the network with mini-batch-SGD; ii) shows that training the network with SAM leads benign overfitting more often than SGD and in particular in many situations where SGD suffers from malign overfitting. These findings are illustrated by a numerical experiment conducted in the idealized situation used for the theoretical analysis.

**Strengths:**

The two contributions presented in this paper are significant because Contribution i) improves on the literature, in particular with respect to the closest work (Kou et al., 2023), which proposes exactly the same analyses but for a gradient descent training; Contribution ii) is, up to my knowledge, the first such result for SAM. The latter result provides a comprehension on the well-known fact that SAM helps networks to generalize better than SGD. This is a real breakthrough, which should be welcomed by the community.

**Weaknesses:**

I was very enthusiastic by the results presented in this paper but I became disillusioned when reading it. The presentation is disordered and in particular, notations are not all defined, making the results (even informally stated) very difficult to understand. After having a glance at the appendix (which seems to be rigorously written), it seems that the manuscript is an assembly of results extracted from the appendix, unfortunately awkwardly built. As for me, it is a pity because I think that the results are important and of interest for the community, but the presentation is inadequate and not clear enough.
Comments:
1) The abstract misses to state Contribution i) (which constitutes a large part of the manuscript and which is clearly reminded in the conclusion).
2) Definition 2.1 is difficult to understand. In particular, the parts “signal contained in each data point” and “the other are given by” are unclear. Besides, even though this model seems common in the recent literature, a discussion regarding its limitations, for instance independence between labels and covariates, would be appreciated. The same remark can be done regarding the architecture of the neural network considered (Section 2.2). In addition, Condition 3.1 is discussed only quickly Line 142 ; a deeper discussion regarding the role of the variances and the polynomial degrees appearing would be appreciated.
3) There is no link between statements/results in the paper and their counterparts in the appendix. Since the reader is supposed to juggle the two, it would be helpful to know where are the formal statements and the proofs in the appendix.
4) I understand that authors try to explain the derivation of their main results but I am not very comfortable with informal statements, first because of the notation problem previously stated and also because some conclusions are given with vague explanations (in my case they are difficult to grasp) while they seem to be quite difficult to obtain formally. This is the case for instance Line 218 regarding the symmetry of $\rho$.
5) Line 117, the authors invoke the discontinuity of the gradient to justify the need of an analysis based on an other technique than the Taylor expansion. As for me an even better argument is that the ReLU function is formally not differentiable everywhere.
6) The statement of Theorem 4.1 is a bit unclear: the parts “we train/we can train” could be replaced by passive forms, “neural networks” refers, as far as I understand, to the chosen architecture. In addition, mentioning SGD in a result concerning SAM may throw the reader. It could be specified beforehand that SAM training uses intrinsically SGD (but at a point which is not the current iterate).
7) Lemma 4.3 is true only for a particular choice of $\tau$, as stated in Theorem 4.1 (I am not sure that this is clear in Lemma C.5). Since this result is important, this should appear in Lemma 4.3 and discussed after.
8) The related work section (Section 6) appears at the end of the manuscript but is an enumeration of papers. Such an enumeration is generally well placed just after the introduction. Placing the related work section after the result statements makes sense if it discusses technical differences with the closest papers. As it happens, this work seems to be based to a certain extent on (Kou et al., 2023). Thus, it would be enriching to discuss the contribution, and particularly the technical novelties (GD → SGD/SAM), of this work with respect to (Kou et al., 2023).
9) Mathematical remarks: Line 47, $t$, $\mathbf W^{(t)}$, $\boldsymbol \mu$, $L_{\mathcal D}^{0-1}$, $p$, $\Omega$ are not defined; Lines 47, 150, 240 and after, the expression “converges to $\epsilon$” should be replaced by “converges to $0$”; Line 66, $l_2$ → $\ell_2$, Line 68, absolute values in $|a_k/b_k|$ seem useless; “omit logarithmic terms” should be defined explicitly; Line 97, $\mathbf W$ is not defined; Line 100, “is a collection of” means “is a matrix”, doesn’t it? Line 101, $[n]$ is not defined; Line 113, it is specified that $\sigma_0^2$ is the variance of the normal distribution but not Line 80; in Equation (5), the 2-norm should be a Frobienus norm; what is the utility of Equation (6) with respect to the equation Line 175? Line 121, $S_i^{(t, b)}$ → $\tilde S_i^{(t, b)}$; Line 219, $T^*$ is not defined; Lines 218 and 226, the authors could remind what are $H$ and $B$; Line 226, I understand that $|\cdot|$ is the cardinality but this is not stated.
10) Typographical remarks: Line 9, “for the certain” → “for a certain”; Line 27, “with minimal gradient” → “with minimal gradient norm”; Figure 1, blue and yellow are inverted; Line 102, “cross-entropy loss function” and “logistic loss” are redundant; Line 115, “indicator function” → “the indicator function”; Line 156, “Bayesian optimal risk” → “Bayes risk”; full points are missing Lines 202, 207, 214; Line 237: “iteration” → “iterations”; Line 251, full point instead of colon; Figure 2, y-label is cut; Line 299, “an generalization” → “a generalization”.

**Questions:**

1) Equations Line 175 and 176 are given as a definition (Definition 3.3), but appear to be a result of the data distribution and of the network architecture chosen. Why are they presented as a definition?
2) The numerical experiment is performed with gradient descent instead of SGD, while the theoretical analysis deeply relies on SGD. Why? Are results identical with SGD?
3) Similarly, are the theoretical results for SAM the same if optimization is performed with gradient descent?

**Limitations:**

See above regarding mathematical assumptions. Societal impact is not addressed.

---

> ### Author Rebuttal · Authors · 2023-08-10
>
> Thank you for your constructive feedback. Due to space limits, we will address your major comments and questions as follows. We will revise the corresponding parts accordingly as well as address the minor points in the final version.
>
> **Q1**. The abstract misses to state Contribution i).
>
> **A1**. Thank you for your suggestion. We will add the contribution i) in the abstract of our final version.
>
> ---
>
> **Q2**. Definition 2.1 is difficult to understand.  A discussion regarding its limitations, for instance independence between labels and covariates, would be appreciated.
>
> **A2**. Thank you for your feedback. We will add more explanation of Definition 2.1 and the architecture of neural networks in our final version for a clear presentation.
> Regarding the “independence between label and covariates,” we believe there is a misunderstanding. For any covariate $x=[x^{(1)},... x^{(P)}]$, there is exactly one $x^{(j)} = y\cdot\mu$, and the others are random Gaussian vectors.  For example, the covariates $x$ could be $[y\cdot\mu, \xi, \ldots, \xi]$, $[\xi,\ldots, y\cdot\mu, \xi]$ or $[\xi, \ldots, \xi, y\cdot\mu]$. The signal patch $y\cdot\mu$ can appear at any position. So $x$ and $y$ are not independent. We will make it clearer in the final version.
>
> ---
>
> **Q3**. It would be helpful to know where the formal statements and the proofs are in the appendix.
>
> **A3**. Thank you for your suggestion. We will add pointers in our final version. In particular, Theorem 1.1 is the informal statement of Theorem 3.2 and Theorem 4.1. Lemma 3.4 is the informal statement of Lemma B.8. Lemma 3.5 is the informal statement of Lemma A.6. Lemma 4.3 is the informal statement of Lemma C.5.
>
> ---
>
> **Q4**. Not very comfortable with informal statements, for instance Line 218 regarding the symmetry of $\rho$.
>
> **A4**. We’re sorry for the confusion. We utilized the term "symmetry" to describe a situation where the summation of $\bar{\rho}$ corresponding to different samples yields similar values. More precisely, the difference between these values can be bounded by a small constant: $\sum_{r=1}^{m}\zeta_{y_i,r,i}^{(t,b_1)}-\sum_{r=1}^{m}\zeta_{y_k,r,k}^{(t,b_2)}\leq\kappa$, as indicated in lines 203-204. We will remove the word  'symmetry' from the paper and revise this part accordingly.
>
>
> ---
>
> **Q5**. Lemma 4.3 is true only for a particular choice of $\tau$, as stated in Theorem 4.1.
>
> **A5**. We’re sorry for missing the condition in Lemma 4.3. It requires $\tau =  \Theta\Big(\frac{m\sqrt{B}}{P\sigma_{p}\sqrt{d}}\Big)$.
>
> ---
>
>
> **Q6**. Discuss the technical novelties of this work with respect to (Kou et al., 2023).
>
> **A6**. The key technical challenges and novelties of our work compared with Kou et al. 2023 are highlighted as follows:
>
> - We studied SGD rather than GD. As the mini-batch update of SGD only utilizes only a small part of samples, different samples contribute to the update of coefficients differently. Consequently, the noisy samples may deviate the learning process. Thus, we have developed techniques to control the update of coefficients at both batch-level and epoch-level. Specifically, Lemma B.7 bounds the batch-level update, and Lemma B.8 controls the epoch-level update and aligns the update of batch-level and epoch-level.
>
> - We also provide a novel analysis for SAM, which is very different from GD/SGD. SAM has a completely different neuron activation pattern from SGD. The activation pattern is based on the perturbed weight $\mathbf{w} + \mathbf{\epsilon}$  in SAM, rather than the ​​unperturbed weight $\mathbf{w}$ as in SGD. The perturbation $ \mathbf{\epsilon}$ introduces difficulties in the analysis. We discussed the difficulties and our techniques to tackle them in Section 4.1. More specifically, we decompose SAM updates into SGD step and perturbation step and connect them through Lemma 4.3. We show that if a neuron is activated by noises in the SGD step, it will subsequently become deactivated for the perturbation step. This technique is new and has never been used in prior works such as Kou et al., 2023. It is pivotal for the analysis of SAM.
>
> ---
>
> **Q7**. Mathematical and typographical remarks.
>
> **A7**.
> - Line 100, The phrase "a collection of" can be interpreted as a new matrix or as a tensor of the weight matrix.
>
> - Equation (6) is a variant of Line 175, which we refer to for technical clarity. By further decomposing the coefficient $\rho_{j,r,i}^{(t,b)}$ into $\overline{\rho}_{j,r,i}^{(t,b)}$ and   $\underline{\rho}\_{j,r,i}\^{(t,b)}$,  we can streamline our proof.
>
> - $T^*$ is defined in section B.1.2 as $T^* = \eta^{-1} \text{poly}(\epsilon^{-1}, d, n,m)$.
>
>
> ---
>
> **Q8**. Why Equations Line 175 and 176 are given as a definition?
>
> **A8**. Thank you for your suggestion. We will present it as a lemma instead of a definition in the revision to ensure clarity.
>
> ---
>
> **Q9**. The numerical experiment is performed with gradient descent instead of SGD, while the theoretical analysis deeply relies on SGD. Why? Are results identical with SGD?
>
> **A9**. Our results also apply to GD, since GD can be viewed as a special case of SGD where the batch size is equal to the dataset size ($B = n$). We have additional experiments of SGD (batch=1024) on the real data set in Appendix D, and it is also consistent with our theoretical analysis. We have also added an experiment of SGD with mini-batch size 10 on the synthetic data in Figure 1 of the uploaded PDF. If we compare Figure 1 of the uploaded PDF and Figure 2 of our main paper, the comparison results remain the same.
>
> ---
>
> **Q10**. Are the theoretical results for SAM the same if optimization is performed with gradient descent?
>
> **A10**. Yes, since SAM with full gradient is a special case of SAM with stochastic gradient when the batch size is equal to the dataset size (B = n), our theoretical results also hold for SAM with full gradient.

---

> > ### Comment · Reviewer_TY4G · 2023-08-11
> >
> > I would like to thank the authors for their rebuttal, which I read carefully. I acknowledge that authors accept to change the paper presentation according to my comments and I agree to increase my score consequently. Since I do not have access to the revised version, the score is increased of one level.

---

> > > ### Author Response · Authors · 2023-08-11
> > > **Thank you!**
> > >
> > > Thank you for raising the score and for your constructive comments! We will be sure to incorporate your suggested changes and revise our paper accordingly in the final version.

---

### Author Rebuttal · Authors · 2023-08-10


We want to thank all the reviewers for their valuable comments. In the uploaded pdf, we include additional empirical results to address reviewers' concern including Figures 1 and 2.

+ **Figure1**:  To address Reviewer TY4G’s concern that our synthetic experiment is only performed with gradient descent instead of SGD,  we have added an experiment of SGD with mini batch size 10 on the synthetic data. If we compare this Figure 1 of the uploaded PDF and Figure 2 of our main paper, the comparison results should remain the same and both support our main Theorems 3.2 and 4.1.

+ **Figure2**: Reviewers jpeN and CEVG raise an intriguing direction that whether tuning ​​learning rate can get better generalization performance and achieve SAM's performance.  Therefore, We've conducted an extended study on the learning rate. Specifically, we experimented with learning rates of 0.001, 0.01, 0.1, and 1 under the same conditions described in Section 5. The results show that for all learning rates, the patterns of harmful and benign overfitting are quite similar and consistent with our Theorem 3.2. We observed that larger learning rates such as 0.1 and 1 can enhance SGD's generalization performance. This observation might be related to the phenomenon pointed out by Reviewer CEVG.  Importantly, for all learning rates, the benign overfitting regions remain smaller than what is empirically observed with SAM. Our conclusion is that while SGD with a more considerable learning rate exhibits improved generalization, it still falls short of matching SAM's performance.

---

### Decision · Program_Chairs · 2023-09-21

**Decision:**

Accept (poster)

**Comment:**

The paper tackles an important question of understanding SAM vs SGD. The reviewers are all positive about the paper and authors response was well received. We encourage the authors to clean up the presentation, connect better with the appendix, provide concrete definitions and formal results alongside high level intuition and pay careful attention to typos. Please also include the discussion and experiments on the effect of learning rate.